# Entropy Guided Semi-Supervised Graph Coarsening

**Subhanu Halder**                                   *eez212381@iitd.ac.in*
*Department of Electrical Engineering*
*IIT Delhi, India*

**Manoj Kumar**                                   *manoj.kumar@mfs.iitr.ac.in*
*Mehta Family School of DS and AI, IIT Roorkee, India*

**Sandeep Kumar**                                   *ksandeep@iitd.ac.in*
*Department of Electrical Engineering*
*Yardi School of AI*
*Bharti School of Telecommunication Technology and Management*
*IIT Delhi, India*

**Reviewed on OpenReview:** *https://openreview.net/forum?id=xZAwzxUP7c*

## Abstract

Graphs are foundational abstractions in data-intensive domains, yet the scale of modern datasets strains computation and memory for downstream learning. From recommender systems to biological networks, graphs have emerged as a fundamental substrate for learning. As graph sizes grow, the cost of training and inference becomes prohibitive, thereby necessitating compact surrogates that retain spectral properties and feature semantics. We propose an entropy-regularized, semi-supervised framework for attributed graph coarsening that jointly leverages the original graph's Laplacian, node features, and partially observed labels. Central to our approach is an information-theoretic regularizer that minimizes the per-supernode Shannon entropy of the node-profile matrix $\phi = C^T Y$, encouraging label-coherent aggregations. We formulate a principled objective that balances structural fidelity and feature alignment, and we solve it using an efficient block MM/BSUM algorithm. We establish learning and show structural guarantees that control Dirichlet-energy distortion, and bound deviations in cut costs and effective resistances. Experiments on standard benchmarks (e.g., Cora, Citeseer, PubMed, coauthor-CS) demonstrate that our method produces a node-profile matrix with low row-wise entropy, in which nodes with the same label are grouped into the same supernode, and achieves competitive node-classification accuracy and link-prediction performance across multiple GNN backbones (GCN, GAT, APPNP), while remaining computationally efficient.

## 1 Introduction

Graphs provide a natural language for representing interactions among entities, and they arise in a wide variety of applications including citation analysis, biological systems, social networks, recommendation, and scientific data analysis (Battaglia et al., 2018; Wu et al., 2020; Zhou et al., 2020; Bruna et al., 2013; Chen et al., 2020; Defferrard et al., 2016; Kumar et al., 2020). As these graphs become larger, however, both storage and computation become increasingly demanding, especially when the downstream goal involves repeated training, inference, or structural analysis on the graph itself. This has motivated the emergence of graph dimensionality reduction techniques, such as graph coarsening (Loukas & Vandergheynst, 2018; Loukas, 2019; Kumar et al., 2023a;b;a; Dorfler & Bullo, 2011; Ron et al., 2010; Hendrickson et al., 1995), condensation (Jin et al., 2021), or graph summarization (Riondato et al., 2017), whose objective is to replace a large graph by a smaller surrogate that remains useful for subsequent learning tasks.

Among these approaches, graph coarsening seeks a reduced graph obtained by aggregating groups of original nodes into supernodes while retaining important structural information. Existing methods differ in the type of information they aim to preserve: some are primarily spectral or heuristic (Loukas & Vandergheynst, 2018; Loukas, 2019), some explicitly incorporate node features (Kumar et al., 2024; 2023a;b), and others adopt deep learning-based strategies, such as the graph neural network for coarsening proposed in (Jin et al., 2021). Despite this progress, existing methods do not directly control label mixing within supernodes, which can degrade downstream classification.

Most graph coarsening methods rely on the graph Laplacian, with or without node features, and infer labels on the coarsened graph via majority voting within supernodes. While label-informed coarsening approaches exist, they are largely deep learning–based and require expensive training on the original large graph through gradient matching or bi-level optimization, thereby failing to eliminate the underlying computational bottleneck and offering limited flexibility.

Optimization-based coarsening frameworks are more scalable and interpretable but typically ignore label structure, focusing instead on sparsity in the node profile matrix. Such sparsity alone does not guarantee label purity within supernodes. In this work, we incorporate label information directly into the coarsening process by regularizing the entropy of the node profile matrix (Section 2.3, (Ghoroghchian et al., 2021)). Unlike Frobenius-norm regularization (Kumar et al., 2024), entropy minimization explicitly enforces row-wise concentration. This objective encourages nodes with the same label to be assigned to the same supernode, thereby yielding coarsenings that are both computationally efficient and better aligned with downstream classification.

Moreover, prior label-aware coarsening methods (Kumar et al., 2024) did not provide explicit guarantees on the extent to which the coarse graph, when lifted back to the original node space, preserves key structural quantities. Unlike Frobenius penalties that encourage a small Frobenius norm $\|\phi\|_F^2$, our entropy term acts on the normalized row-wise label distributions $\phi_i/N_i$, thereby directly penalizing label mixing within each supernode. Concretely, minimizing $\sum_i H\left(\frac{\phi_i}{\|\phi_i\|_1}\right)$ promotes the merging of nodes with the same label into the same supernode and provides a transparent *regularization parameter* to balance supervision against coarsening strength; by contrast, quadratic penalties may still yield supernodes containing nodes from different labels and with small $\ell_2$ error due to majority-class dominance. Finally, we provide such guarantees by deriving three bounds tied to the mapping from original nodes to supernodes: (i) an upper bound on the difference in Dirichlet energy between the original and the lifted coarse graphs, certifying that the lifted coarse Laplacian remains close to the original; (ii) an upper bound on cut preservation, formalizing the view of coarsening as clustering so that cut costs are not overly distorted; and (iii) an upper bound on the deviation of the Laplacian and its pseudoinverse after lifting, which controls how far the lifted operator can drift from the original.

Entropy drives the formation of each supernode, which consists of nodes that share the same label, thereby reducing the projection error and improving the stability/conditioning of the mapping. We show that the error term becomes smaller under an entropy regularizer than under a Frobenius norm regularizer, and the resulting relative cut distortion is correspondingly lower. By contrast, a Frobenius regularizer tends to spread each supernode across many labels, mixing classes and increasing the projection error, thereby yielding looser guarantees. The main contributions of this paper are: **(i) Entropy–Based Regularizer**: We develop a framework that uses an entropy regularizer to improve downstream performance. **(ii) Theoretical guarantees**: We theoretically establish that entropy regularization produces sparser rows of the node profile matrix in contrast to Frobenius norm-based regularization. Beyond this, we derive an explicit upper bound on the Dirichlet energy difference between the original Laplacian and the lifted Laplacian. This also leads to bounds on cut preservation and effective resistance. Finally, we show how entropy controls misclassification. **(iii) Evaluating Label mixing:** introduce three metrics, SII (Supernode Impurity Index) to count how many different labeled nodes are clustered into a supernode, AMM(Average Majority Miss) is the average error if we label each supernode by its majority class, and EIEC(Entropy-Induced Error Ceiling) as an entropy-based upper bound to check consistency with theory.

**Outline.** The remainder of the paper is organized as follows. Section 2 reviews the graph-learning and coarsening preliminaries needed for our development, and introduces the probabilistic node-profile matrix that motivates our entropy-based regularizer. Section 3 presents the ENGC optimization problem and the corresponding MM/BSUM updates. Section 4 develops the theoretical analysis of the proposed method.

Section 4.5 introduces label-mixing evaluation metrics. Section 5 reports experimental results on benchmark datasets. Additional proofs, implementation details, and supplementary experiments are deferred to the appendix.

## 2 Background and Problem Formulation

In this section, we cover the fundamentals of graphs, graph coarsening, node profile matrix, and entropy. Furthermore, we provide details on how entropy is used for grouping and aggregating nodes.

### 2.1 Graphs and Graph Learning from Data

We consider an undirected graph $\mathcal{G} = (V, E, A, X, Y)$, where $V = \{v^1, \ldots, v^p\}$ is the set of nodes, $E \subseteq V \times V$ is the set of edges, and $A \in \mathbb{R}^{p \times p}$ is the adjacency matrix, with $A_{ij} \neq 0$ indicating an edge between nodes $i$ and $j$. Each node $v_i$ is associated with a feature vector $X_i \in \mathbb{R}^f$, and the feature matrix is denoted by $X = [X_1, \ldots, X_p]^T \in \mathbb{R}^{p \times f}$. In the semi-supervised setting, label information is given by $Y \in \{0, 1\}^{p \times l}$, where a labeled node has a one-hot label vector and an unlabeled node has a zero vector. Graphs are commonly represented using either the adjacency matrix or the combinatorial Laplacian matrix $\Theta \in \mathbb{R}^{p \times p}$, defined as $\Theta_{ij} = \Theta_{ji} \leq 0$ for $i \neq j$ and $\Theta_{ii} = -\sum_{j \neq i} \Theta_{ij}$ (Kumar et al., 2020), with the relation $A_{ij} = -\Theta_{ij}$ for $i \neq j$. The Laplacian matrix is symmetric, positive semidefinite, and has zero row sums, which makes it convenient for graph learning and coarsening. Given node features $X$, we learn a connected graph Laplacian by solving the optimization problem in (Kalofolias, 2016):

$$\underset{\Theta \in \mathcal{S}_\Theta}{\text{minimize}} -\gamma \log(\det(\Theta + J)) + \text{tr}(X^T \Theta X) + \beta h(\Theta) \tag{1}$$

which balances feature smoothness, structural regularization, and graph connectivity through a log-determinant term. For graph coarsening, we define the coarse Laplacian as $\Theta_c = C^\top \Theta C$ and the lifted Laplacian as $\widetilde{\Theta} = C \Theta_c C^\top$, which will be used in the following sections.

### 2.2 Graph Coarsening

Given a graph $\mathcal{G}(\Theta, X, Y)$ with $p$ nodes, graph coarsening constructs a reduced graph $\mathcal{G}_c(\Theta_c, \tilde{X}, Y_c)$ with $k \ll p$ nodes while preserving structural and spectral properties. The coarse Laplacian $\Theta_c \in \mathbb{R}^{k \times k}$, feature matrix $\tilde{X} \in \mathbb{R}^{k \times f}$, and label matrix $Y_c \in \mathbb{R}^{k \times l}$ are related to their counterparts by: $\Theta_c = C^\top \Theta C$, $X = C\tilde{X}$, $Y_c = \arg\max(C^T Y)$, where $C \in \mathbb{R}_+^{p \times k}$ is a node-to-supernode mapping matrix. To ensure valid aggregation, $C$ is constrained to

$$\mathcal{C} = \left\{ C \geq 0 \mid \langle C_i, C_j \rangle = 0 \ \forall i \neq j, \ \langle C_i, C_i \rangle = d_i, \|C_i\|_0 \geq 1, \ \|[C^\top]_i\|_0 = 1 \right\}, \tag{2}$$

which enforces non-overlapping assignments and prevents a node from belonging to two different supernodes and $d_i$ is some positive number. We define the reconstruction error matrix as

$$\Delta := \Theta - C \Theta_c C^\top, \tag{3}$$

so that $x^\top \Delta x$ quantifies the error for any $x \in \mathbb{R}^p$.

### 2.3 Probabilistic Node-Profile Matrix

A central object in our formulation is the *node-profile matrix* (Kumar et al., 2024; Ghoroghchian et al., 2021), which summarizes how labels from the original graph are distributed across supernodes in the coarsened graph. Let $C \in \mathbb{R}_+^{p \times k}$ denote the node-to-supernode mapping matrix and let $Y \in \{0, 1\}^{p \times \ell}$ be the partially observed label matrix. We define

$$\phi = C^\top Y \in \mathbb{R}_+^{k \times \ell}. \tag{4}$$

For each supernode $i$ and class $j$, the entry $\phi_{ij}$ records the aggregated amount of label-$j$ assigned to supernode $i$. Thus, each row of $\phi$ describes the class composition of a supernode.

While several existing coarsening methods implicitly aim to control the structure of $\phi$, many rely on objectives such as $\|C^T Y\|_F^2$ (Kumar et al., 2024) , which primarily penalize the magnitude of entries rather than their distribution. Since these objectives do not directly penalize label mixing, which can result in mixed-label supernodes, even though the Frobenius norm remains small.

We introduce the probabilistic node-profile matrix $\phi'$, where each row in $\phi'$ is defined as:

$$[\phi']_{i:} = \frac{\phi_{i:}}{\|[\phi^\top]_i\|_1} \text{ if } \|[\phi^\top]_i\|_1 > 0, \quad [\phi']_{i:} = \mathbf{0}_\ell^\top \text{ otherwise.} \tag{5}$$

Each entry $\phi'$ denoted by $[\phi']_{ij}$ is the probability that a node assigned to the $i$-th supernode has label $j$. $[\phi']_{ij}$ can be interpreted as the conditional probability $[\phi']_{ij} = \mathbb{P}(Y = j \mid \text{node assigned to supernode } i)$

In the context of the original graph shown in Figure 1, there are two coarsened graphs, associated with mapping matrices $C_1$ and $C_2$, and the corresponding probabilistic node-profile matrices $\phi'_1$ and $\phi'_2$ are:

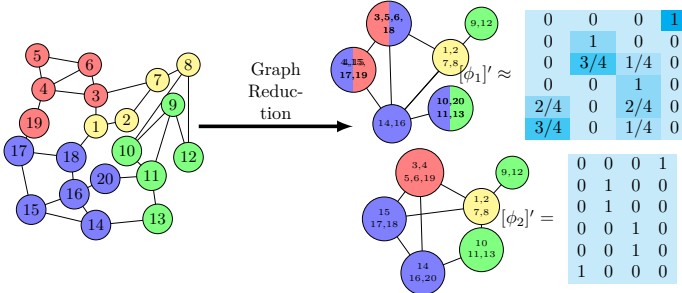

Figure 1: Example of two coarsenings of the same original graph, together with their induced probabilistic node-profile matrices. Compared with the top coarsening, the bottom one produces more label-consistent supernodes, leading to a lower row-wise entropy

The probabilistic node profile matrix $\phi'$ therefore provides a principled way to quantify how nodes with different labels are distributed within each supernode. Since semantic fidelity requires these distributions to be as concentrated as possible, we next employ entropy as an information-theoretic measure to quantify and penalize uncertainty in $\phi'$. This motivates the entropy-based formulation introduced in the following section.

The entropy of a discrete random variable $\mathcal{X} = \{x_1, x_2, \ldots, x_n\}$ with a probability distribution $P = \{p_1, p_2, \ldots, p_n\}$, where $p_i$ represents the probability of each sample $x_i$, is given by the Shannon entropy $H(X)$. It is defined as:

$$H(X) = -\sum_{i=1}^{n} p_i \log p_i \tag{6}$$

Here, $H(X)$ quantifies the uncertainty or information content associated with the random variable $\mathcal{X}$. The entropy of the $i$-th row of the matrix $\phi'$, denoted as $H([(\phi'_i)])$, quantifies the uncertainty associated with the distribution of nodes from the original graph, each labeled $1, 2, \ldots, l$, that are mapped to the $i$-th supernode. This row, $[(\phi'_i)] = [\phi'_{i1}, \phi'_{i2}, \ldots, \phi'_{il}]$, represents the probabilities $\phi'_{ij}$ of nodes with the $j$-th label being mapped to the $i$-th supernode. For a supernode with positive labeled mass, i.e., $\|[\phi^\top]_i\|_1 > 0$, we define $H([\phi'_i]) = -\sum_{j=1}^{\ell} \phi'_{ij} \log \phi'_{ij}$, which measures the uncertainty of the observed label distribution in that supernode. For $\|[\phi^\top]_i\|_1 = 0$, we set $[\phi']_{i:} = 0_\ell^\top$, as in Eq (5).

As illustrated in Figure 1, the probabilistic node-profile matrix corresponding to the two coarsenings $\phi'_1$ and $\phi'_2$ and the corresponding entropy vectors $H(\phi'_1)$ and $H(\phi'_2)$ are:

$$\phi'_1 = \begin{bmatrix} 0 & 0 & 0 & 1 \\ 0 & 1 & 0 & 0 \\ 0 & \frac{3}{4} & \frac{1}{4} & 0 \\ 0 & 0 & 1 & 0 \\ \frac{1}{2} & 0 & \frac{1}{2} & 0 \\ \frac{3}{4} & 0 & \frac{1}{4} & 0 \end{bmatrix} \quad \phi'_2 = \begin{bmatrix} 0 & 0 & 0 & 1 \\ 0 & 1 & 0 & 0 \\ 0 & 1 & 0 & 0 \\ 0 & 0 & 1 & 0 \\ 0 & 0 & 1 & 0 \\ 1 & 0 & 0 & 0 \end{bmatrix} \quad H(\phi'_1) = \begin{bmatrix} 0 \\ 0 \\ -\left(\frac{3}{4}\log\frac{3}{4} + \frac{1}{4}\log\frac{1}{4}\right) \\ 0 \\ \log 2 \\ -\left(\frac{3}{4}\log\frac{3}{4} + \frac{1}{4}\log\frac{1}{4}\right) \end{bmatrix} = \begin{bmatrix} 0 \\ 0 \\ 0.56 \\ 0 \\ 0.69 \\ 0.56 \end{bmatrix} \quad H(\phi'_2) = \begin{bmatrix} 0 \\ 0 \\ 0 \\ 0 \\ 0 \\ 0 \end{bmatrix}$$

$H(\phi_1')$ contains multiple nonzero entries, indicating that nodes with different labels have been merged into some supernodes, whereas $H(\phi_2') = 0$ for all rows, implying supernodes in the second graph are composed of nodes with the same label. Lower row-wise entropy therefore corresponds to merging of nodes sharing the same label and a more informative coarsened graph.

We propose entropy as a structural regularizer to control label mixing within supernodes. By penalizing uncertainty in the aggregated label distribution, entropy regularization promotes label-homogeneous clusters and discourages mixed-label supernodes. In Section 4, we prove how the entropy term affects the coarsening objective and theoretical bounds. We now proceed to the algorithmic development.

## 3 Proposed Entropy Guided Graph Coarsening Algorithm

In this section we present the algorithmic development with the objective of reducing the row-wise entropy of the node profile matrix while preserving semantic and structural properties of the graph. Let $\mathcal{I}_\phi$ denote the set of supernodes with positive labeled mass: $\mathcal{I}_\phi = \left\{ i \in \{1, \dots, k\} \; : \; \|[\phi^\top]_i\|_1 > 0 \right\}$. To this end, we adopt an entropy-based regularizer:

$$r_E(C, Y) = - \sum_{i \in \mathcal{I}_\phi} \sum_{j=1}^{\ell} \phi_{ij}' \log \phi_{ij}' \tag{7}$$

which encourages a low-entropy node-profile matrix by reducing the entropy of each row. Now, we formally propose our problem formulation below.

### 3.1 Proposed Formulation

Consider a graph $\mathcal{G}(\Theta \in \mathbb{R}^{p \times p}, X \in \mathbb{R}^{p \times f}, Y)$, where $Y \in \{0, 1\}^{p \times l}$ denotes the node-label matrix. For each node $v^i$, the row vector $\mathbf{y}_i$ is taken to be the corresponding one-hot label vector when the node is labeled, and $\mathbf{y}_i = 0$ when the node is unlabeled, consistent with the semi-supervised setting. Based on this representation, the coarsened graph is learned through the following optimization problem:

$$\underset{\Theta_c, \tilde{X}, C}{\text{minimize}} \, r_E(C, Y) - \gamma \log \det(\Theta_c + J) + \text{tr}(\tilde{X}^T \Theta_c \tilde{X}) + \beta h(\Theta_c) + \frac{\lambda}{2} g(C) \tag{8}$$

$$\text{s.t. } C \in \mathcal{S}_c, \Theta_c = C^T \Theta C, \ X = C\tilde{X}, \ \Theta_c \in \mathcal{S}_\Theta$$

Where $\mathcal{S}_c = \{C \in \mathbb{R}_+^{p \times k}, \|[C^\top]_i\|_2^2 \leq 1 \quad \forall i = 1 \dots p,\}$ and $\mathcal{S}_\Theta = \{\Theta \in \mathbb{R}^{p \times p}| \; \Theta_{ij} = \Theta_{ji} \leq 0 \text{ for } i \neq j, \quad \Theta_{ii} = -\sum_{j \neq i} \Theta_{ij}\}$. Here, $-\log \det(\Theta_c + J)$ ensures the connectedness of the coarsened graph where, $J = \frac{1}{k}\mathbf{1}_{k \times k}$ is a rank-one matrix with each element equal $\frac{1}{k}$. Directly enforcing the hard constraint $X = C\tilde{X}$ is challenging in optimization. Consequently, we relax it through the penalty term $\|C\tilde{X} - X\|_F^2$. The regularizer $h(\Theta_c) = \|\Theta_c\|_F^2$ encourages sparsity in the coarsened graph, whereas $g(C) = \|C^\top\|_{1,2}^2$ induces structured assignments in which each node is associated with a single supernode and each supernode is assigned at least one node (Kumar et al., 2024). Substituting $\Theta_c = C^\top \Theta C$ into Eq (8) reduces the original three-variable optimization problem to an equivalent problem involving only two variables, given by:

$$\underset{\tilde{X}, C \in \mathcal{S}_c}{\text{minimize}} \quad r_E(C, Y) - \gamma \log \det(C^\top \Theta C + J) + \text{tr}(\tilde{X}^\top C^\top \Theta C \tilde{X})$$

$$+ \frac{\alpha}{2} \|C\tilde{X} - X\|_F^2 + \frac{\lambda}{2} \|C^\top\|_{1,2}^2 + \frac{\beta}{2} \|C^\top \Theta C\|_F^2. \tag{9}$$

The feasible set $\mathcal{S}_c$ is used as a continuous relaxation of the hard assignment set $\mathcal{C}$ in Eq (2). Directly optimizing over $\mathcal{C}$ is combinatorial due to the non-overlap and one-hot assignment constraints. Following optimization-based graph coarsening (Kumar et al., 2023a), we therefore optimize a soft nonnegative assignment matrix $C$. After convergence, the relaxed mapping matrix $C$ is typically row-sparse: one entry in each row becomes dominant while the others are small satisfying the property of $\mathcal{C}$. The first term is incorporated to reduce the entropy of each row of $\phi'$ in the learned coarsened graph. This condition ensures that nodes sharing the same label are consistently mapped to the same supernode, thereby enhancing the coherence and consistency of the mapping process.

The optimization problem in Eq (9) is jointly non-convex. However, the problem becomes convex with respect to each variable when optimized separately, with all remaining variables as constant. To address this, we design an efficient block majorization–minimization (MM) procedure for the resulting multi-block non-convex program. Solving the majorized function at each iteration using a block successive upper-bound minimization (BSUM) which updates one variable at a time keeping the other fixed.

## 3.2 Update of $C$

With $\tilde{X}$ held constant, the optimization problem reduces to the following subproblem in $C$:

$$\underset{C \in \mathcal{S}_c}{\text{minimize}} f(C) = r_E(C,Y) - \gamma \log \det(C^\top \Theta C + J) + \frac{\lambda}{2}\|C^\top\|_{1,2}^2 + \frac{\alpha}{2}\|C\tilde{X} - X\|_F^2 + \text{tr}(\tilde{X}^\top C^\top \Theta C\tilde{X}) + \frac{\beta}{2}\|C^\top \Theta C\|_F^2$$
(10)

Where, $r_E(C,Y) = -\sum_{i \in \mathcal{I}_\phi} \sum_{j=1}^{l} \left( \frac{\phi_{ij}}{\|[\phi^T]_i\|_1} \log \frac{\phi_{ij}}{\|[\phi^T]_i\|_1} \right)$.

Using a first-order Taylor expansion around $C^{(t)}$, we construct a majorizing surrogate for $f(C)$ as follows (Beck & Pan, 2018; Razaviyayn et al., 2012; Sun et al., 2017; Kumar et al., 2024):

$$g(C|C^{(t)}) = f(C^{(t)}) + (C - C^{(t)})\nabla f(C^{(t)}) + \frac{L}{2}\|C - C^{(t)}\|^2$$
(11)

Consider $f(C)$ as the differentiable term in the objective. Assume further that its gradient is $L-$Lipschitz continuous, with constant $L = \max\{L_1, L_2, L_3, L_4, L_5, L_6\}$, where $L_1, \ldots, L_6$ are the Lipschitz constants associated with the gradients of $-\gamma \log \det(C^T \Theta C + J)$, $\text{tr}(\tilde{X}^T C^T \Theta C\tilde{X})$, $\|C\tilde{X} - X\|_F^2$, $\|C^T\|_{1,2}^2$, $\frac{\beta}{2}\|C^T \Theta C\|_F^2$, $r_E(C,Y)$ respectively. The proof that $r_E(\cdot, Y)$ also satisfies this Lipschitz continuity property is deferred to Appendix A.15. Omitting terms that are independent of $C$, the majorized form of Eq (10) is given by

$$\underset{C \in \mathcal{S}_c}{\text{minimize}} \quad \frac{1}{2}C^\top C - C^\top A$$
(12)

where $A = C^{(t)} - \frac{1}{L}\nabla f(C^{(t)})$ and $\nabla f(C^{(t)}) = -2\gamma \Theta C^{(t)}(C^{(t)^\top} \Theta C^{(t)} + J)^{-1} + \alpha \left( C^{(t)}\tilde{X} - X \right)\tilde{X}^\top + 2\Theta C^{(t)}\tilde{X}\tilde{X}^\top + \lambda C^{(t)}\mathbf{1} + \beta \Theta CC^\top \Theta C + \nabla r_E(C,Y)$ Where, $\mathbf{1}$ represents an all-ones matrix of dimension $k \times k$. For each supernode $i \in \mathcal{I}_\phi$, the corresponding $i$-th column of the gradient term is computed as $[\nabla_C r_E(C,Y)]_{:i} = -Y \left[ \frac{1}{\|[\phi^\top]_i\|_1} \left( \log_2\left( \frac{\phi_{i:}}{\|[\phi^\top]_i\|_1} \right) - \frac{1}{\|[\phi^\top]_i\|_1} \sum_{t=1}^{\ell} \phi_{it} \log_2\left( \frac{\phi_{it}}{\|[\phi^\top]_i\|_1} \right) \right) \right]^\top$. For $i \notin \mathcal{I}_\phi$, the corresponding entropy-gradient contribution is set to zero: $[\nabla_C r_E(C,Y)]_{:i} = 0$.

**Lemma 1.** *Using the KKT optimality condition we obtain the optimal solution of Eq (12) as*

$$C^{(t+1)} = \left( C^{(t)} - \frac{1}{L}\nabla f\left( C^{(t)} \right) \right)^+$$
(13)

*where $(X_{ij})^+ = \max(\frac{X_{ij}}{\|[X^T]_i\|_2}, 0)$ and $[X^T]_i$ is the $i$-th row of the matrix $X$.*

*Proof.* The proof is deferred to the Appendix A.2 □

## 3.3 Update of $\tilde{X}$

Fixing $C$, we obtain the following subproblem in $\tilde{X}$:

$$\underset{\tilde{X}}{\text{minimize}} \ f(\tilde{X}) = \text{tr}(\tilde{X}^\top C^\top \Theta C\tilde{X}) + \frac{\alpha}{2}\|C\tilde{X} - X\|_F^2.$$
(14)

Since $C^\top \Theta C$ is positive semidefinite and $C^\top C$ is positive definite, the objective in Eq (14) is strongly convex. Hence, the optimal solution admits a closed-form expression. By equating the gradient to zero we get, $2C^\top \Theta C\tilde{X} + \alpha C^\top(C\tilde{X} - X) = 0$, which yields

$$\tilde{X}^{t+1} = \left( \frac{2}{\alpha}C^\top \Theta C + C^\top C \right)^{-1} C^\top X.$$
(15)

**Theorem 1.** *The sequence $\{C^{(t)}, \tilde{X}^{(t)}\}$ generated by Algorithm 1 converges to the set of Karush–Kuhn–Tucker (KKT) points of Problem Eq (9).*

*Proof.* The detailed proof is deferred to A.3. □

Algorithm 1 summarizes the implementation of our entropy-guided graph coarsening (ENGC) method. In ENGC, the worst-case per-iteration cost for learning the coarsening matrix $C \in \mathbb{R}^{p \times k}$ is $O(p^2 k)$, due to matrix multiplications in the gradient computation. We provide the detailed time-complexity derivation and empirical runtimes in Appendix A.23

---

**Algorithm 1:** ENGC Algorithm

---

**Input:** $\mathcal{G}(X, Y, \Theta), \alpha, \gamma, \lambda, \beta, \delta$

**1** $t \leftarrow 0$;

**2 while** *stopping criteria not met* **do**

**3** $\quad$ Update $C^{t+1}$ and $\tilde{X}^{t+1}$ as in Eq (13) and Eq (15) respectively.

**4** $\quad$ $t \leftarrow t + 1$;

**5 end**

**Output:** $C$, $\Theta_c$, and $\tilde{X}$

---

## 4 Structural Spectral and Learning Guarantees

In this section, we establish theoretical guarantees on the structural, spectral, and label-consistency preservation of the proposed entropy-guided coarsening framework.

### 4.1 Entropy as a Structural Regularizer for Label Homogeneity

Consider a supernode $S_i = \{v_u \in V : C_{ui} > 0\}$ the set of original nodes assigned to the $i$-th supernode. $p_i(j) = \frac{n_{ij}}{|S_i|}$, where $n_{ij}$ is the number of nodes in $S_i$ with label $j$. The probability that two randomly sampled nodes from $S_i$ have different labels is $M(p_i) := 1 - \sum_j p_i(j)^2$, known as the *Gini impurity* which is zero for label-homogeneous supernodes and increases with label heterogeneity. Since Shannon entropy $(H(p))$ dominates the Rényi entropy of order 2, we have $H(p) \geq -\log\left(\sum_j p_j^2\right)$ (Renyi, 1961; Cover & Thomas, 2006). Exponentiating both sides yields $\sum_j p_j^2 \geq e^{-H(p)}$ which implies

$$M(p) = 1 - \sum_j p_j^2 \leq 1 - e^{-H(p)} \tag{16}$$

Thus, reducing entropy also controls label impurity within each supernode, motivating entropy as a regularizer for label-homogeneous coarsening. The next theorem characterizes the structural behavior induced by the two regularizers.

**Theorem 2.** *For each $i$, define $N_i = \sum_j \phi_{ij}$ and the feasible set $S_i = \{\phi_i \in \mathbb{R}_+^\ell : \sum_{j=1}^\ell \phi_{ij} = N_i\}$. Consider,*
$$E_{\text{ent}}(\phi) = \sum_{i=1}^k H(\phi_i/N_i), \qquad E_{\text{fro}}(\phi) = \|\phi\|_F^2.$$

*(i) $E_{\text{ent}}$ is minimized when each row $\phi_i$ is one-hot, $\phi_i = N_i e_{j^\star}$, corresponding to perfectly label-homogeneous supernodes.*

*(ii) $E_{\text{fro}}$ is minimized when each row is uniform, $\phi_{ij} = \frac{N_i}{\ell}$ with minimum value $\sum_{i=1}^k N_i^2/\ell$.*

*Proof.* The proof is provided in Appendix A.4. $\qquad\qquad\qquad\qquad\qquad\qquad\qquad\qquad\qquad\qquad$ $\square$

**Remark 1.** Theorem 2 highlights the structural contrast between the two regularizers. Entropy minimization drives each row of $\phi$ toward a single nonzero entry, grouping nodes with the same label into the same supernode. For unlabeled nodes (rows of $Y$ equal to zero), the entropy term is neutral; their assignments are then determined by the Laplacian and feature terms, typically attaching them to nearby label-consistent supernodes.

In contrast, the Frobenius objective used in LAGC (Kumar et al., 2024) minimizes $\|\phi\|_F^2$, which favors uniform rows in $\phi$. This promotes balanced label proportions within supernodes and can therefore lead to mixed-label clusters.

### 4.2 Controlling Projection Distortion via Label Impurity

This subsection connects entropy-guided semi-supervised graph coarsening with structural distortion guarantees, including Dirichlet-energy, cut, and effective-resistance preservation. These bounds are general once the projection error is controlled. We use the Dirichlet energy induced by the Laplacian $\Theta$: $\|\mathbf{u}\|_\Theta^2 := \mathbf{u}^\top \Theta \mathbf{u}$,

and its matrix version, summing the Dirichlet energies of all columns $\|U\|_{\Theta,F}^2 := \mathrm{Tr}(U^\top \Theta U)$. Given $C$, any supernode-level label matrix $Y_c \in \mathbb{R}^{k \times \ell}$ lifts the original graph as $CY_c \in \mathbb{R}^{p \times \ell}$. We then project the label matrix $Y$ onto the coarse space $\mathrm{range}(C)$ in this norm:

$$Y_c^\star := \arg \min_{Y_c \in \mathbb{R}^{k \times \ell}} \|Y - CY_c\|_{\Theta,F}. \tag{17}$$

This finds the best supernode-level matrix $Y_c$ such that the lifted node-level label signal $CY_c$ matches the true label matrix $Y$ as closely as possible. The optimality condition yields $\Theta_c Y_c = C^\top \Theta Y$, and we select the minimum-$\ell_2$ norm solution provided in A.14 as $Y_c^\star = \Theta_c^\dagger C^\top \Theta Y$. We define the normalized projection error as:

$$\delta^\star := \max_{\mathcal{F} \in [\ell]} \frac{\| Y - CY_c^\star \|_{\Theta,\mathcal{F}}}{\| Y \|_{\Theta,\mathcal{F}}} \tag{18}$$

Thus, $\delta^\star$ quantifies how much label information is lost when we represent node labels using only $k$ supernodes. To understand why entropy regularization improves coarsening quality, we analyze the structure of the projection distortion $\delta^\star$.

**Lemma 2.** *From Eq (17), the Dirichlet residual of the projected labels satisfies* $\|Y - CY_c^\star\|_{\Theta,F}^2 \lesssim \sum_{i=1}^k |S_i|^2 \left(1 - e^{-H(p_i)}\right)$, *Consequently, the projection distortion obeys* $\delta^\star \lesssim \dfrac{\sqrt{\sum_{i=1}^k |S_i|^2 \, h_2^{-1}(H(p_i))}}{\|Y\|_{\Theta,F}}$,

*where $h_2^{-1}$ denotes the inverse binary entropy function.*

Consequently, minimizing the entropy term directly bounds the projection distortion through its control of label impurity within supernodes. Therefore, $\delta^*$ mainly depends on how mixed the labels are inside each supernode. Entropy regularization reduces this mixing and gives a direct bound, while Frobenius regularization can allow more mixing. This explains why entropy-guided coarsening can yield better performance.

*Proof.* The proof is deferred to Appendix A.7 □

## 4.3 Dirichlet-Energy Distortion

**Theorem 3.** *Recall that reconstruction error defined in Eq (3) $\Delta := \Theta - \widetilde{\Theta}$. Then, for the label matrix $Y$,*

$$\mathrm{Tr}(Y^\top \Delta Y) \leq \|\Theta\|_2 \|C\|_2^4 g(\delta^*) \|Y\|_F^2, \qquad g(\delta^*) = 2\delta^* + (\delta^*)^2.$$

*In particular, the Dirichlet energy is preserved up to an error that decreases as $\delta^\star$ decreases.*

*Proof.* The proof is deferred to the Appendix A.5 □

**Corollary 4** (Cut Preservation). *Suppose Theorem 3 holds. Then for any cut indicator $\mathbf{z} \in \{0,1\}^p$ and the associated set $S = \{v : z_v = 1\}$, the cut energy distortion satisfies the following equation,*

$$\left|\mathbf{z}^\top \Theta \mathbf{z} - \mathbf{z}^\top \widetilde{\Theta} \mathbf{z}\right| = \left|\mathbf{z}^\top \Delta \mathbf{z}\right| \leq \varepsilon_{\mathrm{dir}} \|\mathbf{z}\|_2^2 = \varepsilon_{\mathrm{dir}} |S|.$$

*This equivalently encodes that the cut cost is preserved up to the cut size $|S|$*

*Proof.* The proof is deferred to the Appendix A.12 □

## 4.4 Bounding Effective-Resistance Distortion

For nodes $u, v \in \{1, \ldots, p\}$, let $e_u, e_v \in \mathbb{R}^p$ be the standard basis vector of $\mathbb{R}^p$. The effective resistance on the original graph is

$$R_{uv} = (e_u - e_v)^\top \Theta^\dagger (e_u - e_v) \quad \text{and} \tag{19}$$

Effective resistance on the coarse graph is:

$$R_{uv}^{(c)} = (e_u - e_v)^\top \widetilde{\Theta}^\dagger (e_u - e_v), \tag{20}$$

where $^\dagger$ is the Moore–Penrose pseudoinverse (Klein & Randić, 1993; Doyle & Snell, 1984; Chung, 1997; Ben-Israel & Greville, 2003). Thus, to control $|R_{uv} - R_{uv}^{(c)}|$ for all pairs $(u, v)$, it suffices to control the difference between $\|\Theta^\dagger - \widetilde{\Theta}^\dagger\|_2$.

**Lemma 3.** *Let $\Theta$ be the Laplacian of a connected graph with spectral gap $\lambda_{\text{gap}} := \lambda_2(\Theta) > 0$. Assume further that $\text{null}(\widetilde{\Theta}) = \text{span}\{\mathbf{1}_p\}$, and define the residual $\Delta_{err} := \|\Theta - \widetilde{\Theta}\|_2$. If $\Delta_{\text{err}} < \lambda_{\text{gap}}$, then*

$$\|\Theta^\dagger - \widetilde{\Theta}^\dagger\|_2 \ \leq \ \frac{\Delta_{err}}{\lambda_{\text{gap}}\left(\lambda_{\text{gap}} - \Delta_{err}\right)}, \tag{21}$$

*In particular, if $\Delta_{err} \leq \frac{1}{2}\lambda_{\text{gap}}$, then*

$$\|\Theta^\dagger - \widetilde{\Theta}^\dagger\|_2 \ \leq \ \frac{2\,\Delta_{err}}{\lambda_{\text{gap}}^2}. \tag{22}$$

Since $\lambda_{\text{gap}}$ is a fixed property of the original graph, lowering $\Delta_{err}$ directly tightens the resistance-preservation guarantee. We provide formal guarantees on effective resistance preservation under graph coarsening; the detailed analysis and proofs are given in Appendix A.11.

We adopt an $\epsilon$-similarity notion that measures how well the coarsened graph preserves feature smoothness under the graph Laplacian. The precise definition is given in Appendix A.13.

### 4.5 Entropy as an Information-Theoretic Objective

We formalize the probabilistic framework induced by coarsening and relate entropy of the node-profile matrix to information-theoretic quantities and downstream task performance. In particular, lower entropy in node profile matrix implies nodes with same label have merged in the supernode, which in turn reduces the error incurred by assigning each supernode its majority label coming from mixed labeled node. The following theorem makes this connection precise by showing that the entropy of each supernode distribution controls its majority-vote misclassification rate, and consequently bounds the total size-weighted classification error across the coarsened graph.

**Theorem 5.** *Let $p_i = (p_i(1), \ldots, p_i(\ell))$ be the class-distribution within supernode $i$, let $m_i := \max_j p_i(j)$ be the majority mass and $e_i := 1 - m_i$ the majority-vote error fraction in that supernode. Let the entropy be $H(p_i) := -\sum_{j=1}^\ell p_i(j)\log p_i(j)$ and define the binary entropy $h_2(x) := -x\log x - (1-x)\log(1-x)$. Then, for the general multiclass case, $e_i \leq 1 - 2^{-H(p_i)}$. In the binary case, this specializes to the entropy-inverse relation $e_i \leq h_2^{-1}(H(p_i))$, where $h_2^{-1}$ denotes the inverse of the binary entropy function on $[0, 1/2]$. Consequently, with $N_i$ the size of supernode $i$ and $q := \sum_{i=1}^k N_i e_i$ the size-weighted majority error, we have $q \leq \sum_{i=1}^k N_i h_2^{-1}(H(p_i))$.*

*Proof.* The proof is deferred to the Appendix A.9 □

### 4.6 Evaluating Label Mixing:

In the coarsened graph, each supernode should contain mostly nodes from the same class. To evaluate the label mixing in supernodes we use three label-aware metrics. **SII** quantifies the degree of label mixing within supernodes. **AMM** is the average error if we label each supernode by its majority class. **EIEC** converts the entropy value into a data-driven upper bound on that error and should be at least as large as AMM. Together, these metrics reveal mixing (SII), quantify its cost (AMM), and check consistency with theory (EIEC). We have added theory-to-empirics validation experiment in the Appendix A.19 to explicitly connect the theoretical quantities with downstream performance.

**Definition 4.1** (SII — Supernode Impurity Index). *Let $p_i \in \Delta^{\ell-1}$ be the label distribution of supernode $i$, Over the $k$ supernodes with $N_i > 0$, the* Supernode Impurity Index (SII) *is $\frac{1}{k}\sum_{i=1}^k H(p_i)$.*

**Definition 4.2** (AMM — Average Majority Miss). *For supernode $i$, define the majority-vote error $e_i := 1 - \max_j p_i(j)$ Over the $k$ supernodes with $N_i > 0$, the* Average Majority Miss (AMM) *is $\text{AMM} = \frac{1}{k}\sum_{i=1}^k e_i$*

**Definition 4.3** (EIEC — Entropy-Induced Error Ceiling). *Let $H_2(x) = -x\log x - (1-x)\log(1-x)$ be the binary entropy with inverse $H_2^{-1}$ on $[0, \frac{1}{2}]$. For supernode $i$, $b_i := H_2^{-1}(H(p_i)) \in [0, \frac{1}{2}]$,* $\quad\bar{b} := \frac{1}{k}\sum_{i=1}^k b_i$ . Theorem 5 implies $\text{AMM} \leq \text{EIEC}$.

## 5 Experiments

This section presents experiments validating our proposed ENGC algorithm, beginning with experimental settings, followed by a comparative analysis against key baselines, and concluding with a concise demonstration of ENGC's advantages.

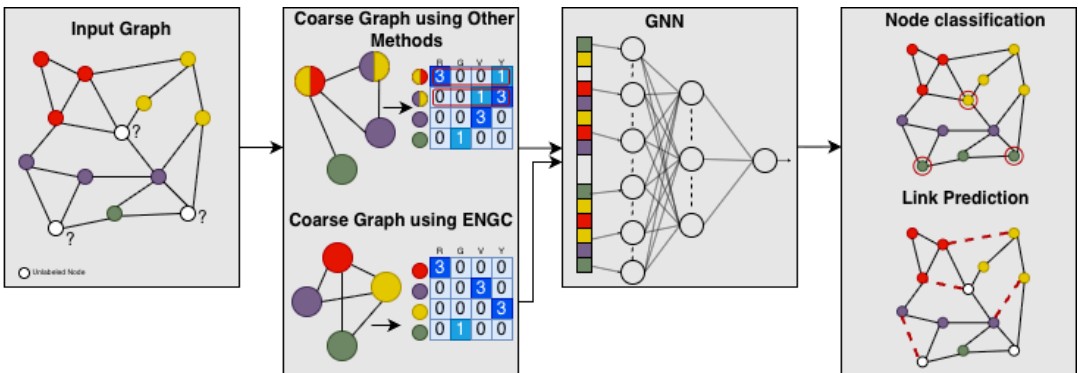

Figure 2: Schematic of the proposed ENGC framework and its use in downstream tasks. Starting from the original partially labeled graph $\mathcal{G}(\Theta, X, Y)$ , ENGC learns a node-to-supernode mapping $C$ and constructs a coarsened graph $\mathcal{G}_c(\Theta_c, \tilde{X}, Y_c)$ where, $Y_c$ is obtained by aggregating the available labels through the learned mapping. A GNN is then trained on the coarsened graph and used for downstream prediction, with performance evaluated on the original graph for tasks such as node classification and link prediction.

## 5.1 Experimental Setup

We evaluate ENGC on standard benchmark datasets to answer the following questions: (i) Does entropy-guided coarsening improve the semantic quality of supernodes? (ii) Does it translate into stronger downstream performance? (iii) What is the computational cost relative to existing baselines?

**Datasets.** We conduct experiments on widely used citation and co-authorship benchmarks given in 4. These datasets vary in graph size, feature dimension, and number of classes, allowing us to test the robustness of the proposed method under different structural regimes. We compare ENGC against representative graph reduction baselines, including GCOND (Jin et al., 2021), SCAL (Huang et al., 2021), FGC (Kumar et al., 2023b) and LAGC (Kumar et al., 2024). We further provide expanded empirical comparisons with baselines, including MGC Halder et al. (2025), BONSAI Gupta et al. (2025), SINGC Cohen & Talmon (2026), GCPA Li et al. (2025) and FedGM Zhang et al. (2025), in Appendix A.22, and report large-scale graph comparisons to showcase the scalability of the method on Flickr, OGBN-Arxiv, and OGBN-Products in Appendix A.21

These methods cover different methodological families, ranging from condensation-based approaches to optimization-based coarsening, and therefore provide a meaningful benchmark for evaluating both predictive performance and runtime. In the semi-supervised node classification setting commonly adopted in Graph Neural Networks (GNNs) Kumar et al. (2024); Dickens et al. (2024), only the training nodes possess label information and contribute directly to the supervised loss, while validation, test, and unlabeled nodes are assigned zero label vectors. Nevertheless, these unlabeled nodes remain part of the graph and actively participate in message passing and feature propagation during training. Consistent with this paradigm, when a supernode in the coarsened graph receives no labeled mass, its normalized node-profile row is treated as unlabeled and therefore excluded from the entropy regularization term and the supervised GNN loss computation. However, the supernode itself is retained in the coarsened graph and continues to participate in representation learning through feature aggregation and propagation via the coarsened Laplacian. In all experiments, ENGC is employed purely as a training-time graph reduction mechanism: the downstream GNN is trained on the coarsened graph while evaluation is performed on the original validation and test nodes, following the standard train-on-coarsened/evaluate-on-original protocol. We report downstream node-classification performance together with runtime. In addition, because our method is explicitly designed to reduce label mixing, we also measure the semantic quality of the learned super nodes using the impurity-based metric introduced in above section.

## 5.2 Node Classification

We follow the standard setup on citation datasets and use the public Planetoid train/validation/test splits. While running ENGC (i.e., during coarsening), we give the method the graph structure and node features,

| Dataset | r=k/p | Baseline | | | Proposed | | Whole dataset |
|---------|-------|----------|------|-----|------|------|---------------|
| | | **GCOND** | **SCAL** | **FGC** | **LAGC** | **ENGC** | |
| CORA | 0.3 | $81.56 \pm 0.62$ | $79.42 \pm 1.71$ | $84.03 \pm 0.08$ | $87.62 \pm 0.01$ | $87.05 \pm 0.05$ | |
| | 0.1 | $81.37 \pm 0.40$ | $77.80 \pm 0.51$ | $79.96 \pm 0.18$ | $86.10 \pm 0.03$ | $86.00 \pm 1.05$ | $89.5 \pm 1.23$ |
| | 0.05 | $78.93 \pm 0.45$ | $64.95 \pm 2.67$ | $77.31 \pm 0.65$ | $82.85 \pm 0.02$ | $86.74 \pm 0.12$ | |
| CITESEER | 0.3 | $72.43 \pm 0.49$ | $68.87 \pm 1.37$ | $72.85 \pm 0.10$ | $78.51 \pm 1.25$ | $79.69 \pm 0.37$ | |
| | 0.1 | $70.46 \pm 0.49$ | $71.38 \pm 3.62$ | $69.46 \pm 0.22$ | $76.00 \pm 0.50$ | $78.65 \pm 1.95$ | $78.09 \pm 1.96$ |
| | 0.05 | $64.03 \pm 2.41$ | $55.32 \pm 7.03$ | $69.02 \pm 0.24$ | $75.70 \pm 0.31$ | $77.30 \pm 0.01$ | |
| CO-PHY | 0.05 | $93.05 \pm 0.26$ | $73.09 \pm 7.41$ | $93.31 \pm 0.11$ | $94.46 \pm 0.58$ | $93.67 \pm 0.06$ | |
| | 0.03 | $92.81 \pm 0.31$ | $63.65 \pm 9.65$ | $92.00 \pm 1.78$ | $94.28 \pm 0.21$ | $94.04 \pm 0.23$ | $96.22 \pm 0.74$ |
| | 0.01 | $90.46 \pm 0.58$ | $61.07 \pm 5.68$ | $91.08 \pm 0.78$ | $93.26 \pm 0.89$ | $93.80 \pm 0.05$ | |
| PUBMED | 0.05 | $78.16 \pm 0.30$ | $72.82 \pm 2.62$ | $78.14 \pm 0.29$ | $82.85 \pm 0.32$ | $83.74 \pm 0.02$ | |
| | 0.03 | $78.04 \pm 0.47$ | $70.24 \pm 2.63$ | $77.60 \pm 0.16$ | $82.10 \pm 0.21$ | $82.73 \pm 0.36$ | $88.89 \pm 0.59$ |
| | 0.01 | $77.20 \pm 0.02$ | $50.49 \pm 10.5$ | $76.10 \pm 1.91$ | $81.27 \pm 0.91$ | $80.8 \pm 0.02$ | |
| CO-CS | 0.05 | $86.29 \pm 0.63$ | $34.45 \pm 10.07$ | $89.12 \pm 0.08$ | $91.36 \pm 0.48$ | $90.93 \pm 0.07$ | |
| | 0.03 | $86.32 \pm 0.45$ | $26.06 \pm 9.29$ | $86.32 \pm 0.43$ | $90.32 \pm 0.97$ | $89.39 \pm 0.49$ | $93.32 \pm 0.62$ |
| | 0.01 | $84.01 \pm 0.02$ | $14.42 \pm 8.51$ | $85.41 \pm 0.24$ | $88.27 \pm 0.34$ | $89.21 \pm 0.04$ | |
| DBLP | 0.05 | $79.15 \pm 0.20$ | $76.52 \pm 2.88$ | $80.08 \pm 0.01$ | $81.64 \pm 0.42$ | $81.34 \pm 0.28$ | |
| | 0.03 | $78.42 \pm 1.26$ | $75.49 \pm 2.84$ | $79.92 \pm 0.48$ | $80.93 \pm 0.12$ | $80.98 \pm 0.66$ | $85.35 \pm 0.80$ |
| | 0.01 | $74.29 \pm 0.57$ | $72.01 \pm 1.83$ | $77.47 \pm 0.33$ | $79.49 \pm 0.53$ | $80.18 \pm 0.70$ | |

Table 1: The table presents the node classification accuracy on real benchmark datasets for the proposed ENGC algorithms. Comparing the results with GCOND, SCAL, FGC and LAGC it is visible that proposed ENGC algorithm is competitive with state-of-the art methods.

but only the training labels. Concretely, we keep labels as one–hot for training nodes and put zeros for validation/test nodes, and then run ENGC on $\mathcal{G}(\Theta, X, Y_{\text{train}})$.

Once we learn the mapping $C$ and the coarsened graph $\mathcal{G}_c = (\Theta_c, \tilde{X})$, we assign a label to each supernode using only the training labels (i.e., by majority vote inside each supernode, e.g., using $C^\top Y_{\text{train}}$). After that, we train the downstream GNN on $\mathcal{G}_c$ using only those supernodes that have at least one training node. We tune all hyperparameters on the validation split. In addition, we evaluate node classification on the coarsened graph and compare it with node classification on the original graph. The runtime complexity of this algorithm is discussed in Appendix A.23.

---

**Algorithm 2:** Node Classification using ENGC

**Input:** Graph $\mathcal{G}(\Theta, X, Y)$; node splits $V_{\text{train}}, V_{\text{val}}, V_{\text{test}}$

**Output:** Predicted test labels $\widehat{Y}_{\text{test}}$

1 Set $Y_{\text{train}}$ by keeping labels only on $V_{\text{train}}$ and setting all other rows to zero;

2 Apply Algorithm 1 on $\mathcal{G}(\Theta, X, Y_{\text{train}})$ to obtain $C, \Theta_c, \widetilde{X}$;

3 **for** $i = 1, \ldots, k$ **do**

4 $\quad m_i \leftarrow \sum_{j=1}^{\ell} (C^\top Y_{\text{train}})_{ij}$;

5 $\quad$ **if** $m_i > 0$ **then**

6 $\quad\quad y_i^c \leftarrow \arg\max_j (C^\top Y_{\text{train}})_{ij}$;

7 $\quad$ **end**

8 **end**

9 Train GNN on $\mathcal{G}_c = (\Theta_c, \widetilde{X})$ using only supernodes with $m_i > 0$: $W^\star \leftarrow \arg\min_W \sum_{i:m_i>0} \ell\left(GNN_{\mathcal{G}_c}(W)_i, y_i^c\right)$;

10 Predict on the original graph: $\widehat{Y}_{\text{test}} \leftarrow \arg\max_j GNN_{\mathcal{G}(\Theta, X)}(W^\star)_{V_{\text{test}}, j}$;

11 **return** $\widehat{Y}_{\text{test}}$;

---

### 5.3 Link prediction

We further evaluate ENGC on the *link prediction* task and compare against strong coarsening baselines, including LAGC (Kumar et al., 2024) and FGC (Kumar et al., 2023b). Experiments are conducted on three standard citation networks: Cora, Citeseer, and PubMed. Following the SEAL framework (Zhang & Chen, 2018), the task is to predict whether an edge exists between a pair of nodes.

| Data set | $r = k/p$ | LAGC | ENGC | FGC | Whole Data |
|----------|-----------|------|------|-----|------------|
| Cora | 0.3 | $0.76 \pm 0.17$ | $0.79 \pm 0.09$ | $0.77 \pm 0.03$ | |
| | 0.1 | $0.77 \pm 0.00$ | $0.78 \pm 0.06$ | $0.75 \pm 0.03$ | $0.83 \pm 0.01$ |
| | 0.05 | $0.75 \pm 0.07$ | $0.76 \pm 0.07$ | $0.72 \pm 0.05$ | |
| CITESEER | 0.3 | $0.73 \pm 0.01$ | $0.74 \pm 0.07$ | $0.71 \pm 0.04$ | |
| | 0.1 | $0.72 \pm 0.08$ | $0.73 \pm 0.04$ | $0.69 \pm 0.00$ | $0.77 \pm 0.06$ |
| | 0.05 | $0.72 \pm 0.01$ | $0.71 \pm 0.02$ | $0.69 \pm 0.00$ | |
| PubMed | 0.05 | $0.76 \pm 0.00$ | $0.78 \pm 0.08$ | $0.69 \pm 0.01$ | |
| | 0.03 | $0.72 \pm 0.03$ | $0.76 \pm 0.09$ | $0.70 \pm 0.07$ | $0.81 \pm 0.03$ |
| | 0.01 | $0.67 \pm 0.02$ | $0.68 \pm 0.02$ | $0.64 \pm 0.03$ | |

Table 2: This table presents the Area Under the ROC Curve (AUC) for the link prediction task using ENGC. We compare our result with FGC (Kumar et al., 2023a) and LAGC (Kumar et al., 2024). The Whole Data column reports link prediction performance when using the entire dataset. Values are reported as mean $\pm$ standard deviation over multiple random splits.

For each dataset, we randomly split the observed edges into disjoint train/validation/test sets with an 80/10/10 ratio and average results over multiple random splits. We ensure the training graph remains connected; if a split breaks connectivity, we move only the minimum required edges back into the training set. After learning $C$ on the training graph, we train the GNN on coarse positive pairs defined by coarse edges and matched coarse negative pairs sampled from absent coarse edges. The learned model is then used for inference on the original validation/test graphs, with evaluation performed on the original positive and negative edge pairs. ENGC is run only on the training graph to learn the coarsened representation. We then train the SEAL link predictor on the coarsened graph using the training positives and corresponding negatives, tune hyperparameters on the validation split, and finally report test performance using ROC-AUC . We also report a full-graph baseline (i.e., training SEAL on the original graph without coarsening) for reference. Note that GCOND (Jin et al., 2021) is designed for node classification and is therefore not included in the link prediction comparisons. The detailed algorithm is given in Algorithm 3

### 5.4 Generalizability of the proposed ENGC Algorithm

To examine the generalizability of our coarsened graph learning method, we conduct node classification experiments using a range of Graph Neural Network (GNN) architectures. Specifically, we employ GCN (Kipf & Welling, 2016), APPNP (Gasteiger et al., 2018), and GAT (Veličković et al., 2017).Table5 in the appendix illustrates that our methods for learning the coarsened graph are compatible with different widely used GNN architectures, yielding nearly identical node classification accuracy obtained on different GNN structures.

### 5.5 Ablation Study

We quantify the contribution of each component appearing in the $C$–update of ENGC by ablating one term at a time from the gradient $\nabla f(C)$ of the subproblem in Eq (10), keeping all other settings fixed. Specifically, starting from the full objective in Eq (9) and the majorized update in Eq (12), we remove a single term $T_i$ from $\nabla f(C)$ and retrain. We report node-classification accuracy (mean $\pm$ s.d.). The detailed discussion of the result is given in Appendix A.20. The gradient used in the $C$–update decomposes as:

$$\nabla f(C) = \underbrace{-2\gamma \Theta C (C^\top \Theta C + J)^{-1}}_{T_1} + \underbrace{\alpha (C\tilde{X} - X)\tilde{X}^\top}_{T_2} + \underbrace{2\Theta C \tilde{X}\tilde{X}^\top}_{T_3} + \underbrace{\lambda C \mathbf{1}}_{T_4} + \underbrace{\beta \Theta CC^\top \Theta C}_{T_5} + \underbrace{\nabla r_E(C, Y)}_{T_6}.$$

| Variant | All | $-T_1$ | $-T_2$ | $-T_3$ | $-T_4$ | $-T_5$ | $-T_6$ |
|---------|-----|--------|--------|--------|--------|--------|--------|
| CORA 0.1 | $86.00 \pm 1.05$ | $77.41 \pm 0.16$ | $79.61 \pm 0.41$ | $84.84 \pm 0.66$ | $82.42 \pm 0.01$ | $82.53 \pm 0.41$ | $78.80 \pm 0.15$ |

Table 3: ENGC ablation on CORA($r = 0.1$). Removing $T_i$ denotes dropping that term from the objective.

# 6 Conclusion

In this work, we introduced Entropy-Guided Semi-Supervised Graph Coarsening (ENGC), a principled framework for attributed graph coarsening that integrates graph topology, node features, and partial supervision through an entropy-based regularization of the node-profile matrix. By explicitly reducing label mixing within supernodes, ENGC yields coarse graphs that are both structurally meaningful and semantically more coherent. We developed an efficient BSUM-based solver, established theoretical guarantees relating entropy minimization to preservation and learning properties, and validated the approach on benchmark datasets for node classification and link prediction. Overall, the proposed framework shows that entropy regularization is a promising and theoretically grounded mechanism for improving semi-supervised graph coarsening.

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

# A    Appendix

## A.1    Notations

We use the following notation throughout. Scalars and vectors are denoted by lower-case and bold lower-case symbols, respectively, while matrices are represented by upper-case letters. Matrix dimensions are omitted whenever they are unambiguous from context. For a matrix $X$, its $(i, j)$-th entry is written as $X_{ij}$, and $X^\dagger$ and $X^\top$ denote its pseudoinverse and transpose, respectively. We use $X_i$ and $[X^\top]_j$ to denote the $i$-th column and $j$-th row of $X$. The zero and one vectors or matrices, with dimensions inferred from context, are denoted by $\mathbf{0}$ and $\mathbf{1}$, respectively. The quantities $\|X\|_1$, $\|X\|_F$, and $\|X\|_{1,2}$ denote the $\ell_1$ norm, Frobenius norm, and $\ell_{1,2}$ norm of $X$, respectively, while $\|X\|_2$ denotes the Euclidean norm of a vector. For a positive semidefinite matrix $X$, $\det(X)$ refers to its generalized determinant, defined as the product of its nonzero eigenvalues. The matrix inner product is defined by $\langle X, Y \rangle = \operatorname{tr}(X^\top Y)$, where $\operatorname{tr}(\cdot)$ denotes the trace. We write $\mathbb{R}_+$ for the set of positive real numbers. For vectors $X_i$ and $X_j$, corresponding to the $i$-th and $j$-th columns of $X$, their inner product is given by $\langle X_i, X_j \rangle = X_i^\top X_j$.

### A.2 Proof of Lemma 1

*Proof.* The Lagrangian function of Eq (13) is

$$L(C, \tilde{X}, \boldsymbol{\mu}_1) = \frac{1}{2} C^\top C - C^\top A - \boldsymbol{\mu}_1^\top C \tag{23}$$

where $\boldsymbol{\mu}_1$ is the dual variable. The KKT conditions of Eq (13) are

$$C - A - \boldsymbol{\mu}_1 = 0, \tag{24}$$
$$\boldsymbol{\mu}^\top C = 0, \tag{25}$$
$$C \geq 0, \tag{26}$$
$$\boldsymbol{\mu}_1 \geq 0 \tag{27}$$

Therefore, the solution $C$ that satisfying the KKT conditions in (24-27) is $C^{t+1} = \left(C^{(t)} - \frac{1}{L}\nabla f\left(C^{(t)}\right)\right)^+$. This concludes the proof. □

### A.3 Proof of Theorem 1

*Proof of Theorem 1.* We show that each limit point $(C^t, \widetilde{X}^t)$ satisfies the KKT conditions of the relaxed problem in Eq (9) over $\mathcal{S}_c$. Let $(C^\infty, \tilde{X}^\infty)$ be a limit point of the generated sequence. The Lagrangian function of Eq (9) is

$$L(C, \tilde{X}, \boldsymbol{\mu}_1, \boldsymbol{\mu}_2) = -\gamma \log \det(C^T \Theta C + J) + \mathrm{tr}(\tilde{X}^T C^T \Theta C \tilde{X}) + \frac{\alpha}{2}\|X - C\tilde{X}\|_F^2 + r(C, Y)$$

$$+ \frac{\beta}{2}\|C^\top \Theta C\|_F^2 + \frac{\lambda}{2}\sum_{i=1}^p \|[C^T]_i\|_1^2 - \boldsymbol{\mu}_1^\top C + \boldsymbol{\mu}_2^T \left[\|C_1^T\|_2^2 - 1 \quad \|C_2^T\|_2^2 - 1 \ldots \|C_p^T\|_2^2 - 1\right]^T \tag{28}$$

where $\boldsymbol{\mu}_1$ and $\boldsymbol{\mu}_2$ are the dual variables.
(1) The KKT condition with respect to $C$ is

$$-2\gamma \Theta C Z^{-1} + \alpha\left(C\tilde{X} - X\right)\tilde{X}^\intercal + 2\Theta C \tilde{X}\tilde{X}^\top + \lambda C \mathbf{1}_{k \times k} - \boldsymbol{\mu}_1$$

$$+ 2\left[\mu_{21}C_1^\top, \ldots, \mu_{2p}C_p^\top\right]^\top + \beta \Theta C C^\top \Theta C + \nabla r_E(C, Y) = 0, \tag{29}$$

$$\boldsymbol{\mu}_2^\top \begin{bmatrix} \|C_1^\top\|_2^2 - 1 \\ \|C_2^\top\|_2^2 - 1 \\ \vdots \\ \|C_p^\top\|_2^2 - 1 \end{bmatrix} = 0 \tag{30}$$

$$\boldsymbol{\mu}_1^\top C = 0 \tag{31}$$

$$\boldsymbol{\mu}_1 \geq 0, \qquad C \geq 0, \qquad \boldsymbol{\mu}_2 \geq 0, \qquad \|[C^\top]_i\|_2^2 \leq 1 \tag{32}$$

where $\mathbf{1}_{k \times k}$ is a $k \times k$ matrix whose all entry are one and $Z = C^T \Theta C + J$. $C$ is derived by using KKT condition from Eq (13):

$$C^\infty - C^\infty + \frac{1}{L}\Big(-2\gamma \Theta C^\infty((C^\infty)^\top \Theta C^\infty + J)^{-1} + \alpha(C^\infty \tilde{X}^\infty - X)(\tilde{X}^\infty)^\top$$

$$+ 2\Theta C^\infty \tilde{X}^\infty(\tilde{X}^\infty)^\top + \lambda C^\infty \mathbf{1}_{k \times k} + \beta \Theta C^\infty (C^\infty)^\top \Theta C^\infty + \nabla r_E(C^\infty, Y)\Big) = 0. \tag{33}$$

For $\boldsymbol{\mu}_1 = 0$ and $\mu_{2i}[C^T]_i^\infty = 0 \ \forall \ i = 1, 2, \ldots p$, we observe that $C^\infty$ satisfies the KKT condition.
(2) The KKT condition with respect to $\tilde{X}$ is

$$2C^T \Theta C \tilde{X} + \alpha C^T (C\tilde{X} - X) = 0$$

This concludes the proof. □

### A.4 Proof of Theorem 2

*Proof of Theorem 2.* We fix a row $i$ and write $p := \phi_i / N_i$. Then $p_j \geq 0$ and $\sum_j p_j = 1$. For each $j$, $-p_j \log p_j \geq 0$, for all $p_j \in [0, 1]$. Hence $H(p) = \sum_j -p_j \log p_j \geq 0$, and $H(p) = 0$ iff $p$ is one–hot. Equivalently, the minimizers are $\phi_i = N_i e_{j^\star}$. This proves (1) after summing over $i$. For (2), we fix row $i$. Then by Cauchy–Schwarz,

$$\left( \sum_{j=1}^{\ell} \phi_{ij} \right)^2 \leq \ell \sum_{j=1}^{\ell} \phi_{ij}^2 \quad \Longrightarrow \quad \sum_{j=1}^{\ell} \phi_{ij}^2 \geq \frac{N_i^2}{\ell},$$

with equality iff all $\phi_{ij}$ are equal, i.e., $\phi_{ij} = N_i / \ell$ for all $j$. Summing over $i$ yields the stated global minimum for $E_{\text{fro}}$. □

### A.5 Proof of Theorem 3

*Proof of Theorem 3.* Consider the matrix projection problem

$$Y_c^\star := \arg \min_{Y_c \in \mathbb{R}^{k \times \ell}} \|Y - CY_c\|_{\Theta, F}, \qquad \|U\|_{\Theta, F}^2 := \text{Tr}(U^\top \Theta U), \qquad \|\mathbf{z}\|_\Theta^2 := \mathbf{z}^\top \Theta \mathbf{z}.$$

Let the residual matrix be

$$R := Y - CY_c^\star \in \mathbb{R}^{p \times \ell}.$$

Optimality for the squared objective

$$F(Y_c) := \|Y - CY_c\|_{\Theta, F}^2 = \text{Tr}\big( (Y - CY_c)^\top \Theta (Y - CY_c) \big)$$

yields the first-order condition

$$C^\top \Theta (Y - CY_c^\star) = C^\top \Theta R = 0, \tag{A.1}$$

Writing $Y = CY_c^\star + R$ and using Eq (A.1) gives the decomposition

$$\text{Tr}(Y^\top \Theta Y) = \text{Tr}\big( (CY_c^\star + R)^\top \Theta (CY_c^\star + R) \big) = \text{Tr}\big( (Y_c^\star)^\top (C^\top \Theta C) Y_c^\star \big) + \text{Tr}(R^\top \Theta R) = \text{Tr}\big( (Y_c^\star)^\top \Theta_c Y_c^\star \big) + \|R\|_{\Theta, F}^2, \tag{A.2}$$

where $\Theta_c := C^\top \Theta C$.

Now let $\widetilde{\Theta} = C\Theta_c C^\top$ and $\Delta := \Theta - \widetilde{\Theta}$. We have

$$\text{Tr}(Y^\top \widetilde{\Theta} Y) = \text{Tr}\big( Y^\top C \Theta_c C^\top Y \big) = \text{Tr}\big( (C^\top Y)^\top \Theta_c (C^\top Y) \big) = \|\Theta_c^{1/2} C^\top Y\|_F^2. \tag{A.3}$$

Substituting $Y = CY_c^\star + R$ gives $C^\top Y = C^\top C Y_c^\star + C^\top R$. Hence, by the triangle inequality,

$$\|\Theta_c^{1/2} C^\top Y\|_F \geq \|\Theta_c^{1/2} C^\top C Y_c^\star\|_F - \|\Theta_c^{1/2} C^\top R\|_F. \tag{A.4}$$

Squaring and using $(a - b)^2 \geq a^2 - 2ab - b^2$ yields

$$\|\Theta_c^{1/2} C^\top Y\|_F^2 \geq \|\Theta_c^{1/2} C^\top C Y_c^\star\|_F^2 - 2\|\Theta_c^{1/2} C^\top C Y_c^\star\|_F \|\Theta_c^{1/2} C^\top R\|_F - \|\Theta_c^{1/2} C^\top R\|_F^2. \tag{A.5}$$

Therefore,

$$\text{Tr}(Y^\top \Delta Y) = \text{Tr}(Y^\top \Theta Y) - \text{Tr}(Y^\top \widetilde{\Theta} Y) \leq 2\|\Theta_c^{1/2} C^\top C Y_c^\star\|_F \|\Theta_c^{1/2} C^\top R\|_F + \|\Theta_c^{1/2} C^\top R\|_F^2, \tag{A.6}$$

where we dropped the nonnegative term $\|\Theta_c^{1/2}C^\top C Y_c^\star\|_F^2$.

We now bound the two factors using spectral norms. First, since

$$\|\Theta_c\|_2 = \|C^\top\Theta C\|_2 \leq \|C\|_2^2\|\Theta\|_2, \quad \text{we have} \quad \|\Theta_c^{1/2}\|_2 \leq \|C\|_2\sqrt{\|\Theta\|_2}. \tag{A.7}$$

Hence

$$\|\Theta_c^{1/2}C^\top R\|_F \leq \|\Theta_c^{1/2}\|_2\,\|C^\top\|_2\,\|R\|_F \leq \|C\|_2^2\sqrt{\|\Theta\|_2}\,\|R\|_F \leq \|C\|_2^2\,\|R\|_{\Theta,F}, \tag{A.8}$$

where the last inequality uses $\|R\|_{\Theta,F}^2 = \mathrm{Tr}(R^\top\Theta R) \leq \|\Theta\|_2\|R\|_F^2$.

Second, by Eq (A.2),

$$\|\Theta_c^{1/2}Y_c^\star\|_F^2 = \mathrm{Tr}\big((Y_c^\star)^\top\Theta_c Y_c^\star\big) \leq \mathrm{Tr}(Y^\top\Theta Y) = \|Y\|_{\Theta,F}^2,$$

so $\|\Theta_c^{1/2}Y_c^\star\|_F \leq \|Y\|_{\Theta,F}$. Using this and $\|C^\top C\|_2 \leq \|C\|_2^2$ gives

$$\|\Theta_c^{1/2}C^\top C Y_c^\star\|_F \leq \|C^\top C\|_2\,\|\Theta_c^{1/2}Y_c^\star\|_F \leq \|C\|_2^2\,\|Y\|_{\Theta,F}. \tag{A.9}$$

Substituting Eq (A.8)–Eq (A.9) into Eq (A.6) yields

$$\mathrm{Tr}(Y^\top\Delta Y) \leq \|C\|_2^4\Big(2\|Y\|_{\Theta,F}\|R\|_{\Theta,F} + \|R\|_{\Theta,F}^2\Big). \tag{A.10}$$

Define the normalized projection error

$$\delta(Y) \;:=\; \frac{\|R\|_{\Theta,F}}{\|Y\|_{\Theta,F}} \;=\; \frac{\|Y - CY_c^\star\|_{\Theta,F}}{\|Y\|_{\Theta,F}}.$$

Then $\|R\|_{\Theta,F} = \delta(Y)\|Y\|_{\Theta,F}$ and

$$\mathrm{Tr}(Y^\top\Delta Y) \leq \|C\|_2^4\big(2\delta(Y) + \delta(Y)^2\big)\,\|Y\|_{\Theta,F}^2. \tag{A.11}$$

Finally, since $\|Y\|_{\Theta,F}^2 = \mathrm{Tr}(Y^\top\Theta Y) \leq \|\Theta\|_2\|Y\|_F^2$, we obtain

$$\mathrm{Tr}(Y^\top\Delta Y) \leq \|\Theta\|_2\,\|C\|_2^4\,g(\delta(Y))\,\|Y\|_F^2, \qquad g(t) = 2t + t^2.$$

This proves the stated bound with $\Delta \leq \|\Theta\|_2\|C\|_2^4 g(\delta(Y))$.

$\square$

## A.6 Proof of Lemma 3

*Proof of Lemma 3.* Because the graph is connected, $\Theta\mathbf{1} = 0$ and $\lambda_2(\Theta) = \lambda_{\mathrm{gap}} > 0$. It is standard that the Moore–Penrose pseudoinverse $\Theta^\dagger$ acts as the true inverse of $\Theta$ on all zero-sum vectors $x$ with $\mathbf{1}^\top x = 0$, and similarly $\widetilde{\Theta}^\dagger$ acts as the true inverse of $\widetilde{\Theta}$ on that same subspace. On this subspace both $\Theta$ and $\widetilde{\Theta}$ are symmetric positive definite.

Let $A := \Theta$, $B := \widetilde{\Theta}$, and $E := A - B = \Theta - \widetilde{\Theta}$, all viewed as operators on the zero-sum subspace. For invertible $A, B$ we have

$$B^{-1} - A^{-1} \;=\; B^{-1}(A - B)A^{-1} \;=\; B^{-1}EA^{-1}.$$

Taking operator norms and using submultiplicativity,

$$\|A^{-1} - B^{-1}\|_2 \;\leq\; \|B^{-1}\|_2\,\|E\|_2\,\|A^{-1}\|_2. \tag{34}$$

By definition, $\|E\|_2 \leq \|\Theta - \widetilde{\Theta}\|_2 = \Delta_{\mathrm{res}}$. Since the smallest nonzero eigenvalue of $\Theta$ is $\lambda_{\mathrm{gap}}$, the inverse of $\Theta$ on this subspace has norm $\|A^{-1}\|_2 = 1/\lambda_{\mathrm{gap}}$ (Horn & Johnson, 2013). Moreover, by Weyl's eigenvalue inequality for Hermitian matrices, the smallest nonzero eigenvalue of $\widetilde{\Theta}$ satisfies

$$\lambda_{\min}(\widetilde{\Theta}) \;\geq\; \lambda_{\mathrm{gap}} - \|\Theta - \widetilde{\Theta}\|_2 \;=\; \lambda_{\mathrm{gap}} - \Delta,$$

so

$$\|B^{-1}\|_2 = \frac{1}{\lambda_{\min}(\widetilde{\Theta})} \leq \frac{1}{\lambda_{\text{gap}} - \Delta}.$$

Substituting these bounds into Eq (34) gives

$$\|A^{-1} - B^{-1}\|_2 \leq \frac{1}{\lambda_{\text{gap}} - \Delta} \cdot \Delta \cdot \frac{1}{\lambda_{\text{gap}}} = \frac{\Delta}{\lambda_{\text{gap}}(\lambda_{\text{gap}} - \Delta)}.$$

Finally, $\Theta^\dagger$ and $\widetilde{\Theta}^\dagger$ agree with $A^{-1}$ and $B^{-1}$ on all zero-sum vectors and vanish on $\mathbf{1}$, so the same bound holds for $\|\Theta^\dagger - \widetilde{\Theta}^\dagger\|_2$, proving Eq (21).

If $\Delta \leq \frac{1}{2}\lambda_{\text{gap}}$, then $\lambda_{\text{gap}} - \Delta \geq \frac{1}{2}\lambda_{\text{gap}}$, which simplifies Eq (21) to Eq (22).

As an immediate consequence of Lemma 3, we obtain for any pair of distinct nodes $u, v$,

$$\begin{aligned}
\left| R_{uv} - R_{uv}^{(L)} \right| &= \left| (\mathbf{e}_u - \mathbf{e}_v)^\top (\Theta^\dagger - \widetilde{\Theta}^\dagger)(\mathbf{e}_u - \mathbf{e}_v) \right| \\
&\leq \|\Theta^\dagger - \widetilde{\Theta}^\dagger\|_2 \|\mathbf{e}_u - \mathbf{e}_v\|_2^2 \leq 4\|\Theta^\dagger - \widetilde{\Theta}^\dagger\|_2.
\end{aligned}$$

Therefore, if $\Delta \ll \lambda_{\text{gap}}$, all effective resistances are uniformly preserved.

$\square$

### A.7 Proof of Lemma 2

*Proof of Lemma2.* Recall that the Dirichlet residual is

$$\|Y - CY_c^\star\|_{\Theta,F}^2 = \text{Tr}\big((Y - CY_c^\star)^\top \Theta (Y - CY_c^\star)\big). \tag{35}$$

Using the Laplacian quadratic form identity,

$$\text{Tr}(A^\top \Theta A) = \sum_{(u,v)\in E} w_{uv} \|a_u - a_v\|_2^2, \tag{36}$$

and noting that $CY_c^\star$ is constant on each supernode $S_i$, we obtain for intra-supernode edges

$$\|y_u - y_v\|_2^2 = \begin{cases} 0, & y_u = y_v, \\ 2, & y_u \neq y_v. \end{cases} \tag{37}$$

Hence the Dirichlet residual contains the exact label-mixing term

$$\|Y - CY_c^\star\|_{\Theta,F}^2 = 2\sum_{i=1}^k \sum_{\substack{(u,v)\in E \\ u,v \in S_i}} w_{uv} \mathbf{1}[y_u \neq y_v] + (\text{nonnegative cross-supernode terms}). \tag{38}$$

Thus, projection distortion is governed by weighted intra-supernode label disagreement. The expected disagreement weight scales as

$$\mathbb{E}\left[ \sum_{\substack{(u,v)\in E \\ u,v \in S_i}} w_{uv} \mathbf{1}[y_u \neq y_v] \right] \propto |S_i|^2 M(p_i). \tag{39}$$

Consequently,

$$\mathbb{E}\big[\|Y - CY_c^\star\|_{\Theta,F}^2\big] \propto \sum_i |S_i|^2 M(p_i), \tag{40}$$

Using Eq (16) above, we get

$$\mathbb{E}\big[\|Y - CY_c^\star\|_{\Theta,F}^2\big] \lesssim \sum_{i=1}^{k} |S_i|^2 \left(1 - e^{-H(p_i)}\right),\tag{41}$$

is controlled by the impurity of label distributions within supernodes.

Moreover, using the entropy–majority from 5,

$$M(p_i) \le 2\, h_2^{-1}\big(H(p_i)\big),\tag{42}$$

we obtain the explicit control as below:

$$\delta^\star \lesssim \frac{\sqrt{\sum_i |S_i|^2\, h_2^{-1}(H(p_i))}}{\|Y\|_{\Theta,F}}.\tag{43}$$

$\square$

## A.8 Multiclass Entropy-to-Majority-Error Bound

Here, we provide the multiclass version of the entropy-to-majority-error relation used above. For the general multiclass case, we use a bound based on the relation between Shannon entropy.

Let $S_i$ denote the set of original nodes assigned to supernode $i$, and let $p_i = (p_i(1), \ldots, p_i(\ell))$ be the class distribution of this supernode. Define the majority-label mass as $m_i = \max_{j \in \{1,\ldots,\ell\}} p_i(j)$, and the corresponding majority-vote error as $e_i = 1 - m_i$.

**Lemma 4.** *For any multiclass distribution $p_i$ over $\ell$ classes, the majority-vote error satisfies*

$$e_i \le 1 - 2^{-H(p_i)},$$

*where*

$$H(p_i) = -\sum_{j=1}^{\ell} p_i(j) \log_2 p_i(j).$$

*Proof.* Let

$$m_i = \max_j p_i(j).$$

Since $m_i$ is the largest entry of $p_i$, we have

$$p_i(j) \le m_i, \qquad j = 1, \ldots, \ell.$$

Therefore,

$$-\log_2 p_i(j) \ge -\log_2 m_i.$$

Multiplying both sides by $p_i(j) \ge 0$ and summing over all classes gives

$$-\sum_{j=1}^{\ell} p_i(j) \log_2 p_i(j) \ge -\sum_{j=1}^{\ell} p_i(j) \log_2 m_i.$$

Since $\sum_{j=1}^{\ell} p_i(j) = 1$, this yields

$$H(p_i) \ge -\log_2 m_i.$$

Equivalently,

$$m_i \ge 2^{-H(p_i)}.$$

Using $e_i = 1 - m_i$, we obtain

$$e_i = 1 - m_i \le 1 - 2^{-H(p_i)}.$$

This proves the result. $\square$

Consequently, the size-weighted majority-vote error over all labeled-mass supernodes satisfies:

$$\sum_{i \in \mathcal{I}_\phi} |S_i| e_i \leq \sum_{i \in \mathcal{I}_\phi} |S_i| \left(1 - 2^{-H(p_i)}\right).$$

Where $S_i$ denotes the supernode as described in 4.1. Thus, reducing the entropy of the node-profile distribution reduces an explicit upper bound on the majority-vote error inside each supernode. In the binary case, when $\ell = 2$ and $e_i \leq 1/2$, the distribution can be written as $(1 - e_i, e_i)$, and therefore

$$H(p_i) = h_2(e_i), \qquad e_i = h_2^{-1}(H(p_i)).$$

Hence, the binary inverse expression used in the main text should be interpreted as the binary-label special case, while the multiclass-valid bound is

$$e_i \leq 1 - 2^{-H(p_i)}.$$

### A.9 Proof of Theorem 5

*Proof of Theorem5.* Fix a supernode $i$ and write $p := p_i$, $m := m_i$, $e := e_i = 1 - m$. Without loss of generality, relabel coordinates so that $p_1 = m = \max_j p_j$. Then $p_1 = 1 - e$, $\sum_{j=2}^\ell p_j = e$, and $p_j \geq 0$ for $j \geq 2$.

Consider the function $\varphi(x) := -x \log x$, which is concave on $[0, 1]$. For fixed $p_1 = 1 - e$, the entropy can be written as

$$H(p) = \varphi(1 - e) + \sum_{j=2}^\ell \varphi(p_j), \text{subject to } \sum_{j=2}^\ell p_j = e, \ p_j \geq 0. \tag{44}$$

Since $\sum_{j=2}^\ell \varphi(p_j)$ is a concave function over $\{(p_2, \ldots, p_\ell) \in \mathbb{R}_+^{\ell-1} : \sum_{j=2}^\ell p_j = e\}$, its minimum over this convex set is attained at an extreme point, i.e. when there is single element which is non zero. Hence

$$\min_{\substack{p_j \geq 0 \\ \sum_{j \geq 2} p_j = e}} H(p) = \varphi(1 - e) + \varphi(e) + \sum_{j=3}^\ell \varphi(0) = h_2(e) \tag{45}$$

Therefore, for every feasible $p$ with majority error $e$,

$$H(p) \geq h_2(e). \tag{46}$$

Next, observe that $h_2$ is strictly increasing on $[0, \frac{1}{2}]$: indeed,

$$h_2'(x) = \log \frac{1-x}{x} > 0 \quad \text{for } x \in (0, \tfrac{1}{2}). \tag{47}$$

Thus $h_2$ admits an inverse $h_2^{-1}$ on $[0, \frac{1}{2}]$, and from $H(p) \geq h_2(e)$ we obtain

$$e \leq h_2^{-1}(H(p)). \tag{48}$$

Restoring the index $i$ yields the stated pointwise bound $e_i \leq h_2^{-1}(H(p_i))$.

So we get,

$$q = \sum_{i=1}^k N_i e_i \leq \sum_{i=1}^k N_i h_2^{-1}(H(p_i)) \tag{49}$$

This completes the proof. $\qquad\square$

### A.10 Error Bounds: Entropy vs. Frobenius

Figure 3 reports three summary metrics. On CORA at $r = 0.1$ ($k = 270$), **entropy regularization** yields purer supernodes and fewer majority-vote mistakes: SII drops from 1.2818 to 0.3193 (a $\approx 75\%$ reduction), and AMM falls from 0.4002 to 0.1242 (also $\approx 75\%$ lower). Its EIEC is 0.4111, comfortably above its AMM (0.1242) as theory predicts, indicating a valid, non-tight ceiling. In contrast, the Frobenius baseline exhibits much higher impurity and error (SII 1.2818, AMM 0.4002); its reported EIEC (0.4407) sits slightly above AMM. Overall, entropy regularization directly encourages label-consistent, compact supernodes and delivers markedly lower error than Frobenius, making it the preferred choice for label-aware graph coarsening with theoretically consistent error control.

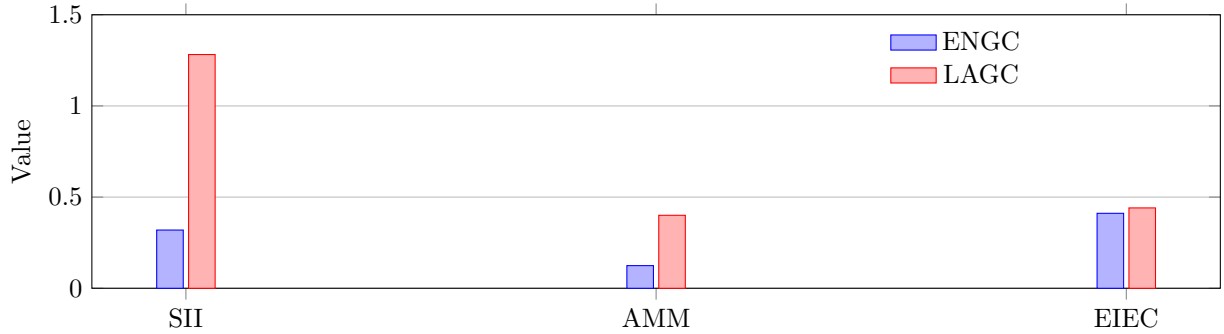

Figure 3: Cora ($r = 0.1, k = 270$): SII, AMM, EIEC for ENGC vs. LAGC.

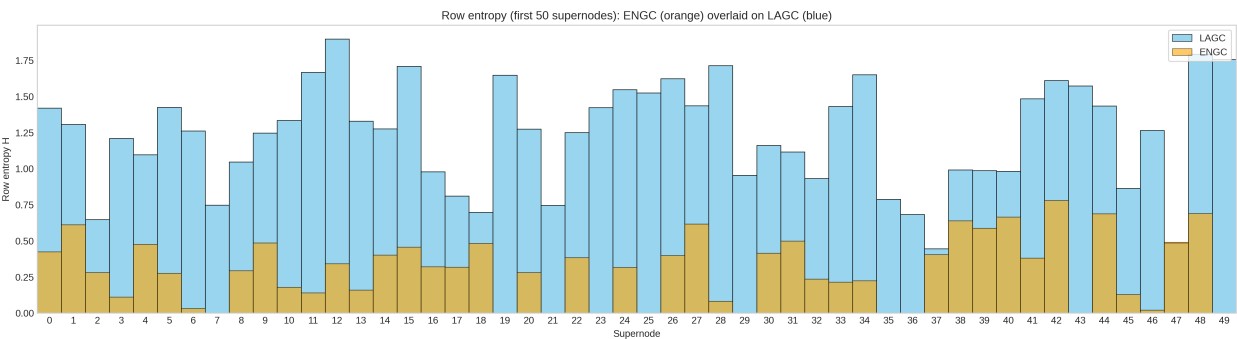

Figure 4: **Row-entropy per supernode (first 50).** Dataset: Cora; label rate $r = 0.1$; $k = 270$ supernodes. Bars are *overlaid* : LAGC/Frobenius in sky blue, ENGC/Entropy in orange. Height equals $H(p_i)$ Lower bars indicate purer label allocation within a supernode; ENGC generally yields lower or comparable $H(p_i)$ than LAGC on the supernodes shown.

### A.11 Resistance preservation.

Let $\mathcal{G}(\Theta, X, Y)$ be a connected graph with Laplacian $\Theta \in \mathbb{R}^{p \times p}$. Denote the all-ones vector by $\mathbf{1}$

For connected $\mathcal{G}$, with smallest eigenvalue $\lambda_{\text{gap}} := \lambda_2(\Theta) > 0$ Fiedler (1973); Chung (1997). Let $C \in \mathbb{R}_+^{p \times k}$ be the mapping matrix, $\Theta_c := C^\top \Theta C \in \mathbb{R}^{k \times k}$ the coarse Laplacian, and $\widetilde{\Theta} := C\Theta_c C^\top \in \mathbb{R}^{p \times p}$ its lifted. Define the energy distortion

$$\Delta_{err} := \big\| \Theta - \tilde{\Theta} \big\|_2. \tag{50}$$

Using $\|e_u - e_v\|_2^2 = 2$ and $|x^\top M x| \le \|M\|_2 \|x\|_2^2$,

$$|R_{uv} - R_{uv}^{(c)}| = |(e_u - e_v)^\top (\Theta^\dagger - \widetilde{\Theta}^\dagger)(e_u - e_v)| \le 2\|\Theta^\dagger - \widetilde{\Theta}^\dagger\|_2.$$

Combining with Eq (21) gives

$$|R_{uv} - R_{uv}^{(c)}| \ \leq \ \frac{2\,\Delta_{err}}{\lambda_{\text{gap}}\,(\lambda_{\text{gap}} - \Delta_{err})}, \tag{51}$$

and under $\Delta \leq \frac{1}{2}\lambda_{\text{gap}}$ the simplified bound

$$|R_{uv} - R_{uv}^{(c)}| \ \leq \ \frac{4\,\Delta_{err}}{\lambda_{\text{gap}}^2}. \tag{52}$$

### A.12    Cut-Preservation

*Proof of Corollary 4.* Let $\mathbf{z} \in \{0,1\}^p$ be a cut indicator and set $S := \{v : z_v = 1\}$. By definition $\Delta := \Theta - \widetilde{\Theta}$, so

$$\mathbf{z}^\top \Theta \mathbf{z} - \mathbf{z}^\top \widetilde{\Theta} \mathbf{z} = \mathbf{z}^\top (\Theta - \widetilde{\Theta})\mathbf{z} = \mathbf{z}^\top \Delta \mathbf{z}.$$

Applying Theorem 3 with $\mathbf{u} = \mathbf{z}$ gives

$$\left|\mathbf{z}^\top \Theta \mathbf{z} - \mathbf{z}^\top \widetilde{\Theta} \mathbf{z}\right| = \left|\mathbf{z}^\top \Delta \mathbf{z}\right| \leq \Delta \|\mathbf{z}\|_2^2.$$

Since $\mathbf{z} \in \{0,1\}^p$, we have $\|\mathbf{z}\|_2^2 = \sum_{v=1}^p z_v^2 = \sum_{v=1}^p z_v = |S|$, and therefore

$$\left|\mathbf{z}^\top \Theta \mathbf{z} - \mathbf{z}^\top \widetilde{\Theta} \mathbf{z}\right| \leq \Delta |S|.$$

$\square$

### A.13    $\epsilon$-similarity

The coarsened graph data $\mathcal{G}_c(\Theta_c, \tilde{X})$ is $\epsilon$ similar to the original graph data $\mathcal{G}(\Theta, X)$, i.e., there exist an $0 \leq \epsilon \leq 1$ such that

$$(1 - \epsilon)\|X\|_\Theta \leq \|\tilde{X}\|_{\Theta_c} \leq (1 + \epsilon)\|X\|_\Theta \tag{53}$$

where $\|X\|_\Theta^2 = \text{tr}(X^T \Theta X)$ and $\|\tilde{X}\|_{\Theta_c}^2 = \text{tr}(\tilde{X}^T \Theta_c \tilde{X})$.

### A.14    Derivation of the optimality condition and the minimum-$\ell_2$ solution

*Proof.* Fix a class $c \in [\ell]$ and consider the $c$-th column of the matrix projection problem

$$Y_c^\star := \arg\min_{Y_c \in \mathbb{R}^{k \times \ell}} \|Y - CY_c\|_{\Theta,F}, \qquad \|U\|_{\Theta,F}^2 := \text{Tr}(U^\top \Theta U) = \sum_{t=1}^\ell \|(Y - CY_c)_{:,t}\|_\Theta^2,$$

where $\|\mathbf{u}\|_\Theta^2 := \mathbf{u}^\top \Theta \mathbf{u}$. Since the objective is a sum over columns, minimizing over $Y_c$ decouples across columns. In particular, the optimal $c$-th column $(Y_c^\star)_{:,c} \in \mathbb{R}^k$ solves

$$(Y_c^\star)_{:,c} \ = \ \arg\min_{\mathbf{y} \in \mathbb{R}^k} \|Y_{:,c} - C\mathbf{y}\|_\Theta.$$

Define $\boldsymbol{\ell} := Y_{:,c} \in \mathbb{R}^p$ and write $\mathbf{y} := (Y_c)_{:,c} \in \mathbb{R}^k$. Minimizing $\|\boldsymbol{\ell} - C\mathbf{y}\|_\Theta$ is equivalent to minimizing its square

$$f(\mathbf{y}) \ := \ \|\boldsymbol{\ell} - C\mathbf{y}\|_\Theta^2 = (\boldsymbol{\ell} - C\mathbf{y})^\top \Theta (\boldsymbol{\ell} - C\mathbf{y}) = \boldsymbol{\ell}^\top \Theta \boldsymbol{\ell} - 2\mathbf{y}^\top C^\top \Theta \boldsymbol{\ell} + \mathbf{y}^\top (C^\top \Theta C)\mathbf{y}.$$

Define the coarse Laplacian $\Theta_c := C^\top \Theta C$, which is symmetric positive semidefinite. Differentiating and using $\nabla_\mathbf{y}(\mathbf{y}^\top \Theta_c \mathbf{y}) = 2\Theta_c \mathbf{y}$ and $\nabla_\mathbf{y}(\mathbf{y}^\top b) = b$ yields

$$\nabla f(\mathbf{y}) \ = \ 2\Theta_c \mathbf{y} - 2C^\top \Theta \boldsymbol{\ell}.$$

Setting $\nabla f(\mathbf{y}) = \mathbf{0}$ gives the optimality condition:

$$\Theta_c \mathbf{y} \ = \ C^\top \Theta \boldsymbol{\ell} \ = \ C^\top \Theta Y_{:,c}.$$

Collecting these optimality conditions for all $c \in [\ell]$ yields the matrix condition

$$\Theta_c Y_c = C^\top \Theta Y.$$

Since $\Theta$ is a Laplacian, $\Theta \mathbf{1}_p = \mathbf{0}$, and thus $\Theta_c \mathbf{1}_k = C^\top \Theta C \mathbf{1}_k = C^\top \Theta \mathbf{1}_p = \mathbf{0}$, implying that $\Theta_c$ is singular and the equation may have infinitely many solutions. Among all solutions, the Moore–Penrose pseudoinverse selects the unique minimum-$\ell_2$-norm solution, hence

$$Y_c^\star = \Theta_c^\dagger C^\top \Theta Y,$$

which completes the derivation.

$\square$

### A.15    Proof of Lipschitz Continuity

**Theorem 6.** *Assume there exist constants $\eta > 0$ and $\varepsilon > 0$ such that for every admissible $C$,*

$$N_i(C) \geq \eta \qquad \forall i \in \mathcal{I}_\phi,$$

*and*

$$[p_i(C)]_j \geq \varepsilon \qquad \forall i \in \mathcal{I}_\phi, \ \forall j = 1, \ldots, \ell.$$

*Rows with $i \notin \mathcal{I}_\phi$ have zero labeled mass, are excluded from the entropy objective, and have zero entropy-gradient contribution.*

*Then $r_E(\cdot, Y)$ is Lipschitz continuous on this restricted set. In particular, for any two matrices $C_1, C_2 \in \mathbb{R}^{p \times k}$*

$$|r_E(C_1, Y) - r_E(C_2, Y)| \leq \frac{2(1 + |\ln \varepsilon|)}{\eta \ln 2} \|C_1 - C_2\|_{1,1}.$$

*Consequently, since $\|M\|_{1,1} \leq \sqrt{pk}\, \|M\|_F$,*

$$|r_E(C_1, Y) - r_E(C_2, Y)| \leq \frac{2(1 + |\ln \varepsilon|)}{\eta \ln 2} \sqrt{pk}\, \|C_1 - C_2\|_F.$$

*Proof.* Let $c_i^{(1)}$ and $c_i^{(2)}$ denote the $i$-th columns of $C_1$ and $C_2$, respectively. Then the $i$-th rows of $C_1^\top Y$ and $C_2^\top Y$ are

$$\Phi_i(C_1) = \left(c_i^{(1)}\right)^\top Y, \qquad \Phi_i(C_2) = \left(c_i^{(2)}\right)^\top Y.$$

Now we define,

$$N_i^{(1)} := N_i(C_1) = \|\Phi_i(C_1)\|_1, \qquad N_i^{(2)} := N_i(C_2) = \|\Phi_i(C_2)\|_1,$$

For $i \in \mathcal{I}_\phi$, define

$$p_i(C_1) = \frac{\Phi_i(C_1)}{N_i^{(1)}}, \qquad p_i(C_2) = \frac{\Phi_i(C_2)}{N_i^{(2)}}.$$

Rows with $i \notin \mathcal{I}_\phi$ have zero labeled mass, are excluded from the entropy objective, and have zero entropy-gradient contribution. Therefore, the Lipschitz argument below applies only to the rows with positive labeled mass where the normalization is well defined.

For $i \in \mathcal{I}_\phi$, we first bound the difference of the $i$-th rows:

$$\Phi_i(C_1) - \Phi_i(C_2) = \left(c_i^{(1)} - c_i^{(2)}\right)^\top Y.$$

Hence,

$$\|\Phi_i(C_1) - \Phi_i(C_2)\|_1 = \left\|\left(c_i^{(1)} - c_i^{(2)}\right)^\top Y\right\|_1.$$

Write $x := c_i^{(1)} - c_i^{(2)} \in \mathbb{R}^p$. Then

$$x^\top Y = \sum_{u=1}^p x_u\, y_u^\top,$$

so by the triangle inequality,

$$\|x^\top Y\|_1 \le \sum_{u=1}^p |x_u|\, \|y_u\|_1 \le \sum_{u=1}^p |x_u| = \|x\|_1.$$

Therefore,

$$\|\Phi_i(C_1) - \Phi_i(C_2)\|_1 \le \|c_i^{(1)} - c_i^{(2)}\|_1.$$

Next, by the reverse triangle inequality,

$$|N_i^{(1)} - N_i^{(2)}| = |\|\Phi_i(C_1)\|_1 - \|\Phi_i(C_2)\|_1| \le \|\Phi_i(C_1) - \Phi_i(C_2)\|_1 \le \|c_i^{(1)} - c_i^{(2)}\|_1.$$

Now,

$$p_i(C_1) - p_i(C_2) = \frac{\Phi_i(C_1)}{N_i^{(1)}} - \frac{\Phi_i(C_2)}{N_i^{(2)}}.$$

Add and subtract $\Phi_i(C_2)/N_i^{(1)}$:

$$p_i(C_1) - p_i(C_2) = \frac{\Phi_i(C_1) - \Phi_i(C_2)}{N_i^{(1)}} + \Phi_i(C_2)\left(\frac{1}{N_i^{(1)}} - \frac{1}{N_i^{(2)}}\right).$$

Taking $\ell_1$-norms gives

$$\|p_i(C_1) - p_i(C_2)\|_1 \le \frac{\|\Phi_i(C_1) - \Phi_i(C_2)\|_1}{N_i^{(1)}} + \|\Phi_i(C_2)\|_1 \left|\frac{1}{N_i^{(1)}} - \frac{1}{N_i^{(2)}}\right|.$$

Since $\|\Phi_i(C_2)\|_1 = N_i^{(2)}$, this becomes

$$\|p_i(C_1) - p_i(C_2)\|_1 \le \frac{\|\Phi_i(C_1) - \Phi_i(C_2)\|_1}{N_i^{(1)}} + \frac{|N_i^{(1)} - N_i^{(2)}|}{N_i^{(1)}}.$$

Using $N_i^{(1)} \ge \eta$, we obtain

$$\|p_i(C_1) - p_i(C_2)\|_1 \le \frac{2}{\eta}\,\|c_i^{(1)} - c_i^{(2)}\|_1.$$

Next, consider the entropy function

$$H(p) = -\sum_{j=1}^\ell p_j \log_2 p_j.$$

Its partial derivatives are

$$\frac{\partial H}{\partial p_j} = -\frac{\ln p_j + 1}{\ln 2}.$$

Since $[p_i(C)]_j \ge \varepsilon$, we have on the restricted set

$$\|\nabla H(p)\|_\infty \le \frac{1 + |\ln \varepsilon|}{\ln 2}.$$

Therefore, by the mean value theorem,

$$|H(p) - H(q)| \le \frac{1 + |\ln \varepsilon|}{\ln 2}\,\|p - q\|_1$$

for all $p, q$ in this truncated simplex.

Applying this with $p = p_i(C_1)$ and $q = p_i(C_2)$, we get

$$|H(p_i(C_1)) - H(p_i(C_2))| \leq \frac{1 + |\ln \varepsilon|}{\ln 2} \|p_i(C_1) - p_i(C_2)\|_1.$$

Using the bound above,

$$|H(p_i(C_1)) - H(p_i(C_2))| \leq \frac{1 + |\ln \varepsilon|}{\ln 2} \cdot \frac{2}{\eta} \|c_i^{(1)} - c_i^{(2)}\|_1.$$

Finally, summing over $i \in \mathcal{I}_\phi$,

$$|r_E(C_1, Y) - r_E(C_2, Y)| = \left| \sum_{i \in \mathcal{I}_\phi} H(p_i(C_1)) - \sum_{i \in \mathcal{I}_\phi} H(p_i(C_2)) \right|$$

$$\leq \sum_{i \in \mathcal{I}_\phi} |H(p_i(C_1)) - H(p_i(C_2))| \leq \frac{2(1 + |\ln \varepsilon|)}{\eta \ln 2} \sum_{i \in \mathcal{I}_\phi} \|c_i^{(1)} - c_i^{(2)}\|_1.$$

Since

$$\sum_{i \in \mathcal{I}_\phi} \|c_i^{(1)} - c_i^{(2)}\|_1 \leq \sum_{i=1}^{k} \|c_i^{(1)} - c_i^{(2)}\|_1,$$

we obtain

$$|r_E(C_1, Y) - r_E(C_2, Y)| \leq \frac{2(1 + |\ln \varepsilon|)}{\eta \ln 2} \|C_1 - C_2\|_{1,1}.$$

Using $\|C_1 - C_2\|_{1,1} \leq \sqrt{pk} \|C_1 - C_2\|_F$, we further get

$$|r_E(C_1, Y) - r_E(C_2, Y)| \leq \frac{2(1 + |\ln \varepsilon|)}{\eta \ln 2} \sqrt{pk} \|C_1 - C_2\|_F.$$

This proves the claim. $\square$

### A.16  Dataset Description

**Dataset** We have performed the experiments on the datasets as shown in the Table 4.

| Dataset | Nodes | Edges | Features | Classes |
|---|---|---|---|---|
| CORA | 2,708 | 5,429 | 1,433 | 7 |
| CITESEER | 3,327 | 9,104 | 3,703 | 6 |
| DBLP | 17,716 | 52,867 | 1,639 | 4 |
| CO-CS | 18,333 | 163,788 | 6,805 | 15 |
| PUBMED | 19,717 | 44,338 | 500 | 3 |
| CO-PHYSICS | 34,493 | 247,962 | 8,415 | 5 |
| Flickr | 89,250 | 899756 | 500 | 7 |
| OGBN-Arxiv | 169343 | 1166243 | 128 | 40 |
| OGBN-Products | 2449029 | 61859140 | 100 | 47 |

Table 4: Overview of the datasets employed for node classification

## A.17 Performance Across Different GNN Architectures

| Dataset | r | Architecture | LAGC | ENGC |
|---------|-----|-------------|------|------|
| **CORA** | **0.1** | **GCN** | $84.45 \pm 0.18$ | $86.00 \pm 1.05$ |
| | | **GAT** | $80.23 \pm 0.25$ | $81.37 \pm 0.00$ |
| | | **APPNP** | $86.05 \pm 0.41$ | $87.85 \pm 0.71$ |
| **CITESEER** | **0.1** | **GCN** | $75.61 \pm 0.62$ | $78.65 \pm 1.95$ |
| | | **GAT** | $72.72 \pm 0.97$ | $77.54 \pm 0.01$ |
| | | **APPNP** | $76.40 \pm 0.21$ | $78.46 \pm 0.01$ |
| **PUBMED** | **0.01** | **GCN** | $80.91 \pm 0.08$ | $82.73 \pm 0.36$ |
| | | **GAT** | $73.92 \pm 0.20$ | $74.68 \pm 0.00$ |
| | | **APPNP** | $79.62 \pm 0.61$ | $79.55 \pm 0.02$ |

Table 5: Resulting Accuracy, where $r$ is the Coarsening Ratio

## A.18 Link Prediction Algorithm

---
**Algorithm 3:** Link Prediction using ENGC

---
**Input:** Graph $\mathcal{G}(\Theta, X, Y)$ with observed edge set $E$
**Output:** Trained link predictor $W^\star$ and test AUC

**1** Split observed edges into disjoint sets $E_{\text{train}}$, $E_{\text{val}}$, and $E_{\text{test}}$;
**2** Sample negative edge sets $N_{\text{train}}$, $N_{\text{val}}$, and $N_{\text{test}}$;
**3** Construct the training graph $\mathcal{G}_{\text{train}}(\Theta_{\text{train}}, X, Y)$ using only $E_{\text{train}}$;
**4** Apply Algorithm 1 on $\mathcal{G}_{\text{train}}$ to obtain $C$, $\Theta_c$, and $\widetilde{X}$;
**5** Define the coarsened training graph $\mathcal{G}_c = (\Theta_c, \widetilde{X})$;
**6** Form coarsened positive pairs $E_{\text{train}}^c$ from the edges of $\mathcal{G}_c$, and sample coarsened negative pairs $N_{\text{train}}^c$ from absent coarse edges;
**7** Train the link predictor on the coarsened graph $\mathcal{G}_c$:

$$W^\star \leftarrow \arg\min_W \ell_{\text{BCE}}\left(\text{GNN}_{\mathcal{G}_c}(W), E_{\text{train}}^c, N_{\text{train}}^c\right),$$

where $E_{\text{train}}^c$ and $N_{\text{train}}^c$ denote the positive and negative training pairs induced on the coarsened graph;
**8** Tune hyperparameters using $E_{\text{val}}$ and $N_{\text{val}}$;
**9** Apply the trained predictor to the original graph $\mathcal{G}(\Theta, X)$ and score original test pairs:

$$s_{uv} \leftarrow \text{GNN}_{\mathcal{G}(\Theta, X)}(W^\star)_{uv}, \qquad (u, v) \in E_{\text{test}} \cup N_{\text{test}}.$$

**10** Compute ROC-AUC using the scores $\{s_{uv}\}$ on the original test pairs;
**11** **return** $W^\star$ and test AUC;

---

## A.19 Theory-to-Empirics Validation

Table 6 provides a direct validation of the proposed theory-to-empirics mechanism. FGC does not use label information during coarsening and therefore produces comparatively mixed supernodes, as reflected by larger SII, AMM, $\delta^\star$, and $D_{\text{Dir}}$. LAGC improves over FGC by incorporating label information through a Frobenius-type regularizer, but it still does not directly penalize the uncertainty of the row-wise label distribution. In contrast, ENGC explicitly minimizes the entropy of the node-profile distribution, which leads to substantially lower label impurity and majority-vote error.

On Cora at $r = 0.1$, ENGC reduces SII from 1.2818 under LAGC to 0.3193, and AMM from 0.4002 to 0.1242. This directly supports Theorem 5, which states that lower entropy controls the majority-vote error inside each supernode. The same trend is also reflected in the projection and structural quantities: ENGC obtains a much lower $\delta^\star$ than FGC and LAGC, and also achieves the smallest Dirichlet distortion. The downstream accuracy is competitive with LAGC and clearly better than FGC, indicating that the cleaner coarsening does not come at the cost of task performance.

Table 6: Theory-to-empirics validation at $r = 0.1$. FGC is feature-based and label-agnostic, LAGC is Frobenius label-aware, and ENGC is entropy label-aware. Lower SII, AMM, EIEC, $\delta^\star$, and $D_{\mathrm{Dir}}$ indicate cleaner supernodes and better preservation of the quantities controlled by our theory. Metrics are computed post-hoc using all labels after the coarsening map has been learned.

| Dataset | Method | SII ↓ | AMM ↓ | EIEC ↓ | $\delta^\star$ ↓ | $D_{\mathrm{Dir}}$ ↓ | GCN Acc. ↑ |
|---|---|---|---|---|---|---|---|
| Cora | FGC | 1.8250 | 0.3773 | 0.4546 | 0.9226 | 0.3762 | $79.96 \pm 0.18$ |
| Cora | LAGC | 1.2818 | 0.4002 | 0.4407 | 0.6155 | 0.1434 | $86.10 \pm 0.03$ |
| Cora | ENGC | **0.3193** | **0.1242** | **0.4111** | **0.4354** | **0.1408** | $86.00 \pm 1.05$ |
| CiteSeer | FGC | 0.8078 | 0.4144 | 0.4604 | 0.9268 | 0.7730 | $69.46 \pm 0.22$ |
| CiteSeer | LAGC | 0.2080 | 0.1036 | 0.2023 | 0.8919 | 0.4849 | $76.00 \pm 0.50$ |
| CiteSeer | ENGC | **0.1360** | **0.0538** | **0.0573** | **0.5599** | **0.1410** | $\mathbf{78.65 \pm 1.95}$ |

On CiteSeer, the connection is even clearer. ENGC achieves the lowest SII, AMM, $\delta^\star$, and $D_{\mathrm{Dir}}$, and also obtains the best GCN accuracy. This shows that the theory-aligned quantities are not merely diagnostic: improvements in label purity and projection quality are accompanied by stronger downstream learning. Overall, the table supports the mechanism predicted by our analysis: entropy regularization reduces label mixing, lowers majority-vote error and projection distortion, and produces a coarsened graph that is more useful for downstream node classification.

### A.20 Ablation result discussion

Referring to Table 3 Removing $T_1$ (log-det/spectral connectivity) yields the largest degradation ($-8.59$ points), followed by ablating $T_6$ (entropy regularization, $-7.20$) and $T_2$ (feature reconstruction, $-6.39$). This indicates that (i) maintaining a well-conditioned coarse connectivity through the log-det term, (ii) enforcing label coherence via $r_E(C, Y)$, and (iii) aligning coarse and fine features are most critical for downstream accuracy on CORA. In contrast, dropping $T_4$ and $T_5$ produces moderate declines ($-3.58$ and $-3.47$), consistent with their role as stabilizing regularizers. The effect of $T_3$ is comparatively small ($-1.16$). Overall, the ablation confirms that the principal gains stem from the spectral connectivity, entropy guided coarsening, and feature similarity components introduced in the ENGC objective.

Table 7: ENGC ablation study on Cora, CiteSeer, and PubMed. Removing $T_i$ denotes dropping that term from the objective.

| Dataset / Ratio | All | $-T_1$ | $-T_2$ | $-T_3$ | $-T_4$ | $-T_5$ | $-T_6$ |
|---|---|---|---|---|---|---|---|
| Cora $r = 0.1$ | $86.00 \pm 1.05$ | $77.41 \pm 0.16$ | $79.61 \pm 0.41$ | $84.84 \pm 0.66$ | $82.42 \pm 0.01$ | $82.53 \pm 0.41$ | $78.80 \pm 0.15$ |
| Cora $r = 0.3$ | $87.05 \pm 0.05$ | $79.10 \pm 0.77$ | $79.61 \pm 1.87$ | $85.05 \pm 0.71$ | $82.19 \pm 1.29$ | $84.24 \pm 0.08$ | $80.00 \pm 0.49$ |
| CiteSeer $r = 0.1$ | $78.65 \pm 1.95$ | $67.50 \pm 0.07$ | $75.23 \pm 0.05$ | $75.77 \pm 0.25$ | $76.55 \pm 0.00$ | $75.93 \pm 0.48$ | $69.89 \pm 0.01$ |
| CiteSeer $r = 0.3$ | $79.69 \pm 0.37$ | $67.02 \pm 0.07$ | $77.28 \pm 0.57$ | $78.80 \pm 0.11$ | $75.58 \pm 0.18$ | $76.27 \pm 0.62$ | $71.58 \pm 0.45$ |
| PubMed $r = 0.01$ | $80.80 \pm 0.02$ | $70.57 \pm 0.14$ | $78.00 \pm 0.26$ | $74.12 \pm 0.11$ | $76.04 \pm 0.01$ | $76.18 \pm 0.95$ | $72.58 \pm 0.81$ |
| PubMed $r = 0.03$ | $82.73 \pm 0.36$ | $73.48 \pm 0.02$ | $80.33 \pm 0.95$ | $77.31 \pm 1.27$ | $77.00 \pm 0.94$ | $78.43 \pm 0.00$ | $74.00 \pm 0.97$ |

### A.21 Large-Scale Graph Comparison

To further evaluate scalability beyond standard citation and co-authorship benchmarks, we report additional large-scale experiments on Flickr, OGBN-Arxiv, and OGBN-Products. The comparison uses matched coarsening/condensation ratios and includes recent scalable baselines. We also report the train/validation/test split sizes, feature dimensions, and number of classes to make the large-scale setting explicit.

| Dataset | $r$ | Train/Val/Test Nodes | Features | Classes | Bonsai | GCPA | ENGC (Ours $\uparrow$) |
|---|---|---|---|---|---|---|---|
| Flickr | 0.01 | 44,625/22,312/22,313 | 500 | 7 | $48.36 \pm 0.36$ | $47.20 \pm 0.30$ | $\mathbf{48.74 \pm 0.29}$ |
| OGBN-Arxiv | 0.01 | 90,941/29,799/48,603 | 128 | 40 | $58.59 \pm 0.10$ | $\mathbf{64.09 \pm 0.20}$ | $62.36 \pm 0.07$ |
| OGBN-Products | 0.001 | 196,615/39,323/2,213,091 | 100 | 47 | $48.36 \pm 0.36$ | $68.10 \pm 0.20$ | $\mathbf{69.60 \pm 0.01}$ |

Table 8: Baseline comparison on Flickr, OGBN-Arxiv, and OGBN-Products. Higher accuracy is better.

## A.22 Additional Baseline Comparison

We further provide expanded empirical comparisons with baselines, including MGC Halder et al. (2025), BONSAI Gupta et al. (2025), SINGC Cohen & Talmon (2026), GCPA Li et al. (2025) and FedGM Zhang et al. (2025) below.

| Dataset | $r$ | MGC | SINGC | Bonsai | FedGM | GCPA | ENGC (Ours $\uparrow$) |
|---|---|---|---|---|---|---|---|
| CORA | 0.3 | $84.56 \pm 1.40$ | $84.51 \pm 0.33$ | $86.64 \pm 1.18$ | $82.19 \pm 0.81$ | $83.53$ | $\mathbf{87.05 \pm 0.05}$ |
| CORA | 0.1 | $76.02 \pm 0.93$ | $82.76 \pm 0.32$ | $85.35 \pm 1.91$ | $77.18 \pm 1.44$ | $81.60$ | $\mathbf{86.00 \pm 1.05}$ |
| CITESEER | 0.3 | $74.60 \pm 2.31$ | $76.66 \pm 0.27$ | $74.47 \pm 0.40$ | $73.19 \pm 0.94$ | $75.43$ | $\mathbf{79.69 \pm 0.37}$ |
| CITESEER | 0.1 | $70.57 \pm 1.25$ | $69.71 \pm 0.72$ | $75.98 \pm 0.52$ | $70.95 \pm 0.20$ | $73.73$ | $\mathbf{78.65 \pm 1.95}$ |

Table 9: Baseline comparison on Cora and CiteSeer. Higher accuracy is better.

## A.23 Run-time Complexity:

For an input graph with $p$ nodes, $E_1$ edges, and node features of dimension $n$, the computational cost of node classification with an $l$-layer Graph Convolutional Network (GCN) is $\mathcal{O}(lp^2 n + lpE_1 n)$ Blakely et al. (2021).

In our ENGC method, the worst-case cost of a single iteration for learning the coarsened graph is $\mathcal{O}(p^2 k)$. When coarsening is followed by node classification on the reduced graph, the total complexity becomes $\mathcal{O}(p^2 k + lk^2 n + lkE_2 n)$, where $k$ is the number of nodes and $E_2$ is the number of edges in the coarsened graph. Since typically $p \gg k$ and $E_1 \gg E_2$, and with $k < n$, the combined cost of coarsening and classification is substantially smaller than performing node classification directly on the original graph. This advantage is also reflected in Table A.23, which shows that ENGC is considerably faster than the baseline methods and has a runtime comparable to FGC Kumar et al. (2023a;b).

| Dataset($\tau$) | r = k/p | GCOND | SCAL | FGC | LAGC | ENGC | Whole dataset |
|---|---|---|---|---|---|---|---|
| CORA | 0.05 | 329.86 | 27.76 | 1.71 | 1.55 | 1.72 | 2.86 |
| CITESEER | 0.05 | 331.33 | 56.21 | 2.15 | 2.03 | 1.90 | 5.24 |
| PUBMED | 0.05 | 202.04 | 54.09 | 19.81 | 20.35 | 17.98 | 58.85 |
| CO-CS | 0.05 | 1600.32 | 180.16 | 34.45 | 49.87 | 37.04 | 72.31 |

Table 10: This table reports the runtime ($\tau$), measured in seconds, for graph coarsening and node classification at a coarsening ratio of 0.05. The results show that the proposed ENGC method is faster than several existing state-of-the-art approaches and remains competitive with the strongest optimization-based baselines. Furthermore, the combined cost of coarsening and node classification under the proposed method is lower than the cost of performing node classification directly on the original graph in the reported settings.

