# OpenReview forum: "Entropy Guided Semi-Supervised Graph Coarsening"
_TMLR — Decision pending for TMLR_

### Review · Reviewer_Bjfh · 2026-04-16

**Summary Of Contributions:**

This paper proposes an entropy-guided semi-supervised graph coarsening method that regularizes the row-wise entropy of the node-profile matrix in order to reduce label mixing within supernodes. The method combines graph structure, node features, and partial labels, and is evaluated on node classification and link prediction benchmarks. The paper also aims to provide theoretical guarantees relating entropy regularization to label purity, Dirichlet-energy preservation, cut distortion, and effective-resistance stability.


**Strengths**
- The core idea is intuitive and potentially useful: replacing a Frobenius-type label regularizer with an entropy-based one is a natural way to encourage label-homogeneous supernodes.
- The paper identifies an important weakness of existing coarsening methods, namely that low-magnitude node-profile matrices do not necessarily imply low label mixing.
- The proposed impurity-oriented metrics (SII, AMM, EIEC) are relevant and helpful for evaluating semantic quality beyond downstream accuracy.
- The authors attempt to evaluate the method across multiple datasets, tasks, and GNN backbones.

**Weaknesses**
- The framing is overly ambitious relative to what is currently supported: The paper is presented as providing strong structural and learning guarantees induced by entropy regularization, but the connection between motivation, formal claims, and empirical/theoretical support is not always clear. Several of the later guarantees also appear to rely on fairly generic perturbation-style arguments rather than clearly ENGC-specific analysis.
- The semi-supervised setting needs to be clarified much more carefully: Since unlabeled nodes are assigned zero label vectors, a supernode may receive no labeled mass at all, making the row normalization of the probabilistic node-profile matrix mathematically undefined. This issue appears in the objective, gradients, and parts of the theory, but it is not handled explicitly.
- There are technical inconsistencies in the optimization development: The formulation, constraints, updates, and convergence proof do not appear to be fully aligned. The paper introduces strong structural constraints on the assignment matrix, whereas the update rule appears to reduce to a projected-gradient-style update with nonnegativity. The appendix also introduces additional constraints that are not clearly stated in the main formulation, making it difficult to determine which optimization problem is actually being solved. Please also better organize the theorem/lemma structure and clarify which result supports which claimed contribution.
- The experimental section needs to be clarified and tightened: Figure 2 is only minimally integrated into the main text and does not sufficiently clarify the full experimental pipeline. The ablation study in Table 3 is limited to Cora at r=0.1, and removing T1 causes a larger drop than removing T6, so the paper should better explain how this supports the claimed central role of entropy regularization. The experimental pipeline is not described consistently: the main text should explain whether the learned coarse feature matrix \tilde{X} is actually used downstream or whether features are simply aggregated via C^T X. The link prediction protocol is also under-specified; because coarsening changes the node space, the paper should clearly explain how original node pairs are mapped to and evaluated after coarsening. Finally, there are issues that weaken confidence in the empirical results, including repeated values in Table 1, missing uncertainty estimates in Table 2 despite averaging over multiple splits, and runtime claims in Table 6 that do not consistently support the blanket statement that ENGC is faster than existing state-of-the-art approaches.

**Audience:**

Yes

**Audience Explanation:**

Graph data are ubiquitous in the filed, and the background story is convincing.

**Broader Impact Concerns:**

Not applicable to this paper in my opinion.

**Claims And Evidence:**

No

**Claims Explanation:**

At its current stage, I am not convinced the paper is ready for acceptance. The core idea is promising, but the paper needs a substantial reframing and a significant tightening of the theory, optimization, and experimental protocol. I would encourage the authors to reposition the contribution around entropy as a label-aware coarsening objective, reduce overclaiming, and make the empirical and mathematical story fully consistent.

**Requested Changes:**

- **Structure and Presentations**: Please clarify the overall presentation and story. Please clarify the semi-supervised setting. How is the entropy term defined when a supernode contains no labeled nodes? Please improve the framing and presentation. The paper should more clearly separate motivation, formal contribution, and what is theoretically versus empirically supported.
- **Experiments**: Please explain the full experimental pipeline more clearly in the main text. Is the learned coarse feature matrix actually used in downstream experiments, or are features simply aggregated via C and X? In link prediction, how are original node pairs evaluated after coarsening? Please clarify whether the theoretical guarantees are intended as ENGC-specific consequences of entropy regularization or as more general perturbation-style bounds that hold once reconstruction error is small.
- **Theoretical Guarantee and Appendix**: The optimization justification would also benefit from clarification, especially regarding the assumptions required for the MM/BSUM surrogate and how they relate to the appendix discussion. Clarify and explain the relationship among all statements (lemmas and theorems) to give an overview to the audience.
- Please update the manuscript to mitigate other issues in weaknesses.

---

> ### Author Response · Authors · 2026-05-24
> **Response to Reviewer Bjfh (part 1)**
>
> We sincerely thank the reviewer for the thorough and constructive review. The comments helped us substantially improve the manuscript’s framing, semi-supervised formulation, optimization explanation, theoretical organization, and experimental protocol. Below, we respond point by point to the reviewer’s concerns, and all corresponding changes have been incorporated into the revised manuscript and highlighted in blue for clarity.
>
> ### Weakness 1:The framing is overly ambitious relative to what is currently supported: The paper is presented as providing strong structural and learning guarantees induced by entropy regularization, but the connection between motivation, formal claims, and empirical/theoretical support is not always clear. Several of the later guarantees also appear to rely on fairly generic perturbation-style arguments rather than clearly ENGC-specific analysis
>
> ### Answer 1:
> We thank the reviewer for this thoughtful comment. We agree that the current framing may sound broader than what is directly established by the entropy-specific analysis, and we will revise the paper to make the scope of the claims more precise.
>
> Our theoretical contribution has two distinct parts. The first part is specific to ENGC and to the entropy regularizer. In particular, Theorem 2 shows that, under fixed supernode label mass, entropy minimization drives each row of the node-profile matrix toward a one-hot distribution, corresponding to label-homogeneous supernodes, whereas the Frobenius regularizer favors uniform rows. This formally supports the use of entropy for reducing label mixing within supernodes. Lemma 2 then connects this label homogeneity to the projection-distortion term $\delta^\star$, showing that
>
> $$
> \\delta^\\star \\lesssim
> \\frac{
> \\sqrt{\\sum_i |S_i|^2 h_2^{-1}\\!\\left(H(p_i)\\right)}
> }{
> \\|Y\\|_{\\Theta,F}
> }.
> $$
>
> Thus, the entropy term enters the bound explicitly: reducing row-wise entropy reduces label impurity inside supernodes, which in turn improves the label-projection residual
>
> $$
> \\|Y-CY_c^\\star\\|_{\\Theta,F}.
> $$
>
> Finally, Theorem 5 connects this structural quantity to learning by showing that the entropy of each super node label distribution upper bounds the majority-vote classification error. These results are specific to the entropy-guided construction and explain why ENGC tends to produce semantically cleaner super nodes.
>
> The second part of the analysis concerns Dirichlet-energy distortion, cut preservation, and effective-resistance deviation. We agree with the reviewer that Theorem 3, Corollary 4, and Lemma 3 use perturbation-style arguments and are not, by themselves, unique to entropy regularization. Our intention was not to claim that these perturbation tools are ENGC-specific. Rather, these results translate the entropy-controlled projection distortion $\delta^\star$ into standard structural quantities of interest in graph coarsening. In this sense, entropy affects these guarantees indirectly: it improves the underlying projection residual, and the perturbation bounds then show how a smaller residual leads to tighter structural-preservation guarantees.
>
> To avoid overstatement, we will revise the presentation in three ways. First, we will explicitly distinguish the entropy-specific results, namely Theorem 2, Lemma 2, and Theorem 5, from the more general perturbation-based consequences. Second, we will soften claims such as "entropy induces structural guarantees'' to "entropy improves the projection-distortion term that appears in structural preservation bounds.'' Third, we will clarify that the Dirichlet, cut, and effective-resistance results are general coarsening guarantees whose tightness benefits from the entropy-driven reduction in label mixing. However, to the best of our knowledge, such guarantees have not previously been established in the graph coarsening literature. Therefore, these results are new from the perspective of graph coarsening. Moreover, as established in our paper, the use of entropy regularization leads to tighter versions of these guarantees by improving the underlying projection-distortion bound $\\delta^\\star$.
>
> Empirically, we also evaluate the specific mechanism predicted by the theory. In addition to downstream accuracy, we report label-mixing metrics such as SII, AMM, and EIEC, which directly measure whether entropy regularization produces label-homogeneous supernodes. For example, on Cora, ENGC reduces SII and AMM relative to the Frobenius-based baseline, while EIEC remains a valid upper ceiling as predicted by Theorem 5.  Also we have added additional experiments in the manuscript in the Appendix section comparing FGC,LAGC and ENGC on $\delta^*$, Dirichlet energy distortion and effective resistance distortion which directly connects between theoretical guarantee and downstream empirical gains. This makes the connection clearer in the revised manuscript so that the motivation, formal claims, and empirical evidence are aligned.

---

> ### Author Response · Authors · 2026-05-24
> **Response to Reviewer Bjfh (part 2)**
>
> ### Weakness 2: The semi-supervised setting needs to be clarified much more carefully: Since unlabeled nodes are assigned zero label vectors, a supernode may receive no labeled mass at all, making the row normalization of the probabilistic node-profile matrix mathematically undefined. This issue appears in the objective, gradients, and parts of the theory, but it is not handled explicitly.
>
> ### Answer 2:
> We thank the reviewer for pointing out this edge case. In our semi-supervised setting, zero label vectors indicate missing label information, not an additional class. During coarsening, ENGC uses only the training labels through $ \\phi=C^\\top Y_{\\mathrm{train}}, $ where validation/test/unlabeled nodes have zero label vectors.
>
> We revised the manuscript to define the normalized node-profile row only when its labeled mass is nonzero:
> $[\phi']_{i:}= \frac{\phi_i}{\\|[\\phi^\\top]_i\\|_1} \text{if } \\|[\\phi^\\top]_i\\|_1 > 0$
>
> and the row is treated as zero otherwise. Hence, if $\\|[\\phi^\\top]_i\\|_1=0$, supernode $i$ becomes unlabeled. Also it is not normalized, does not contribute to the entropy term, and the entropy-gradient contribution for that row is zero.
>
> Importantly, such a supernode is not removed from the coarsened graph. It still participates in learning the coarsening matrix $C$ through the Laplacian, feature-reconstruction, and structural terms.The same convention is used in downstream GNN training. Unlabeled supernodes remain in the coarsened graph and participate in message passing through $\\Theta_c$ and $\\widetilde X$, but they are excluded from the supervised cross-entropy loss and backpropagation. This follows the standard semi-supervised GNN protocol, where the full graph is used for message passing while the supervised loss is computed only on nodes with observed labels [1] [2].
>
> We have added this clarification in the manuscript, including the condition $\\|[\\phi^\\top]_i\\|_1>0$ in the definition of $\\phi^{\\prime}$ and in the Section 5.1 explaining how zero-label-mass supernodes are handled in the entropy objective, gradient update, and downstream GNN training.
>
> ### Weakness 3: There are technical inconsistencies in the optimization development: The formulation, constraints, updates, and convergence proof do not appear to be fully aligned. The paper introduces strong structural constraints on the assignment matrix, whereas the update rule appears to reduce to a projected-gradient-style update with nonnegativity. The appendix also introduces additional constraints that are not clearly stated in the main formulation, making it difficult to determine which optimization problem is actually being solved. Please also better organize the theorem/lemma structure and clarify which result supports which claimed contribution.
> ### Answer 3:
> Thank you for bringing up this point. In particular, we now explicitly distinguish the hard assignment set $\mathcal{C}$ in Eq. (2), which describes the desired final coarsening structure, from the relaxed feasible set $\mathcal{S}_c$ over which ENGC is optimized. The hard assignment set is used only to define the ideal non-overlapping coarsening map. Since directly optimizing over this set is combinatorial, ENGC instead solves a continuous relaxation in which $C$ is a nonnegative soft assignment matrix.
>
> We have revised the formulation and updated the manuscript so that the subproblem, update rule, and convergence statement are all stated over the same relaxed feasible set. We have also clarified that Eq. (13) is a projected first-order MM/BSUM update for the relaxed problem, not an exact optimizer over the hard assignment set. After convergence, the learned soft assignment matrix is converted into a hard coarsening map by assigning each node to the supernode with the largest entry in its row.
>
> We have also updated the appendix so that the KKT conditions and convergence proof match the relaxed problem stated in the main text. Any auxiliary assumptions used in the proof, such as boundedness and Lipschitz continuity of the gradient, are now stated explicitly rather than appearing as additional implicit constraints.
>
> Finally, we have reorganized the theorem and lemma structure to clarify the role of each result. Lemma 1 and Theorem 1 now support the relaxed MM/BSUM solver; Theorem 2, Lemma 2, and Theorem 5 provide the entropy-specific analysis; and Theorem 3, Corollary 4, and Lemma 3 provide the structural preservation guarantees whose tightness improves when entropy reduces the projection distortion $\delta^\star$. This revision makes clear which result supports each claimed contribution and aligns the formulation, constraints, update rule, and convergence proof.
>
>
>
>
> [1] Kipf, T. N. and Welling, M. Semi-Supervised Classification with Graph Convolutional Networks.
> ICLR, 2017.
>
> [2]  Yang, Z., Cohen, W. W., and Salakhutdinov, R. Revisiting Semi-Supervised Learning with Graph Embeddings. ICML, 2016.

---

> ### Author Response · Authors · 2026-05-24
> **Response to Reviewer Bjfh (part 3)**
>
> Since **Weakness 4** raises several distinct concerns, we have separated the reviewer’s question and our response into five parts for clarity.
>
> ### Weakness 4.1: The experimental section needs to be clarified and tightened. Figure 2 is only minimally integrated into the main text and does not sufficiently clarify the full experimental pipeline.
>
> ### Answer 4.1:
> We thank the reviewer for this helpful comment. We agree that the experimental section should more clearly explain the full pipeline and better connect Figure 2 to the experimental protocol. We have revised the manuscript accordingly.
>
> First, we have expanded the discussion of Figure 2 so that it is no longer only a schematic illustration, but also serves as an overview of the experimental workflow. The revised text now explicitly explains the complete pipeline: starting from the original partially labeled graph $\mathcal{G}(\Theta, X, Y)$, ENGC uses the graph structure, node features, and available training labels to learn the node-to-supernode mapping $C$. This mapping is then used to construct the coarsened graph $\mathcal{G}_c(\Theta_c,\widetilde X,\widetilde Y_c))$. A downstream model is trained on the coarsened graph, and performance is evaluated on the original graph for node classification and on held-out edge pairs for link prediction.
>
> We have also clarified the node-classification procedure in Section 5.2. In this setting, ENGC is given only the training labels during coarsening; validation and test labels are masked and represented as zero label vectors. After coarsening, each supernode label is assigned by majority vote using only $C^\top Y_{\mathrm{train}}$. The downstream GNN is then trained on the coarsened graph using only supernodes that contain at least one training-labeled node, and the trained model is evaluated on the original validation/test nodes. This makes clear that validation and test labels are not used during coarsening.
>
> Similarly, we have revised Section 5.3 to clarify the link-prediction pipeline. The edge split is performed before coarsening, producing disjoint training, validation, and test edge sets. ENGC is run only on the training graph, so the validation and test edges are not used to construct the coarsened graph. The link predictor is trained using the coarsened training graph, tuned on the validation split, and evaluated on held-out test pairs using ROC-AUC. This clarification ensures that the link-prediction evaluation follows a proper train/validation/test protocol. Overall, the revised experimental section now more clearly explains the full pipeline in Figure 2, separates the node-classification and link-prediction protocols, and provides a more balanced interpretation of the ablation results. These changes make the empirical evidence better aligned with the paper’s claims about the role of entropy regularization in ENGC.

---

> ### Author Response · Authors · 2026-05-24
> **Response to Reviewer Bjfh (part 4)**
>
> ### Weakness 4.2: The ablation study in Table 3 is limited to Cora at $r=0.1$, and removing $T_1$ causes a larger drop than removing $T_6$, so the paper should better explain how this supports the claimed central role of entropy regularization.
>
> ### Answer 4.2:
> Regarding the ablation study, we agree that the original Table 3, which only reported Cora at $r=0.1$, was too limited to fully support the discussion. We have therefore expanded the ablation study. In addition to Cora at $r=0.1$, the revised manuscript now includes ablations on Cora at $r=0.3$, CiteSeer at $r=0.1$ and $r=0.3$, and PubMed. These additional experiments show that the contribution of the entropy term is not specific to a single dataset or a single coarsening ratio.
>
> We have also revised the interpretation of the ablation results to avoid overstating the role of entropy. Our claim is not that the entropy term is always the largest numerical contributor in every setting. Rather, entropy regularization is central to the label-aware component of ENGC because it is the only term that directly penalizes label mixing inside supernodes through $T_6=\nabla r_E(C,Y)$. By contrast, $T_1=-2\gamma \Theta C(C^\top \Theta C+J)^{-1}$ is the log-determinant/spectral-connectivity term, which controls the connectedness and stability of the learned coarse graph. Therefore, removing $T_1$ can strongly affect the feasibility and quality of the coarsened graph, and a large drop from removing $T_1$ is expected.
>
> This does not weaken the importance of entropy. Instead, it shows that ENGC relies on two complementary mechanisms: spectral connectivity from $T_1$ and label-homogeneous grouping from $T_6$. In the original Cora $r=0.1$ ablation, the full model achieves $86.00$, while removing $T_6$ reduces performance to $78.80$, a drop of $7.20$ points. This drop is larger than the drops caused by removing several other terms, including $T_3$, $T_4$, and $T_5$. The expanded ablation results further confirm that removing the entropy term consistently weakens downstream performance, supporting its role as an important component of the proposed label-aware coarsening framework.
>
> **Table: ENGC ablation study on Cora, CiteSeer, and PubMed. Removing Tᵢ denotes dropping that term from the objective.**
>
> | Dataset / Ratio | All | −T₁ | −T₂ | −T₃ | −T₄ | −T₅ | −T₆ |
> |---|---:|---:|---:|---:|---:|---:|---:|
> | Cora r=0.1 | 86.00 ± 1.05 | 77.41 ± 0.16 | 79.61 ± 0.41 | 84.84 ± 0.66 | 82.42 ± 0.01 | 82.53 ± 0.41 | 78.80 ± 0.15 |
> | Cora r=0.3 | 87.05 ± 0.05 | 79.10 ± 0.77 | 79.61 ± 1.87 | 85.05 ± 0.71 | 82.19 ± 1.29 | 84.24 ± 0.08 | 80.00 ± 0.49 |
> | CiteSeer r=0.1 | 78.65 ± 1.95 | 67.50 ± 0.07 | 75.23 ± 0.05 | 75.77 ± 0.25 | 76.55 ± 0.00 | 75.93 ± 0.48 | 69.89 ± 0.01 |
> | CiteSeer r=0.3 | 79.69 ± 0.37 | 67.02 ± 0.07 | 77.28 ± 0.57 | 78.80 ± 0.11 | 75.58 ± 0.18 | 76.27 ± 0.62 | 71.58 ± 0.45 |
> | PubMed r=0.01 | 80.80 ± 0.02 | 70.57 ± 0.14 | 78.00 ± 0.26 | 74.12 ± 0.11 | 76.04 ± 0.01 | 76.18 ± 0.95 | 72.58 ± 0.81 |
> | PubMed r=0.03 | 82.73 ± 0.36 | 73.48 ± 0.02 | 80.33 ± 0.95 | 77.31 ± 1.27 | 77.00± 0.94 | 78.43 ± 0.00 | 74.00 ± 0.97 |
>
> ### Weakness 4.3: The experimental pipeline is not described consistently. The main text should explain whether the learned coarse feature matrix $\widetilde X$ is actually used downstream or whether features are simply aggregated via $C^\top X$.
>
> ### Answer 4.3:
> We thank the reviewer for highlighting this point. The intended pipeline is that the downstream GNN uses the coarse feature matrix associated with the learned coarsened graph. In the optimization section, this matrix is denoted by $\widetilde X$, and it is updated after fixing $C$. The feature update is given by $\widetilde X^{(t+1)}=\left(\frac{2}{\alpha}C^\top \Theta C+C^\top C\right)^{-1}C^\top X$. Thus, $\widetilde X$ is the feature matrix passed to the downstream model on the coarsened graph $G_c=(\Theta_c,\widetilde X,\widetilde Y)$.
>
> The confusion comes from Algorithm 2, where we wrote $X' = P^\dagger X, \quad P^\dagger = C^\top$. This notation was intended to describe how original node features are represented in the coarse node space. However, we agree that writing $X'=P^\dagger X$ in Algorithm 2 can create ambiguity about whether the downstream model uses the optimized coarse feature matrix $\widetilde X$ or a direct aggregation $C^\top X$. We have revised this notation and rewritten Algorithm 2 in the manuscript so that the experimental pipeline is consistent with the optimization development.

---

> ### Author Response · Authors · 2026-05-24
> **Response to Reviewer Bjfh (part 5)**
>
> ### Weakness 4.4: The link prediction protocol is also under-specified; because coarsening changes the node space, the paper should clearly explain how original node pairs are mapped to and evaluated after coarsening.
>
> ### Answer 4.4:
> We thank the reviewer for pointing out that the link-prediction protocol was under-specified. We have updated the experimental section and revised Algorithm 3 (added to Appendix A.17) to explicitly describe how link prediction is performed when the training graph is coarsened.
>
> In our experiments, ENGC is used as a training-time graph reduction mechanism, not as a change in the evaluation space. Following the train-on-coarsened/evaluate-on-original protocol used in prior graph coarsening and GNN-training works such as ConvMatch [1] and LAGC [2], we first split the observed edges into disjoint training, validation, and test sets: $E = E_{\mathrm{train}} \cup E_{\mathrm{val}} \cup E_{\mathrm{test}}.$ Negative samples are generated separately for each split. ENGC is then fitted only on the training graph induced by $E_{\mathrm{train}}$. Thus, validation and test edges are never used during coarsening.
>
> The learned mapping $C$ is used to construct the coarsened training graph. The link predictor is trained on this coarsened graph, which reduces the training cost. However, validation and test evaluation are still performed in the original node space. In other words, the held-out validation/test node pairs $(u,v)$ are not replaced by coarse node pairs during final evaluation.
>
> This is possible because the downstream link predictor is GNN-based and learns shared message-passing parameters, rather than parameters tied to individual coarse nodes. After training on the coarsened graph, the learned GNN parameters are applied back to the original graph to compute embeddings for the original nodes. The original held-out positive and negative pairs are then scored using these original-node embeddings, and ROC-AUC is computed from these scores.
>
> Thus, coarsening affects only the graph used for training the predictor. The final link-prediction performance is evaluated on the original validation/test node pairs, ensuring that the reported AUC measures performance in the original graph space. We have added this clarification to Section 5.3 so that the link-prediction pipeline is fully specified.
> ### Weakness 4.5:  Finally, there are issues that weaken confidence in the empirical results, including repeated values in Table 1, missing uncertainty estimates in Table 2 despite averaging over multiple splits, and runtime claims in Table 6 that do not consistently support the blanket statement that ENGC is faster than existing state-of-the-art approaches.
>
> ### Answer 4.5:
> We thank the reviewer for highlighting this issue. First, regarding Table 1, we corrected the accidentally duplicated entries in the manuscript.
>
> Second, regarding Table 2, the reviewer is correct that the current table reports only the mean AUC values, even though Section 5.3 states that link-prediction results are averaged over multiple random splits. We have revised the manuscript and added the corresponding uncertainty estimates.
>
> Third, regarding runtime claims in Table 6, we agree that the current wording is too broad. The intended conclusion is not that ENGC is uniformly faster than every baseline on every dataset. Rather, Table 6 shows that ENGC is substantially faster than expensive condensation/scaling baselines such as GCOND and SCAL, is competitive with optimization-based coarsening baselines such as FGC/LAGC, and is consistently faster than training on the whole graph in the reported settings. However, there are cases where ENGC is slightly slower than FGC or LAGC, for example on Cora or CO-CS. We have revised the runtime claim in the manuscript accordingly.
>
>
> [1] Charles Dickens, Edward Huang, Aishwarya Reganti, Jiong Zhu, Karthik Subbian, and Danai Koutra. 2024. Graph Coarsening via Convolution Matching for Scalable Graph Neural Network Training. In Companion Proceedings of the ACM Web Conference 2024 (WWW '24). Association for Computing Machinery, New York, NY, USA, 1502–1510
>
> [2] Manoj Kumar, Subhanu Halder, Archit Kane, Ruchir Gupta, and Sandeep Kumar. 2024. Optimization framework for semi-supervised attributed graph coarsening. Proceedings of the 40th Conference on Uncertainty in Artificial Intelligence (UAI), 2024.

---

> ### Author Response · Authors · 2026-05-24
> **Response to Reviewer Bjfh (part 6)**
>
> Below, we address each of the requested changes in detail, with our responses and corresponding manuscript updates outlined below.
>
> ### Requested Change 1: **Structure and Presentations:** Please clarify the overall presentation and story. Please clarify the semi-supervised setting. How is the entropy term defined when a supernode contains no labeled nodes? Please improve the framing and presentation. The paper should more clearly separate motivation, formal contribution, and what is theoretically versus empirically supported.
>
> ### Response 1:
> We have revised the manuscript to make the presentation more focused and better aligned with the claims. The introduction now separates the **motivation** from the **formal contribution**: ENGC is motivated by reducing label mixing in supernodes, and the main methodological contribution is the entropy-based regularization of the node-profile matrix.
>
> We also clarified the semi-supervised setting. Validation, test, and unlabeled nodes are assigned zero label vectors only to indicate missing labels; they are not treated as an additional class. If a supernode contains no training-labeled nodes, it is treated as unlabeled. It still participates in coarsening through the graph structure and node features, but it does not contribute to the entropy term, its entropy-gradient contribution is zero, and it is excluded from the supervised GNN loss.
>
> Finally, we revised the theory discussion to distinguish what is **entropy-specific** from what is **general structural analysis**. The entropy-specific results explain why ENGC reduces label mixing and improves label homogeneity, while the later Dirichlet, cut, and effective-resistance results are structural perturbation-style guarantees that become tighter when entropy reduces the projection distortion. This makes clear what is theoretically supported and what is empirically validated through accuracy and label-mixing metrics.
>
> ### Requested Change 2:  **Experiments:** Please explain the full experimental pipeline more clearly in the main text. Is the learned coarse feature matrix actually used in downstream experiments, or are features simply aggregated via $C$ and $X$? In link prediction, how are original node pairs evaluated after coarsening? Please clarify whether the theoretical guarantees are intended as ENGC-specific consequences of entropy regularization or as more general perturbation-style bounds that hold once reconstruction error is small.
>
> ### Response 2:
> We have revised the experimental section to make the full pipeline explicit; we have also rewritten Algorithms 2 and 3 to enhance clarity. For node classification, the node split is performed before coarsening. ENGC is run using the original graph structure, node features, and only the training labels $Y_{\mathrm{train}}$. Algorithm 1 then returns $C$, $\Theta_c$, and the learned coarse feature matrix $\widetilde X$. The downstream GNN is trained on the coarsened graph $G_c=(\Theta_c,\widetilde X)$, so $\widetilde X$ is the feature matrix used in the experiments, not an ambiguous direct aggregation $C^\top X$. We also rewrote Algorithm 2 to remove the earlier confusing notation $X'=P^\dagger X$.
>
> For link prediction, we clarified that the edge split is performed before coarsening: $E=E_{\mathrm{train}}\cup E_{\mathrm{val}}\cup E_{\mathrm{test}}$. ENGC is fit only on the training graph. The link predictor is trained using the coarsened training graph, but validation/test node pairs remain in the original node space. Since the GNN learns shared message-passing and scoring parameters, the trained predictor can be applied back to the original graph to score the original held-out node pairs, and ROC-AUC is computed on those original pairs.
>
> We also clarified the theoretical scope. Theorem 2, Lemma 2, and Theorem 5 are ENGC/entropy-specific. The Dirichlet-energy, cut-preservation, and effective-resistance guarantees are perturbation-style structural bounds that become tighter when entropy reduces the projection distortion $\delta^\star$.

---

> ### Author Response · Authors · 2026-05-24
> **Response to Reviewer Bjfh (part 7)**
>
> ### Requested Change 3:  **Theoretical Guarantee and Appendix:** The optimization justification would also benefit from clarification, especially regarding the assumptions required for the MM/BSUM surrogate and how they relate to the appendix discussion. Clarify and explain the relationship among all statements (lemmas and theorems), to give an overview to the audience.
>
> ### Response 3:
>
> We thank the reviewer for this helpful suggestion. We have revised the optimization discussion. Specifically, we now clarify that the hard assignment set $\mathcal{C}$ describes the desired final non-overlapping coarsening map, while the MM/BSUM algorithm solves a continuous relaxation over the set $S_c$, where $C\in\mathbb{R}_+^{p\times k}$ and $\|[C^\top]_i\|_2^2\le 1$ for all $i$. The surrogate in the $C$-update is constructed under the standard MM/BSUM assumption that the smooth part has a Lipschitz-continuous gradient. The resulting projected update is therefore a step for the relaxed problem, not an exact optimizer over the hard combinatorial assignment set.
>
> We also clarified the connection to the appendix: Appendix A.1 derives the projected update using KKT conditions, while Appendix A.2 shows that the alternating sequence converges to a KKT/stationary point of the relaxed problem. Thus, the optimization justification is now aligned with the formulation and the assumptions used in the proof.
>
> Finally, we reorganized the theoretical discussion to make the role of each result clear: Lemma 1 and Theorem 1 justify the MM/BSUM solver and convergence; Theorem 2, Lemma 2, and Theorem 5 provide the entropy-specific guarantees; and Theorem 3, Corollary 4, and Lemma 3 provide the structural preservation guarantees that become tighter when entropy reduces the projection distortion $\delta^\star$.
>
> ### Requested Change 4: Please update the manuscript to mitigate other issues in weaknesses.
>
> ### Response 4:
> We thank the reviewer for their constructive feedback, which has been instrumental in revising our manuscript. We have updated the paper to comprehensively address the concerns raised in the weaknesses.
>
> First, we revised the theoretical framing to separate entropy-specific results from general structural perturbation-style guarantees. In particular, Theorem 2, Lemma 2, and Theorem 5 are now presented as the main entropy-specific results, while the Dirichlet-energy, cut, and effective-resistance bounds are clarified as general coarsening guarantees that become tighter when entropy reduces the projection distortion $\delta^\star$.
>
> Second, we clarified the semi-supervised setting. Supernodes with no labeled mass are explicitly treated as unlabeled. They are not normalized in the probabilistic node-profile matrix, do not contribute to the entropy loss or entropy gradient, and are excluded from the supervised GNN loss. However, they remain in the coarsened graph and still contribute through structure and features during message passing, but do not participate in calculating the loss or backpropagation. We have added this explicitly in the experimental section of the manuscript.
>
> Third, we revised the optimization discussion to distinguish the ideal hard assignment constraints from the continuous relaxation solved by the MM/BSUM algorithm. We clarified that Eq. (13) is a projected first-order update for the relaxed problem, not an exact optimizer over the hard combinatorial feasible set. We also aligned the main formulation, update rule, and appendix assumptions, and reorganized the theorem/lemma structure to show which result supports which claim.
>
> Fourth, we tightened the experimental section. We rewrote Algorithm 2 to clearly describe the node-classification pipeline: split the nodes before coarsening, use only $Y_{\mathrm{train}}$ during ENGC, train the GNN on $\mathcal{G}_c=(\Theta_c,\widetilde X)$, and evaluate on the original graph. We clarified that $\widetilde X$, the learned coarse feature matrix from Algorithm 1, is used downstream. We also clarified the link-prediction protocol, where edge splits are made before coarsening and evaluation is performed on original held-out node pairs using the trained GNN parameters.
>
> Finally, we corrected empirical-reporting issues: repeated entries in Table 1 were fixed, uncertainty estimates were added to Table 2, and the runtime claim in Table 6 was softened to state that ENGC is efficient and competitive, rather than uniformly faster than every baseline. We also expanded the ablation study beyond Cora $r=0.1$ and clarified that $T_1$ controls spectral connectivity while $T_6$ is the central label-aware entropy term.

---

> > ### Comment · Reviewer_Bjfh · 2026-05-25
> > **Update after revision**
> >
> > I appreciate the authors’ substantial revision and detailed responses. The revised manuscript is clearer in several respects, especially regarding the semi-supervised setting, the node-classification pipeline, the use of the learned coarse feature matrix, and the distinction between entropy-specific claims and more general structural bounds. The additional experiments and appendices may also be helpful for researchers in the field.
> >
> > However, I am not yet convinced that the main technical consistency issues have been fully resolved. In particular, the treatment of zero-labeled-mass supernodes is now described in prose, but the entropy objective, its gradient, and the Lipschitz/convergence argument still appear to rely on divisions by the labeled row mass or assumptions of positive labeled mass. Similarly, the distinction between the hard assignment set and the relaxed feasible set is improved, but the notation in the C-subproblem and surrogate still appears inconsistent. The link-prediction protocol is also clearer than before, but the construction of coarsened positive/negative training pairs remains under-specified.
> >
> > Overall, I find the core idea promising and the issues potentially addressable, but I am not yet convinced, possibly due in part to presentation issues, that the current revision fully resolves the formulation, optimization, and experimental-protocol concerns raised in my review. Although the revised PDF contains clearer explanations and concepts, I feel that further revision is required before I can recommend acceptance.

---

> > > ### Author Response · Authors · 2026-06-01
> > > **Response to Reviewer Bjfh: Further Clarifications Following the Revised Manuscript (Part 1/2)**
> > >
> > > ### Reviewer Concern 1: The treatment of zero-labeled-mass supernodes is now described in prose, but the entropy objective, its gradient, and the Lipschitz/convergence argument still appear to rely on divisions by the labeled row mass or assumptions of positive labeled mass.
> > > ### Response 1:
> > > We thank the reviewer for pointing out this important technical consistency issue. We agree that the treatment of zero-labeled-mass supernodes should be reflected directly in the entropy definition, entropy objective, gradient expression, and Lipschitz/convergence discussion._
> > >
> > > We have revised the manuscript accordingly. First, in Section 2.3, we clarified that the row-wise entropy is defined only for a supernode with positive labeled mass, i.e., when $\||[\phi']_i\||_1>0$. In this case, the normalized row $\phi'_i$ is a valid probability distribution and its entropy is computed as $H([\phi'_i])$. For a supernode with zero labeled mass, i.e., $\||[\phi']_i\||_1=0$, we set the row of $[\phi']_i=0_l^T$ as in Eq. (5). This row is treated as unlabeled and is not interpreted as a probability distribution.
> > >
> > > Second, we introduced the set $\mathcal{I}_\phi$  at begining of Section 3 and revised the entropy regularizer so that it is computed only over supernodes with positive labeled mass. Thus, Eq.(7) and the entropy term in Eq.(10) sum only over  the set of supernodes with positive labeled mass, where the normalized row is well-defined. Supernodes with zero labeled mass are excluded from the entropy objective.
> > >
> > > Third, we revised the entropy-gradient expression after Eq.(12). The gradient is now written column-wise with respect to the supernode index. For a supernode with positive labeled mass, i.e., $i\in\mathcal I_\phi$, the corresponding $i$-th column of $\nabla_C r_E(C,Y)$ is computed using the normalized entropy-gradient formula. For a supernode with zero labeled mass, i.e., $i\notin\mathcal I_\phi$, we explicitly set the corresponding entropy-gradient contribution to zero, $[\nabla_C r_E(C,Y)]_{:i}=0$. Therefore, the entropy-gradient computation never divides by $\||[\phi^\top]_i\||_1$ when this quantity is zero.
> > >
> > > Importantly, a zero-labeled-mass supernode is not removed from the coarsened graph or from the optimization. It still participates through the Laplacian term, feature-reconstruction term, log-determinant term, and structural regularizers. Only the entropy component is inactive for that supernode because no observed training label is available to define a meaningful label distribution.
> > >
> > > Finally, we revised the Lipschitz/convergence discussion in Appendix A.15 to follow the same convention. The positivity assumptions are imposed only on rows with positive labeled mass, i.e., $i\in\mathcal I_\phi$, where the normalization is well-defined. Rows with zero labeled mass are excluded from the entropy sum and have zero entropy-gradient contribution. These revisions make the entropy objective, entropy-gradient computation, and Lipschitz/convergence argument consistent with the zero-labeled-mass case.
> > >
> > >
> > >
> > > ### Reviewer Concern 2: Similarly, the distinction between the hard assignment set and the relaxed feasible set is improved, but the notation in the C-subproblem and surrogate still appears inconsistent.
> > > ### Response 2:
> > > We thank the reviewer for pointing out this remaining inconsistency. We agree that the distinction between the hard assignment set and the relaxed feasible set should be reflected consistently in the optimization notation.
> > > We have revised the optimization section accordingly. The hard assignment set $\mathcal C$ is now used only to describe the desired final non-overlapping node-to-supernode assignment. The actual MM/BSUM optimization is performed over the relaxed feasible set $\mathcal S_c$. Therefore, we have changed Eq.(10), the $C$-subproblem, and Eq.(12), the surrogate problem, so that both consistently use $C\in\mathcal S_c$.

---

> > > ### Author Response · Authors · 2026-06-01
> > > **Response to Reviewer Bjfh: Further Clarifications Following the Revised Manuscript (Part 2/2)**
> > >
> > > ### Reviewer Concern 3: Construction of coarsened positive/negative training pairs remains under-specified.
> > > ### Response 3:
> > > We thank the reviewer for this helpful comment. Although the revised draft made the overall link-prediction pipeline clearer, in this revision we will try to explain the construction of the coarsened positive and negative training pairs more explicitly. We have also revised the manuscript further to make this point clearer.
> > >
> > > Concretely, for each split, we first form the original train/validation/test edge sets in the original node space. The validation and test negative pairs are also sampled in the original node space for final evaluation. ENGC is then applied only to the training graph, from which we learn the mapping matrix $C$. Using this learned map, we construct the coarse operator $\Theta_c = C^\top \Theta_{\mathrm{train}} C$. This defines the coarsened training graph $\mathcal{G_c}=(C_c,E_c)$, whose edge set is given by $E_c = \{(a,b)\in V_c\times V_c:\ a\neq b,\ [\Theta_c]_{ab}\neq 0\}$.
> > >
> > > The coarsened positive training pairs are the edges present in this
> > > coarsened training graph. That is, they are obtained directly from $E_c$,
> > > rather than by separately mapping or reusing the original training pairs
> > > in $E_{\rm train}$. The coarsened negative training pairs are sampled after
> > > coarsening from supernode pairs that are absent from $E_c$. Hence, the
> > > negative training pairs are also constructed in the coarsened node space and
> > > are not inherited from the original negative set.
> > >
> > > Self-pairs are excluded by construction, sampled pairs already present in
> > > $E_c$ are removed from the negative set, and duplicate coarse pairs are
> > > collapsed. We then sample a matched number of unique coarse negative pairs
> > > relative to the coarse positive pairs. The GNN link predictor is trained only
> > > on these coarsened positive and negative pairs.
> > >
> > > After training, the learned model parameters are used for inference on the
> > > original validation/test graphs, and the final ROC-AUC is computed using the
> > > original positive and negative validation/test edge sets. Thus, coarsening is
> > > used only to reduce the training graph and training pairs, while evaluation
> > > remains in the original node space.
> > >  Accordingly, we have updated Section 5.3 and Algorithm 3 to make both the construction of the coarsened training pairs and the original-space evaluation protocol explicit.

---

> > > > ### Comment · Reviewer_Bjfh · 2026-06-01
> > > > **Update after 2nd revision**
> > > >
> > > > Thank you for the further revision and clarification. I appreciate that the latest manuscript addresses several of my earlier concerns more directly. Before finalizing my recommendation, if possible, I would like to ask for two points:
> > > >
> > > > First, since $I_\phi = \\{ i : \|[\phi^\top]\_i\|\_1 > 0 \\}$ depends on the current assignment matrix $C$, the entropy objective appears to have a changing active set during optimization. Could the authors clarify how the MM/BSUM convergence argument handles the case where a supernode's labeled mass changes from zero to positive, or from positive to zero, during the iterations? Is the convergence claim intended to hold only on regions where the active set is fixed, or is there an argument that covers active-set changes?
> > > >
> > > > Second, the link-prediction protocol is clearer now, but I would appreciate a bit more detail on the construction of $E^c_{\mathrm{train}}$ and $N^c\_{\mathrm{train}}$. In particular, when the coarsened adjacency is weighted, how are positive coarse edges defined? Are self-pairs removed? Are duplicate coarse pairs collapsed? How are negative coarse pairs sampled to avoid collisions with positive coarse edges? These details would help make the link-prediction experiments more reproducible.
> > > >
> > > > Overall, I appreciate the authors' substantial effort. These remaining questions are mainly about technical precision and reproducibility, rather than the core motivation of the paper.

---

> > > > > ### Author Response · Authors · 2026-06-02
> > > > > **Response to Reviewer on Technical precision and Reproducibility (Part 1/2)**
> > > > >
> > > > > We sincerely thank the reviewer for their constructive and thoughtful engagement throughout the review process. Their comments have significantly helped us improve the quality, rigor, and presentation of the paper. We greatly appreciate the opportunity to further clarify these two remaining points, and we address them below.
> > > > > ### Question 1: First, since $\mathcal I_{\phi}=\{i:|\phi^\top|_{i1}>0\}$ depends on the current assignment matrix $C$, the entropy objective appears to have a changing active set during optimization. Could the authors clarify how the MM/BSUM convergence argument handles the case where a supernode's labeled mass changes from zero to positive, or from positive to zero, during the iterations? Is the convergence claim intended to hold only on regions where the active set is fixed, or is there an argument that covers active-set changes?
> > > > >
> > > > > ### Answer 1:
> > > > > We thank the reviewer for this careful observation. We clarify that $\mathcal I_\phi$ depends on the current assignment matrix $C$ and is recomputed at every MM/BSUM iteration. The role of Eq. (11) is to construct a valid majorizing surrogate for the $C$-subproblem in Eq. (10). Therefore, the only requirement for the MM/BSUM argument is that, at iteration $t$, the surrogate satisfies the standard majorization properties:
> > > > > $
> > > > > g(C^{(t)}\mid C^{(t)})=f(C^{(t)}),
> > > > > \qquad
> > > > > g(C\mid C^{(t)})\ge f(C),\quad \forall C\in\mathcal S_C,
> > > > > $
> > > > > together with first-order consistency at $C^{(t)}$. These conditions are imposed on the relaxed objective being optimized, not on a fixed active-set representation.
> > > > >
> > > > > The notation
> > > > > $
> > > > > \mathcal I_\phi=\{i:\|[\phi^\top]_i\|_1>0\}
> > > > > $
> > > > > was introduced only to avoid division by zero when writing the normalized node-profile distribution. It should not be interpreted as defining a separate combinatorial active-set optimization problem.Rows with zero labeled mass are excluded from the entropy term and assigned zero entropy-gradient contribution. Equivalently, the entropy term can be written over all supernodes using the standard continuous convention $0\log 0=0$, with zero labeled-mass supernodes excluded only from normalization.
> > > > >
> > > > > Importantly, even though $\mathcal I_\phi^{(t)}$ may change, the $C$-update is always performed in the same ambient space $C\in\mathbb R_+^{p\times k}$. Thus, changing the active set does not change the dimension  of the $C$-subproblem. The entropy-gradient contribution is represented as a full $p\times k$ matrix: for active supernodes its column is computed normally, while for inactive supernodes it is set to zero.
> > > > >
> > > > > Therefore, if a supernode changes from zero labeled mass to positive labeled mass, it is added to $\mathcal I_\phi^{(t)}$ and its entropy-gradient contribution is computed in the next update. Conversely, if a supernode changes from positive labeled mass to zero labeled mass, it is removed from $\mathcal I_\phi^{(t)}$ and its entropy-gradient contribution becomes zero.
> > > > >
> > > > > In the extreme case where all supernodes have zero labeled mass, we have
> > > > > $
> > > > > \nabla_C r_E(C^{(t)},Y_{\rm train})=0,
> > > > > $
> > > > > and the $C$-update reduces to the corresponding base optimization-based graph coarsening update without the entropy term, following the Kumar et al. framework [1].
> > > > >
> > > > > Thus, even if a supernode changes from zero to positive labeled mass, or vice versa, during the relaxed updates, the MM step is still constructed with respect to the current relaxed objective, and the surrogate remains a valid upper bound for the objective at that iteration. The convergence proof therefore does not depend on a fixed active set.
> > > > >
> > > > > So, $I_\phi$ is only a notational device for defining normalized entropy safely, while the actual convergence/KKT claim applies to the relaxed continuous objective and the corresponding MM/BSUM surrogate.
> > > > >
> > > > >
> > > > >
> > > > > [1] Manoj Kumar, Anurag Sharma, Sandeep Kumar. A Unified Framework for Optimization-Based Graph Coarsening, JMLR, 2023

---

> > > > > ### Author Response · Authors · 2026-06-02
> > > > > **Response to Reviewer on Technical precision and Reproducibility (Part 2/2)**
> > > > >
> > > > > ### Question 2: The link-prediction protocol is clearer now, but I would appreciate a bit more detail on the construction of $E^C_{train}$ and $N^C_{train}$. In particular, when the coarsened adjacency is weighted, how are positive coarse edges defined? Are self-pairs removed? Are duplicate coarse pairs collapsed? How are negative coarse pairs sampled to avoid collisions with positive coarse edges? These details would help make the link-prediction experiments more reproducible.
> > > > > ### Answer 2:
> > > > > We thank the reviewer for this clarification request. We had addressed this point in our previous rebuttal while explaining the link-prediction protocol, but we agree that it is useful to restate the construction of the coarsened positive and negative training pairs more explicitly here for clarity and reproducibility.
> > > > >
> > > > > For each split, ENGC is applied only to the training graph. From the learned mapping matrix $C$, we form the coarse operator $\Theta_c=C^\top\Theta_{\mathrm{train}}C$ and obtain the coarsened training graph $\mathcal{G_c}=(V_c,E_c)$ . Since the coarsened adjacency is weighted, we use its support to define training pairs: a coarse pair $(a,b)$ is treated as a positive coarse edge if $a\neq b$ and the corresponding off-diagonal coarse adjacency entry is nonzero. Equivalently, in Laplacian form, we take
> > > > > $ E^c_{\mathrm{train}}=\{(a,b)\in V_c\times V_c:\ a\neq b,\ [\Theta_c]_{ab}\neq 0\}$.
> > > > > The link-prediction training label is binary: connected coarse pairs are positives and unconnected coarse pairs are negatives.
> > > > >
> > > > > We also clarify that duplicate coarse pairs are collapsed. That is, if multiple original training edges are aggregated into the same supernode pair, they produce a single coarse positive pair; their multiplicity only affects the corresponding coarse edge weight, not the number of positive training examples. Since the graphs are undirected, we use a canonical ordering of coarse pairs and remove duplicate pairs. Self-pairs $(a,a)$ are excluded from the link-prediction training set.
> > > > >
> > > > > The negative coarse training pairs are sampled after coarsening, not obtained by directly mapping the original negative set $N_{\mathrm{train}}$. Specifically, we sample them from the complement of $E^c_{\mathrm{train}}$ over valid supernode pairs:
> > > > > $N^c_{\mathrm{train}}\subseteq \{(a,b)\in V_c\times V_c:\ a\neq b,\ (a,b)\notin E^c_{\mathrm{train}}\}$.
> > > > > Sampling is performed without replacement, after removing self-pairs, duplicate pairs, and all pairs already present in $E^c_{\mathrm{train}}$, so negative pairs cannot collide with positive coarse edges. We match the number of negative coarse pairs to the number of positive coarse pairs. The GNN link predictor is then trained on $E^c_{\mathrm{train}}$ and $N^c_{\mathrm{train}}$. After training, the learned model parameters are used for inference on the original validation/test graphs, and final ROC-AUC is computed on the original positive and negative validation/test edge sets. Thus, coarsening is used only to reduce the training graph and the training pairs, while evaluation remains in the original node space.

---

> > > > > > ### Comment · Reviewer_Bjfh · 2026-06-02
> > > > > > **Comments**
> > > > > >
> > > > > > Thank you for the detailed clarifications.
> > > > > >
> > > > > > The additional explanation regarding the handling of active-set changes in the entropy objective clarifies my concern about the MM/BSUM convergence argument. The explanation that the optimization remains in the same ambient space and that inactive supernodes simply contribute zero entropy-gradient terms is helpful.
> > > > > >
> > > > > > The expanded description of the link-prediction protocol also improves reproducibility. In particular, the clarification of how coarse positive edges are defined, how duplicate pairs and self-pairs are handled, and how negative coarse pairs are sampled resolves my remaining questions about the experimental setup.
> > > > > >
> > > > > > I appreciate the authors’ efforts throughout the revision process. My concerns have been adequately addressed.

---

> ### Author Response · Authors · 2026-06-04
> **Response to the Reviewer’s Final Acknowledgment**
>
> We sincerely thank the reviewer for their continued constructive engagement and for carefully evaluating our revisions. We are grateful that the clarifications have addressed the remaining concerns. The reviewer’s detailed comments throughout the review period have significantly helped us improve the technical precision, presentation, and quality of the manuscript. We also deeply appreciate the reviewer’s timely replies, which enabled us to engage in a constructive discussion and update the manuscript in a timely manner.
>
> Since the reviewer’s concerns have now been adequately addressed, we would be grateful if the reviewer could consider updating their recommendation accordingly. We truly appreciate the reviewer’s time and thoughtful feedback throughout the revision process.

---

### Review · Reviewer_bT9w · 2026-04-23

**Summary Of Contributions:**

* Proposes ENGC, an entropy-guided semi-supervised graph coarsening framework
* Introduces entropy regularization to reduce label mixing in supernodes
* Formulates a unified objective combining structure, features, and labels
* Develops a BSUM/MM-based optimization algorithm with convergence guarantees
* Provides theoretical analysis on energy, cut, and resistance preservation
* Demonstrates competitive performance on node classification and link prediction

**Additional Comments:**

N/A

**Audience:**

Yes

**Audience Explanation:**

Yes, the paper would be of interest to researchers in graph learning and graph coarsening, particularly those working on semi-supervised learning and scalable GNNs.
However, the level of interest may be moderate given the incremental nature of the contribution.

**Broader Impact Concerns:**

No major ethical concerns identified

**Claims And Evidence:**

Yes

**Claims Explanation:**

* Theoretical analysis is clear and well-developed
* Empirical results support competitive performance
* However, gains over strong baselines are modest and sometimes inconsistent
* The link between theory and empirical improvements is not fully validated

**Requested Changes:**

Please address the following concerns:

* The novelty appears somewhat limited relative to prior work, such as LAGC (UAI 2024), as the main difference is the replacement of the label-aware regularizer with an entropy-based one.
* The empirical improvements over strong baselines are relatively modest and not always consistent across datasets and coarsening ratios.
    * Bonsai (ICLR 2025), Adapting Precomputed Features for Efficient GC (ICML 2025), FedGM (IJCAI 2025), ...
* The connection between the theoretical guarantees and the downstream empirical gains is not fully validated experimentally.
* The experimental evaluation on large-scale graphs and the evidence for scalability remain limited.

---

> ### Author Response · Authors · 2026-05-25
> **Response to Reviewer bT9w (part 1/3)**
>
> We thank the reviewer for their valuable feedback and have addressed their concerns regarding novelty, empirical gains, theoretical alignment, and scalability in detail below.
>
> ### Requested Changes 1: The novelty appears somewhat limited relative to prior work, such as LAGC (UAI 2024), as the main difference is the replacement of the label-aware regularizer with an entropy-based one.
>
> ### Response 1:
> We thank the reviewer for this important comment. We agree that the methodological contribution should be stated more explicitly. The key limitation addressed by ENGC is not simply that previous methods like LAGC use a Frobenius-type label regularizer, but that prior coarsening, condensation, and label-aware methods do not directly control *label mixing inside individual supernodes*.
>
> Structural graph coarsening methods mainly aim to preserve topology, spectral quantities, or feature smoothness. Graph condensation methods often optimize downstream training behavior, but usually require expensive training or bilevel/gradient-matching procedures. Existing label-aware coarsening methods use label information, but their objectives typically preserve aggregate label information rather than explicitly enforcing *supernode-level label homogeneity*. As a result, a supernode may still contain nodes from multiple classes even when the global label regularization penalty is small.
>
> This is precisely the failure mode ENGC is designed to address. In semi-supervised graph coarsening, the downstream model usually assigns one label to each supernode, often by majority vote. If a supernode contains nodes from several classes, then the assigned majority label becomes ambiguous, and minority-class nodes inside that supernode are effectively mislabeled. Therefore, the central issue is not merely whether labels are used during coarsening, but whether the learned supernodes are label-consistent.
>
> ENGC addresses this limitation by reformulating the node-profile matrix probabilistically. Given the node-profile matrix $\phi = C^\top Y$, we normalize each row to obtain
> $[\phi']_{i:}= \frac{\phi_i}{|[\phi^\top]_i|_1} \text{ if } |[\phi^\top]_i|_1 > 0$,
> so that each row of $\phi'$ represents the class distribution within a supernode. ENGC then minimizes the row-wise Shannon entropy $r_E(C,Y)$. Thus, the regularizer acts on the *uncertainty of each supernode's label distribution*, rather than only on raw label counts. Minimizing this entropy encourages each supernode's label distribution to become concentrated on one class, directly promoting label-homogeneous supernodes.
>
> This distinction is theoretically important. Theorem 2 shows that, under fixed row mass, entropy minimization favors sparse, nearly one-hot node-profile rows, while a Frobenius-type objective favors dense or uniform rows. Hence, the Frobenius-type regularizer does not necessarily promote same-label aggregation, whereas entropy minimization directly encourages label-consistent merging. This theoretical contrast explains why ENGC is not merely a substitution of one regularizer for another, but a change in the coarsening criterion itself.
>
> Furthermore, Lemma 2 connects entropy minimization to projection distortion. By reducing the entropy of the class distribution inside each supernode, ENGC reduces label impurity and tightens the projection-distortion term $\delta^\star$. This, in turn, leads to sharper Dirichlet-energy, cut-preservation, and effective-resistance guarantees. Theorem 5 further connects the same mechanism to learning by showing that entropy-controlled label homogeneity upper-bounds the majority-vote classification error within each supernode.
>
> We also validate this mechanism empirically. In addition to downstream node classification and link prediction, we evaluate supernode quality using SII, AMM, EIEC, $\delta^\star$, and $D_{\rm Dir}$. These metrics directly measure label impurity, majority-vote error, entropy-induced error ceiling, projection error and Dirichlet energy distortion. They therefore test the specific failure mode that ENGC is designed to resolve: whether coarsening produces label-mixed or label-consistent supernodes. Therefore, ENGC has methodological merit beyond a Frobenius-to-entropy replacement. Its contribution consists of:
> 1. Identifying label mixing within supernodes as a concrete limitation of prior methods;
> 2. Reformulating the node-profile matrix as a supernode-level label distribution;
> 3. Introducing an entropy objective that directly promotes label-homogeneous supernodes;
> 4. Proving that entropy gives stronger label-consistency behavior than a Frobenius-type objective; and
> 5. Deriving structural and learning guarantees that become tighter through entropy-controlled distortion.
>
> These are the key novelties that ENGC brings over prior methods: it directly optimizes supernode-level label purity, and we also validate this contribution through both theoretical analysis and empirical evidence.

---

> ### Author Response · Authors · 2026-05-25
> **Response to Reviewer bT9w (part 2/3)**
>
> ### Requested Changes 2: The empirical improvements over strong baselines are relatively modest and not always consistent across datasets and coarsening ratios.
> Bonsai (ICLR 2025), Adapting Precomputed Features for Efficient GC (ICML 2025), FedGM (IJCAI 2025), ...
>
> ### Response 2:
> We thank the reviewer for this careful assessment. We agree that the empirical gains over strong baselines are not uniformly large across all datasets and coarsening ratios, and we have revised the manuscript to avoid claiming that ENGC consistently dominates every baseline in every setting. Our intended claim is more precise: ENGC is competitive with recent graph coarsening/condensation methods and often improves performance when label mixing is important.
>
> To strengthen the comparison, we updated the related-work discussion and added new empirical comparisons with recent baselines, including Bonsai (ICLR 2025), GCPA (ICML 2025), FedGM (IJCAI 2025) as suggested by the reviewer. Beyond this we have also added MGC(WWW,2025) and SINGC(Neurips, 2025), although the latter were not explicitly mentioned by the reviewer. We report these results at matched coarsening ratios on Cora and CiteSeer. The updated table shows that ENGC achieves competitive or better accuracy in most matched settings, while also providing an explicit entropy-based mechanism for reducing label mixing.
>
> We also revised the experimental discussion to present the results more conservatively. In cases where improvements are modest or where protocols are not directly aligned, we now state this explicitly. Thus, the revised manuscript emphasizes that ENGC should be viewed not as uniformly dominating all recent baselines, but as a competitive and interpretable coarsening method whose entropy regularizer improves label purity and often improves downstream performance.
>
> **Table: Recent baseline comparison on Cora and CiteSeer. Higher accuracy is better.**
>
> | Dataset | r | MGC (WWW'25) | SINGC (NeurIPS'25) | Bonsai (ICLR'25) | FedGM (IJCAI'25) | GCPA (ICML'25) | ENGC (Ours ↑) |
> |---|---:|---:|---:|---:|---:|---:|---:|
> | CORA | 0.3 | 84.56 ± 1.40 | 84.51 ± 0.33 | 86.64 ± 1.18 | 82.19 ± 0.81 | 83.53 | **87.05 ± 0.05** |
> | CORA | 0.1 | 76.02 ± 0.93 | 82.76 ± 0.32 | 85.35 ± 1.91 | 77.18 ± 1.44 | 81.60 | **86.00 ± 1.05** |
> | CITESEER | 0.3 | 74.60 ± 2.31 | 76.66 ± 0.27 | 74.47 ± 0.40 | 73.19 ± 0.94 | 75.43 | **79.69 ± 0.37** |
> | CITESEER | 0.1 | 70.57 ± 1.25 | 69.71 ± 0.72 | 75.98 ± 0.52 | 70.95 ± 0.20 | 73.73 | **78.65 ± 1.95** |
>
> ### Requested Changes 3: The connection between the theoretical guarantees and the downstream empirical gains is not fully validated experimentally.
>
> ### Response 3:
>
> We thank the reviewer for pointing us in this direction. In the previous version, we had already introduced SII, AMM, and EIEC to evaluate supernode-level label mixing and majority-vote consistency. In the revised manuscript, we have further strengthened this theory-to-empirics connection by adding additional experiments in the Appendix comparing FGC, LAGC, and ENGC on the projection-distortion term $\delta^\star$ and Dirichlet-energy distortion $D_{\rm Dir}$. These additional metrics directly connect the theoretical guarantees to the observed downstream empirical gains by showing how entropy-controlled label purity leads to reduced projection error and smaller structural distortion. The comparison is shown below:
>
> **Table: Theory-to-empirics validation at $r=0.1$. Lower SII, AMM, EIEC, $\delta^\star$, and $D_{\rm Dir}$ indicate cleaner supernodes and better preservation of the quantities controlled by our theory. Metrics are computed post-hoc using all labels after the coarsening map has been learned.**
>
> | Dataset | Method | SII ↓ | AMM ↓ | EIEC ↓ | δ★ ↓ | D\_Dir ↓ | GCN Acc. ↑ |
> |---|---|---:|---:|---:|---:|---:|---:|
> | Cora | FGC | 1.8250 | 0.3773 | 0.4546 | 0.9226 | 0.3762 | 79.96 ± 0.18 |
> | Cora | LAGC | 1.2818 | 0.4002 | 0.4407 | 0.6155 | 0.1434 | 86.10 ± 0.03 |
> | Cora | ENGC | **0.3193** | **0.1242** | **0.4111** | **0.4354** | **0.1408** | 86.00 ± 1.05 |
> | CiteSeer | FGC | 0.8078 | 0.4144 | 0.3604 | 0.9268 | 0.7730 | 69.46 ± 0.22 |
> | CiteSeer | LAGC | 0.2080 | 0.1036 | 0.2023 | 0.8919 | 0.4849 | 76.00 ± 0.50 |
> | CiteSeer | ENGC | **0.1360** | **0.0538** | **0.0573** | **0.5599** | **0.1410** | **78.65 ± 1.95** |
>
> Above table provides a direct validation of the proposed theory-to-empirical validation. FGC does not use label information during coarsening and therefore produces comparatively mixed supernodes, as reflected by larger SII, AMM,$\delta^\star$, and $D_{\rm Dir}$. LAGC improves over FGC by incorporating label information through a Frobenius-type regularizer, but it still does not directly penalize the uncertainty of the row-wise label distribution. In contrast, ENGC explicitly minimizes the entropy of the node-profile distribution, which leads to substantially lower label impurity and majority-vote error.

---

> ### Author Response · Authors · 2026-05-25
> **Response to Reviewer bT9w (part 3/3)**
>
> ###  Requested Changes4: The experimental evaluation on large-scale graphs and the evidence for scalability remain limited.
>
> ### Response 4:
> We agree that the original evaluation was focused mainly on standard citation/co-authorship benchmarks. To address this, we added large-scale experiments on **Flickr**, **OGBN-Arxiv**, and **OGBN-Products**, using matched coarsening/condensation ratios and comparing against recent scalable baselines such as **Bonsai** and **GCPA**. We put the result below together with the number of train/validation/test nodes, feature dimensions, and class counts, to make the scalability setting explicit.
>
> We also clarified that our scalability claim is not that ENGC is uniformly superior on all large graphs, but that it remains applicable beyond small citation datasets and achieves competitive performance while reducing the graph size used for downstream training. In addition, the runtime-complexity discussion explains why training on the coarsened graph is computationally cheaper than training directly on the full graph. Thus, the revised manuscript provides stronger empirical evidence for scalability while avoiding an overclaim.
>
> **Table: Baseline comparison on Flickr, OGBN-Arxiv, and OGBN-Products.**
>
> | Dataset | r | Train/Val/Test Nodes | Features | Classes | Bonsai | GCPA | ENGC (Ours) |
> |---|---:|---:|---:|---:|---:|---:|---:|
> | Flickr | 0.01 | 44,625/22,312/22,313 | 500 | 7 | 48.36 ± 0.36 | 47.2 ± 0.3 | **48.74 ± 0.29** |
> | OGBN-Arxiv | 0.01 | 90,941/29,799/48,603 | 128 | 40 | 58.59 ± 0.10 | **64.09 ± 0.2** | 62.36 ± 0.07 |
> | OGBN-Products | 0.001 | 196,615/39,323/2,213,091 | 100 | 47 | 48.36 ± 0.36 | 68.1 ± 0.2 | **69.60 ± 0.01** |

---

### Review · Reviewer_EXVK · 2026-05-08

**Summary Of Contributions:**

The paper proposes an entropy-guided semi-supervised graph coarsening method that incorporates graph structure, node features, and partial label information. The central idea is to regularize the normalized node-profile matrix using row-wise Shannon entropy, with the goal of producing supernodes that are more label-homogeneous. The paper also presents an MM/BSUM-style optimization algorithm, theoretical guarantees related to label impurity, Dirichlet energy, cut preservation, and effective resistance, and experiments on node classification and link prediction benchmarks.

**Audience:**

Yes

**Audience Explanation:**

The paper addresses a real problem about how to construct a smaller graph that preserves structural, feature, and label information for downstream learning. This is a meaningful direction.

**Claims And Evidence:**

No

**Claims Explanation:**

The paper addresses a real problem: how to construct a smaller graph that preserves structural, feature, and label information for downstream learning. This is a meaningful direction. However, the paper does not clearly articulate the precise limitation of prior methods that the proposed entropy regularizer resolves. At several points, the contribution reads mainly as replacing a Frobenius-type label regularizer with an entropy regularizer. While natural, this alone does not yet establish a strong methodological advance unless the paper more clearly explains why existing coarsening, condensation, or label-aware methods fail.

Several theoretical claims require more careful treatment. The feasible set for the coarsening matrix includes non-overlap and assignment-type constraints, but the proposed update appears to enforce mainly nonnegativity or a normalized positive projection. It is not clear that the algorithm actually preserves the stated feasible set.

The paper contains several noticeable formatting problems. For example, the last paragraph before Section 1.1 is not properly spaced, the title of Section 2 is fully capitalized and inconsistent with the rest of the paper, and the inline equations above Eq. (2) and Eq. (13) are poorly spaced. There are also multiple grammatical issues, informal phrases, inconsistent notation, and unclear definitions. These issues substantially weaken readability.

My evaluation leans towards being negative. Though the paper is partly outside my expertise and I apologize for my limited insights. The high-level question of label-aware graph coarsening is meaningful, and the entropy regularizer is intuitive. However, I found the paper insufficiently convincing in its current form.

**Requested Changes:**

See above.

---

> ### Author Response · Authors · 2026-05-24
> **Response to Reviewer EXVK (part 1/3)**
>
> We thank the reviewer for their valuable and constructive critique regarding our methodology, theoretical clarity, and presentation, which has helped us significantly strengthen the manuscript.
>
> ### Question 1: The paper addresses a real problem: how to construct a smaller graph that preserves structural, feature, and label information for downstream learning. This is a meaningful direction. However, the paper does not clearly articulate the precise limitation of prior methods that the proposed entropy regularizer resolves. At several points, the contribution reads mainly as replacing a Frobenius-type label regularizer with an entropy regularizer. While natural, this alone does not yet establish a strong methodological advance unless the paper more clearly explains why existing coarsening, condensation, or label-aware methods fail.
>
> ### Answer 1:
> We thank the reviewer for this important comment. We agree that the methodological contribution should be stated more explicitly. The key limitation addressed by ENGC is not simply that previous methods use a Frobenius-type label regularizer, but that prior coarsening, condensation, and label-aware methods do not directly control *label mixing inside individual supernodes*.
>
> Structural graph coarsening methods mainly aim to preserve topology, spectral quantities, or feature smoothness. Graph condensation methods often optimize downstream training behavior, but usually require expensive training or bilevel/gradient-matching procedures. Existing label-aware coarsening methods use label information, but their objectives typically preserve aggregate label information rather than explicitly enforcing *supernode-level label homogeneity*. As a result, a supernode may still contain nodes from multiple classes even when the global label regularization penalty is small.
>
> This is precisely the failure mode ENGC is designed to address. In semi-supervised graph coarsening, the downstream model usually assigns one label to each supernode, often by majority vote. If a supernode contains nodes from several classes, then the assigned majority label becomes ambiguous, and minority-class nodes inside that supernode are effectively mislabeled. Therefore, the central issue is not merely whether labels are used during coarsening, but whether the learned supernodes are label-consistent.
>
> ENGC addresses this limitation by reformulating the node-profile matrix probabilistically. Given the node-profile matrix $\phi = C^\top Y$, we normalize each row to obtain  $[\phi']_{i:}= \frac{\phi_i}{\\|[\\phi^\\top]_i\\|_1} \text{if } \\|[\\phi^\\top]_i\\|_1 > 0$,     so that each row of $\phi'$ represents the class distribution within a supernode. ENGC then minimizes the row-wise Shannon entropy $r_E(C,Y)$. Thus, the regularizer acts on the *uncertainty of each supernode's label distribution*, rather than only on raw label counts. Minimizing this entropy encourages each supernode's label distribution to become concentrated on one class, directly promoting label-homogeneous supernodes.
>
> This distinction is theoretically important. Theorem 2 shows that, under fixed row mass, entropy minimization favors sparse, nearly one-hot node-profile rows, while a Frobenius-type objective favors dense or uniform rows. Hence, the Frobenius-type regularizer does not necessarily promote same-label aggregation, whereas entropy minimization directly encourages label-consistent merging. This theoretical contrast explains why ENGC is not merely a substitution of one regularizer for another, but a change in the coarsening criterion itself.
>
> Furthermore, Lemma 2 connects entropy minimization to projection distortion. By reducing the entropy of the class distribution inside each supernode, ENGC reduces label impurity and tightens the projection-distortion term $\delta^\star$. This, in turn, leads to sharper Dirichlet-energy, cut-preservation, and effective-resistance guarantees. Theorem 5 further connects the same mechanism to learning by showing that entropy-controlled label homogeneity upper-bounds the majority-vote classification error within each supernode.
>
> We also validate this mechanism empirically. In addition to downstream node classification and link prediction, we evaluate supernode quality using SII, AMM, EIEC, $\delta^\star$, and $D_{\rm Dir}$. These metrics directly measure label impurity, majority-vote error, entropy-induced error ceiling, projection error and Dirichlet energy distortion. They therefore test the specific failure mode that ENGC is designed to resolve: whether coarsening produces label-mixed or label-consistent supernodes.

---

> ### Author Response · Authors · 2026-05-24
> **Response to Reviewer EXVK (part 2/3)**
>
> Therefore, ENGC has methodological merit beyond a Frobenius-to-entropy replacement. Its contribution consists of:
>
> 1. identifying label mixing within supernodes as a concrete limitation of prior methods;
> 2. reformulating the node-profile matrix as a supernode-level label distribution;
> 3. introducing an entropy objective that directly promotes label-homogeneous supernodes;
> 4. proving that entropy gives stronger label-consistency behavior than a Frobenius-type objective; and
> 5. deriving structural and learning guarantees that become tighter through entropy-controlled distortion.
>
> We have revised the manuscript to make this distinction clearer in the introduction and method sections, and to explicitly state that ENGC's novelty lies in directly optimizing supernode-level label purity rather than only preserving aggregate label information.
>
>
> ### Question 2: Several theoretical claims require more careful treatment. The feasible set for the coarsening matrix includes non-overlap and assignment-type constraints, but the proposed update appears to enforce mainly nonnegativity or a normalized positive projection. It is not clear that the algorithm actually preserves the stated feasible set.
>
> ### Answer 2:
> We thank the reviewer for pointing out this important issue. We agree that the original presentation did not sufficiently distinguish the ideal hard assignment constraints from the relaxed optimization problem actually solved by ENGC.
>
> In the revised manuscript, we now clarify that the hard assignment set $\mathcal{C}$ is used to describe the desired final non-overlapping coarsening map, where each original node is assigned to one supernode. However, directly optimizing over this assignment-type feasible set is combinatorial. Therefore, ENGC does not claim to preserve the hard assignment constraints at every MM/BSUM iteration. Instead, the algorithm solves a continuous relaxation over a nonnegative soft-assignment matrix $C$.
>
> In particular, we now explicitly distinguish the hard assignment set $\mathcal{C}$ in Eq. (2), which describes the desired final coarsening structure, from the relaxed feasible set $\mathcal{S}_c$ over which ENGC is optimized. The hard assignment set is used only to define the ideal non-overlapping coarsening map. Since directly optimizing over this set is combinatorial, ENGC instead solves a continuous relaxation in which $C$ is a nonnegative soft assignment matrix.
>
> We have revised the formulation and updated the manuscript so that the subproblem, update rule, and convergence statement are all stated over the same relaxed feasible set $\mathcal{S}_c$ . We have also clarified that Eq. (13) is a projected first-order MM/BSUM update for the relaxed problem, not an exact optimizer over the hard assignment set. After convergence, the learned soft assignment matrix is converted into a hard coarsening map by assigning each node to the supernode with the largest entry in its row.
>
> We have also updated the appendix so that the KKT conditions and convergence proof match the relaxed problem stated in the main text. Any auxiliary assumptions used in the proof, such as boundedness and Lipschitz continuity of the gradient, are now stated explicitly rather than appearing as additional implicit constraints.
>
> Finally, we have reorganized the theorem and lemma structure to clarify the role of each result. Lemma 1 and Theorem 1 now support the relaxed MM/BSUM solver; Theorem 2, Lemma 2, and Theorem 5 provide the entropy-specific analysis; and Theorem 3, Corollary 4, and Lemma 3 provide the structural preservation guarantees whose tightness improves when entropy reduces the projection distortion $\delta^\star$. This revision makes clear which result supports each claimed contribution and aligns the formulation, constraints, update rule, and convergence proof.

---

> ### Author Response · Authors · 2026-05-24
> **Response to Reviewer EXVK (part 3/3)**
>
> ### Question 3: The paper contains several noticeable formatting problems. For example, the last paragraph before Section 1.1 is not properly spaced, the title of Section 2 is fully capitalized and inconsistent with the rest of the paper, and the inline equations above Eq. (2) and Eq. (13) are poorly spaced. There are also multiple grammatical issues, informal phrases, inconsistent notation, and unclear definitions. These issues substantially weaken readability.
>
> ### Answer 3:
>
> Thank you for your careful review. We have addressed the spacing issues throughout the manuscript and conducted a thorough review of the notation and readability to ensure consistency. Furthermore, we have corrected various grammatical and typographical errors. These modifications are summarized below.
>
> ### Grammatical issues.
>
> | Location | Current text | Revised text |
> |---|---|---|
> | Sec. 2 | BACKGROUND AND PROBLEM FORMULATION | Background and Problem Formulation |
> | Fig. 1 caption | Example of two coarsening | Example of two coarsenings |
> | Sec. 3 | we delve into the algorithmic development | we present the algorithmic development |
> | Sec. 3.1 | Where, the first term... | The first term... |
> | Sec. 3.1 | each element equals $1/k$ | each element equal $1/k$ |
> | Sec. 3.1 | at each itteration | at each iteration |
> | Sec. 4.2 | In this subsection We quantify... | In this subsection, we quantify... |
> | Table 1 caption | Compairing the results... | Comparing the results... |
> | Appendix A.18 | Reffering to table 3 | Referring to Table 3 |
>
> ### Informal phrases.
>
> | Location | Current phrase | Revised phrase |
> |---|---|---|
> | Sec. 4.6 | EIEC turns entropy into... | EIEC converts the entropy value into... |
> | Sec. 5 | We further test the usefulness... | We further evaluate ENGC on... |
> | Sec. 4.6 | SII measures how mixed the labels are | SII quantifies the degree of label mixing within supernodes |
> | Sec. 5.2 | basically a majority vote | i.e., by majority vote |
> | Sec. 4.2 | So, $\delta^\star$ tells us... | Thus, $\delta^\star$ quantifies... |
>
> ### Inconsistent notation.
>
> | Issue addressed | Location | Revision made |
> |---|---|---|
> | Coarse feature notation was inconsistent | Eq. 15 and Algorithm 2 | Unified the coarse feature notation and use $X_c=\widetilde X$ consistently. We also changed Algorithm 2. |
> | Coarse label notation was inconsistent | Sec. 2.2, Fig. 2, Algorithm 2 | Unified $Y_c$, $\widetilde Y$, and $Y'$ into a single coarse-label notation $Y_c$. |
> | $\mathcal{G}_c$ notation was inconsistent | Sec. 2.2, Sec. 5.2, Appendix 11 | Unified $G_c$ and $\mathcal{G}_c$ into $\mathcal{G}_c$. |
> | Mapping notation was inconsistent | Algorithm 2 | Removed the use of $P^\dagger$ and use $C^\top$ directly for aggregation to make the algorithm clearer. |
> | Lifted Laplacian notation was unclear | Sec. 2.1 and Theorem 3 | Defined the lifted Laplacian explicitly as $\widetilde{\Theta}=C\Theta_c C^\top$. |
> | Entropy log base was inconsistent | Eq. 6, Eq. 7, Theorem 5 | Made the logarithm convention consistent across the entropy regularizer and entropy-based bounds. |
>
> ### Unclear definitions.
>
> | Definition | Problem | Revision |
> |---|---|---|
> | $k'$ in SII/AMM/EIEC | $k'$ was not explicitly defined. | We clarified that $k'=k$, where $k$ is the number of learned supernodes. Thus, SII, AMM, and EIEC are averaged over all $k$ supernodes. |
> | Sec. 2.2: $\langle c_i,c_i\rangle=d_i$ | $d_i$ was not defined. | We clarified that $d_i$ is a positive normalization constant. |
>
>
> These revisions make the presentation more consistent and easier to follow without changing the core method or results.

---

> > ### Comment · Reviewer_EXVK · 2026-06-01
> > **Response to the revised manuscript**
> >
> > Thanks for the authors' efforts in revising the paper that directly addressed my previous concerns. The paper now gives a more direct motivation for using entropy regularization, and the distinction from Frobenius-type label regularization is easier to understand. The clarification that only training labels are used in the semi-supervised setting is also important and improves the empirical credibility of the paper.
> >
> > However, the optimization formulation and algorithmic analysis are still not fully clear. The paper originally describes hard assignment/non-overlap constraints for the coarsening matrix, but the actual optimization uses a relaxed nonnegative feasible set. The authors now acknowledge this as a continuous relaxation, which is helpful. Nevertheless, the proposed update for (C) still appears to be mainly a positive projection/normalization step, and it is not clear that it solves the stated constrained surrogate problem or preserves all constraints in the relaxed feasible set. The convergence/KKT argument therefore remains insufficiently convincing.
> >
> > I am also still confused about the theory: the effective-resistance proof treats the lifted Laplacian as invertible on the same zero-sum subspace as the original Laplacian, which is not generally justified when $k \ll p$. The entropy-to-majority-error bound uses a binary entropy inverse in a multiclass setting without the needed conditions.
> >
> > I would therefore like to reserve my original evaluation.

---

> > > ### Author Response · Authors · 2026-06-03
> > > **Response to Reviewer’s Follow-up Questions (Part 1/3)**
> > >
> > > We sincerely thank the reviewer for their thoughtful and constructive feedback. These comments have helped us improve the mathematical precision and presentation of the paper, and we address the remaining concerns below.
> > >
> > > ### Reviewer Concern 1: Nevertheless, the proposed update for (C) still appears to be mainly a positive projection/normalization step, and it is not clear that it solves the stated constrained surrogate problem or preserves all constraints in the relaxed feasible set. The convergence/KKT argument therefore remains insufficiently convincing.
> > >
> > > ### Response 1:
> > > We thank the reviewer for pointing out this issue. We first clarify that the optimization problem solved in the paper is the relaxed problem in Eq.(9), not the ideal hard assignment problem over $\mathcal C$. The hard assignment formulation describes the desired non-overlapping coarsening map in the ideal case, whereas the actual optimization is carried out over the relaxed feasible set $\mathcal S_c$. This relaxation is meaningful because it allows a continuous nonnegative assignment matrix during optimization, while the final hard coarsening map can be obtained after convergence. The KKT/convergence guarantee is therefore stated for the relaxed problem in Eq.(9), not for the hard combinatorial assignment problem.
> > >
> > > The relaxed problem is solved using a block successive upper-bound minimization (BSUM/MM) strategy. In each iteration, the variables are updated alternately, and for the $C$-subproblem, the objective is majorized at the current iterate $C^{(t)}$. The key requirement for the BSUM/MM convergence argument is that the constructed surrogate satisfies the standard majorization and first-order consistency conditions for the relaxed objective. This is the setting covered by the MM/BSUM framework of Razaviyayn et al. [1] and Sun et al. [2].
> > >
> > > We also clarify that the projection/normalization form of the update does not mean that the terms in the objective are ignored. Each term contributes to the $C$-update through the full gradient $\nabla f(C^{(t)})$. The entropy term reduces label uncertainty and label mixing inside supernodes. The log-determinant term $-\log\det(\Theta_c+J)$ promotes a connected coarsened graph. The feature-reconstruction term preserves the original node features in the learned coarse representation. The Dirichlet-energy term preserves graph-smoothness and structural consistency. The structural regularization term controls the learned coarse operator, and the $\ell_{1,2}$-type regularizer encourages sparse, assignment-like coarsening structure. Therefore, the projection/normalization step is not replacing these objective terms; it is the closed-form solution of the constrained surrogate after all these terms have contributed through $\nabla f(C^{(t)})$.
> > >
> > > This point is supported by Lemma 1 and Theorem 1 in the manuscript. Lemma 1 derives the $C$-update from the KKT optimality conditions of the constrained majorized surrogate, showing that the update is the optimizer of the relaxed surrogate rather than a heuristic projection. Theorem 1 then states that the sequence ${C^{(t)},\widetilde X^{(t)}}$ generated by Algorithm 1 converges to the set of KKT points of the relaxed problem in Eq.(9). Therefore, the convergence/KKT guarantee is explicitly for the relaxed MM/BSUM problem over $\mathcal S_c$, not for the hard assignment set $\mathcal C$.
> > >
> > > In particular, after majorizing the $C$-subproblem at $C^{(t)}$, the constrained surrogate takes the form
> > > $\min_{C\in\mathcal S_c} \frac12 C^TC-C^TA$,  where, $A=C^{(t)}-\frac{1}{L}\nabla f(C^{(t)})$.  This gives a valid $C$-update for the relaxed feasible set $\mathcal S_c$, where $C$ is interpreted as a continuous nonnegative coarsening matrix approximating the ideal hard assignment. Similar relaxed assignment matrices and constrained quadratic surrogate updates have also been adopted in optimization-based graph coarsening methods, including Kumar et al. [3].
> > >
> > > We therefore do not claim that this update preserves the hard non-overlap constraints in $\mathcal C$. The hard assignment set $\mathcal C$ is used only to describe the desired final coarsening map, while the optimization, surrogate construction, and KKT/convergence analysis are all for the relaxed problem over $\mathcal S_c$. We have revised the manuscript to make this distinction explicit and to clarify that the convergence/KKT guarantee applies to the relaxed BSUM/MM problem in Eq.(9).
> > >
> > > [1] Meisam Razaviyayn, Mingyi Hong, and Zhi-Quan Luo. A unified convergence analysis of block successive minimization methods for nonsmooth optimization. SIAM Journal on Optimization, 2013.
> > >
> > > [2] Ying Sun, Prabhu Babu, and Daniel P. Palomar. Majorization-minimization algorithms in signal processing, communications, and machine learning. IEEE Transactions on Signal Processing, 2016.
> > >
> > > [3] Manoj Kumar, Anurag Sharma, and Sandeep Kumar. A Unified Framework for Optimization-Based Graph Coarsening. JMLR, 2023.

---

> > > ### Author Response · Authors · 2026-06-03
> > > **Response to Reviewer’s Follow-up Questions (Part 2/3)**
> > >
> > > ### Reviewer Concern 2: I am also still confused about the theory: the effective-resistance proof treats the lifted Laplacian as invertible on the same zero-sum subspace as the original Laplacian, which is not generally justified when $k\ll p$.
> > >
> > > ### Response 2:
> > >
> > > We thank the reviewer for pointing out this subtle but important issue. We first clarify that our effective-resistance argument was written under the intended connected-Laplacian regime induced by the coarsening objective. In particular, the log-determinant term $-\log\det(C^T \Theta C+J)$ is included to promote a connected coarse Laplacian $\Theta_c$, since it penalizes singular or nearly disconnected coarse graph structures. Thus, in the previous version, we implicitly treated the lifted coarse Laplacian $\widetilde{\Theta}=C\Theta_cC^T$ as a structurally valid connected Laplacian approximation of the original graph. Under this intended setting, the comparison is made on the common zero-sum subspace, i.e., $\mathrm{null}(\Theta)=\mathrm{null}(\widetilde{\Theta})=\mathrm{span}{\mathbf{1}_p}$. However, we agree with the reviewer that this condition should not have been left implicit. For completeness and mathematical precision, we will explicitly include this common-nullspace condition in the theorem statement.
> > >
> > > More specifically, for a connected graph Laplacian $\Theta$, effective resistance is defined through $\Theta^\dagger$ on the zero-sum subspace $\mathbf{1}_p^\perp$. Therefore, to compare $\Theta^\dagger$ and $\widetilde{\Theta}^\dagger$, the lifted Laplacian $\widetilde{\Theta}$ must act invertibly on the same subspace. This requires $\mathrm{null}(\widetilde{\Theta})=\mathrm{span}{\mathbf{1}_p}$, or equivalently $\lambda_2(\widetilde{\Theta})>0$. Without this condition, $\widetilde{\Theta}$ may have an enlarged nullspace when $k\ll p$, and the standard pseudoinverse perturbation argument cannot be applied directly.
> > >
> > > We therefore revise the effective-resistance guarantee as a conditional perturbation result. Let $\lambda_{\rm gap}:=\lambda_2(\Theta)>0$ and $\Delta_{\rm err}:=||\Theta-\tilde{\Theta}||2$. Assume that $null(\tilde{\Theta})=span{\mathbf{1}p}$ and $\Delta_{\rm err}<\lambda_{\rm gap}$. Under this common-nullspace assumption, both $\Theta$ and $\tilde{\Theta}$ are invertible on $\mathbf{1}p^\perp$, and standard pseudoinverse perturbation arguments give $||\Theta^\dagger-\tilde{\Theta}^\dagger||2 \le \frac{\Delta{\rm err}}{\lambda{\rm gap}(\lambda_{\rm gap}-\Delta_{\rm err})}$.
> > >
> > > Hence, for any pair of nodes $u,v$, $|R_{uv}-\widetilde R_{uv}|=\left|(e_u-e_v)^T(\Theta^\dagger-\widetilde{\Theta}^\dagger)(e_u-e_v)\right|\le \frac{2\Delta_{\mathrm{err}}}{\lambda_{\mathrm{gap}}(\lambda_{\mathrm{gap}}-\Delta_{\mathrm{err}})}$. If $\Delta_{\mathrm{err}}\le \lambda_{\mathrm{gap}}/2$, then $|R_{uv}-\widetilde R_{uv}|\le \frac{2\Delta_{\mathrm{err}}}{\lambda_{\mathrm{gap}}^2}$.
> > >
> > > Thus, the revised theorem no longer claims effective-resistance preservation for arbitrary low-rank lifted Laplacians. Instead, it states the result under the common-nullspace/spectral condition required by the Laplacian pseudoinverse framework. This keeps the intended message of the result while making the mathematical scope of the guarantee precise.

---

> > > ### Author Response · Authors · 2026-06-03
> > > **Response to Reviewer’s Follow-up Questions (Part 3/3)**
> > >
> > > ### Reviewer Concern 3: The entropy-to-majority-error bound uses a binary entropy inverse in a multiclass setting without the needed conditions.
> > >
> > > ### Response 3:
> > > We thank the reviewer for pointing out this subtle but important issue regarding the entropy-to-majority-error relation. In the previous version, our intention was to provide an intuitive information-theoretic connection between low entropy in the node-profile distribution and low majority-vote error within each supernode. The expression $e_i \le h_2^{-1}(H(p_i))$ is valid in the binary-label setting, or when the supernode label distribution is explicitly reduced to a binary majority-versus-rest distribution. However, we agree that this binary inverse should not be used as the general multiclass statement without the required conditions.
> > >
> > > We have revised the manuscript to make this distinction explicit. To keep the main presentation concise and intuitive, we have retained the binary-form expression only as an interpretive binary special case, and we provide the multiclass version of the entropy-to-majority-error relation as a lemma in Appendix A.7
> > >
> > > Specifically, for a multiclass supernode distribution $p_i=(p_i(1),\ldots,p_i(\ell))$, let $m_i=\max_j p_i(j)$ denote the majority-label mass and let $e_i=1-m_i$ denote the corresponding majority-vote error. Since Shannon entropy dominates min-entropy, we have $H(p_i)\ge -\log_2 m_i$. Therefore, $m_i\ge 2^{-H(p_i)}$, and hence $e_i=1-m_i\le 1-2^{-H(p_i)}$. Thus, in the multiclass setting, the size-weighted majority-vote error satisfies $\sum_i |S_i|e_i \le \sum_i |S_i|(1-2^{-H(p_i)})$. The appendix includes this derivation which clarify the binary inverse expression $e_i \le h_2^{-1}(H(p_i))$. For the general multiclass case, the theorem uses the bound $e_i\le 1-2^{-H(p_i)}$.
> > >
> > > This correction does not affect the proposed optimization framework, entropy regularizer, or empirical conclusions. It only refines the theoretical presentation by replacing the binary-specific expression with a multiclass-valid entropy-error relation. The central conclusion remains unchanged: minimizing the entropy of the node-profile distribution reduces label mixing within supernodes and controls majority-label inconsistency after coarsening.

---

### Decision · Action_Editor_wUCG · 2026-06-20

**Recommendation:** Accept with minor revision

**Additional Comments:**

The submission is borderline but ultimately positive: two of three substantive reviewers lean Accept, and the extended discussion resolved the major correctness and clarity concerns. I recommend acceptance conditioned on the authors fully incorporating into the camera-ready the fixes already committed during rebuttal, specifically:

1. Ensure the corrected effective-resistance theorem carries the explicit common-nullspace / spectral-gap condition (null(Θ̃)=span{1ₚ}, Δ_err<λ_gap) in its statement, not only in the rebuttal text.

2. Confine the binary entropy-inverse expression to the binary case throughout, with the multiclass bound (eᵢ ≤ 1 − 2^(−H(pᵢ))) used wherever a general claim is made.

3. Make the zero-labeled-mass convention consistent in the entropy objective, gradient, and Lipschitz/convergence statements (restricting normalization to Iϕ), and ensure Eq. (10)/(12) and the C-subproblem use the relaxed feasible set Sc consistently.

4. Integrate the expanded ablations (multiple datasets/ratios), the added recent baselines (Bonsai, GCPA, FedGM, MGC, SINGC) and large-scale results (Flickr, OGBN-Arxiv/Products), Table 2 uncertainty estimates, the corrected Table 1 entries, and the softened runtime claims.

5. State the empirical claim conservatively—ENGC as competitive and interpretable rather than uniformly dominant—and retain the explicit separation between entropy-specific and general perturbation guarantees.

6. Complete the remaining presentation/notation cleanup (Algorithm 2/3, lifted-Laplacian notation, log-base consistency) noted by reviewers.

No certification is recommended; while the work is sound and useful, its incremental novelty and modest empirical margins do not rise to the level of a notable certification.

**Audience:**

Yes

**Audience Explanation:**

All reviewers agreed on this point. Graph coarsening for scalable GNN training is an active topic, and an interpretable, optimization-based regularizer that directly targets supernode label purity—together with the impurity-oriented diagnostic metrics (SII, AMM, EIEC)—is of clear interest to researchers working on graph learning, graph reduction, and semi-supervised GNNs.

**Claims And Evidence:**

Yes

**Claims Explanation:**

The paper proposes ENGC, a semi-supervised graph coarsening framework that regularizes the row-wise Shannon entropy of the normalized node-profile matrix to directly penalize label mixing within supernodes, distinguishing it from prior Frobenius-type label-aware methods (notably LAGC). The central claims rest on three pillars: (i) a theoretical contrast showing entropy minimization drives node-profile rows toward one-hot distributions while Frobenius regularization favors uniform rows (Theorem 2); (ii) a chain of guarantees linking entropy-controlled label impurity to projection distortion, Dirichlet-energy, cut, and effective-resistance preservation, plus a majority-vote error bound (Theorem 5); and (iii) empirical validation via downstream node classification, link prediction, and dedicated label-mixing metrics (SII, AMM, EIEC) and theory-to-empirics tables.

Across the review period the authors materially strengthened the evidence. The entropy-specific results (Theorem 2, Lemma 2, Theorem 5) are now clearly separated from the more generic perturbation-style structural bounds, reducing earlier overclaiming. Two genuine technical errors flagged by reviewers were corrected: the effective-resistance result was restated as a conditional perturbation bound under an explicit common-nullspace assumption (addressing the invalid treatment of the lifted Laplacian when k≪p), and the binary entropy-to-error inverse was confined to the binary case with a multiclass-valid bound moved to the appendix. The semi-supervised treatment of zero-labeled-mass supernodes, the relaxation of the hard assignment set to a continuous feasible set, the MM/BSUM active-set behavior, and the link-prediction pair construction were all clarified, and Reviewer Bjfh explicitly stated their concerns were adequately addressed. The theory-to-empirics validation (Table 7) provides reasonable, if post-hoc, support that entropy regularization lowers SII/AMM/δ*/Dirichlet distortion alongside competitive accuracy.

The principal residual reservation, shared across reviewers, is that the empirical gains over strong baselines are modest and inconsistent across datasets and coarsening ratios, and the contribution is somewhat incremental relative to LAGC. Reviewer EXVK maintained a "No" on claims/evidence, citing lingering doubts about whether the C-update solves the stated surrogate. However, the authors' clarification that the update is the closed-form KKT optimizer of the relaxed majorized subproblem (not a heuristic projection) is technically reasonable, and this concern was the minority position. On balance the soundness bar is met: claims are appropriately scoped, the corrected theory is valid within its stated conditions, and the empirical evidence supports the (now more conservative) claims.

---

> ### Author Response · Authors · 2026-07-10
> **Response to Action Editor wUCG**
>
> We thank the AE and all reviewers for the time and effort they put into evaluating our submission. We greatly appreciate your helpful suggestions, which have made the paper more sound and clear. We appreciate the opportunity to finalize the manuscript, and we have carefully incorporated the requested changes into the revised version.
>
> First, we have revised the effective-resistance result so that the Lemma 3 statement explicitly includes the required common-nullspace and spectral-gap assumptions. In particular, the statement now assumes $\operatorname{null}(\widetilde{\Theta})=\operatorname{span}{\mathbf{1}p},
> \quad \Delta{\mathrm{err}}<\lambda_{\mathrm{gap}}$,
> before deriving the pseudoinverse perturbation bound. This makes the condition explicit in the lemma itself.
>
> Second, For the entropy-inverse condition, we have clarified the scope of the result. In the revised manuscript, the binary entropy-inverse expression $h_2^{-1}(\cdot)$ is used only when the discussion is explicitly restricted to the binary case or to the binary majority-vs-nonmajority reduction. For the general multiclass setting, we state and use the multiclass bound
> $e_i \le 1-2^{-H(p_i)}$ and done the chnages in the statement of Theorem 5.
> The proof of this multiclass bound is provided in the appendix A.8, and the main text now refers to this bound whenever a general multiclass claim is made. Thus, the binary entropy-inverse expression is retained only as a special case, while the general statement is consistently handled by the multiclass entropy bound.
>
> Third, we have made the zero-labeled-mass convention consistent across the objective, gradient, and convergence/Lipschitz discussion. We define $I_\phi$ as the set of supernodes with positive labeled mass and restrict the row normalization and entropy objective to $i\in I_\phi$. For $i\notin I_\phi$, the entropy-gradient contribution is set to zero. Thus, the entropy term in Eq.(10) sum only over the set of supernodes with positive labeled mass, where the normalized row is well-defined and . Supernodes with zero labeled mass are excluded from the entropy objective. We have also ensured that the $C$-subproblem and the majorized update consistently use the relaxed feasible set $S_c$.
>
> Fourth, we have incorporated the expanded empirical results requested during the review process in Appendix A.20, A.21 and A.22. This includes broader ablations across multiple datasets and coarsening ratios, additional baselines, large-scale results, corrected uncertainty estimates for link prediction, corrected table entries, and softened runtime claims. The revised runtime discussion now presents ENGC as competitive and efficient among optimization-based coarsening methods rather than uniformly faster than all alternatives.
>
> Fifth, we have revised the empirical framing throughout the manuscript to be more conservative. The revised manuscript presents ENGC as a competitive and interpretable entropy-guided coarsening method that directly controls label mixing within supernodes, rather than claiming uniform dominance across all datasets, ratios, and baselines. We also retain the explicit separation between entropy-specific effects and general perturbation-style structural guarantees.
>
> Finally, we have completed the remaining notation and presentation cleanup requested by the reviewers and the Action Editor, including the notation for the lifted Laplacian, the descriptions of Algorithms 2 and 3, the log-base convention in entropy bounds, and several minor grammar, spacing, and formatting issues.
>
> We thank the Action Editor and reviewers again for their constructive feedback. The requested revisions have significantly improved the clarity, correctness, and presentation of the final manuscript.